# Particulate inorganic carbon pools by coccolithophores in low oxygen/low pH waters off the Southeast Pacific margin

Francisco Díaz-Rosas[1,2], Cristian A. Vargas[2,3], and Peter von Dassow[1,2]

[1]Facultad de Ciencias Biológicas, Departamento de Ecología, Pontificia Universidad Católica de Chile, Santiago, Chile
[2]Millennium Institute of Oceanography (IMO), Universidad de Concepción, Concepción, Chile
[3]Coastal Ecosystems & Global Environmental Change Lab (ECCALab), Department of Aquatic Systems, Faculty of Environmental Sciences, Universidad de Concepción, Concepción, Chile

*Correspondence to: Francisco Díaz-Rosas (fjdiaz@bio.puc.cl)*

**Abstract.** A predicted consequence of ocean acidification is decreasing coccolithophore-produced Particulate Inorganic Carbon (PIC) pools. PIC is thought to enhance the sinking of Particulate Organic Carbon (POC) to deeper waters, potentially influencing the depth of organic matter remineralization and subsurface oxygen levels. To explore these potential feedbacks, we examined the relationships between PIC, coccolithophores, carbonate chemistry, and dissolved oxygen in the Southeast Pacific open-ocean oxygen minimum zone–a region characterized by naturally low dissolved oxygen, low pH, and high $p$CO$_2$ levels. Measurements of PIC and coccolithophore abundance from late-spring 2015 and mid-summer 2018 revealed that coccolithophores, particularly *Gephyrocapsa (Emiliania) huxleyi*, were major contributors to PIC through the shedding of coccoliths. On average, about half of the PIC was attributed to reliably enumerated coccospheres and detached coccoliths, with significantly diminished pools below the euphotic zone. Temperature, oxygen, and pH emerged as key factors influencing PIC variability. PIC pools and PIC:POC ratios in both surface and subsurface waters in this naturally low pH/low O$_2$ zone are lower than available data from most oceanic regions, with the exception of the Western Arctic. Our findings support the prediction that in upwelling regions with a shallow oxygen minimum zone, POC production is promoted by phytoplankton other than PIC-producing coccolithophores due to the injection of nutrient-rich but low-pH water. This process decreases PIC:POC ratios, suggesting that the role of PIC in POC sedimentation might be decreased under such conditions. We emphasize that comparing PIC dynamics across diverse upwelling systems will be valuable for understanding how low pH and O$_2$ conditions influence its role in POC fluxes.

## 1 Introduction

The Particulate Inorganic Carbon (PIC) pool is a key component of marine carbon cycles and atmospheric reservoirs (Ridgwell and Zeebe, 2005). It originates from various sources, including land-derived inputs to coastal margins (Cai, 2011) and biological processes such as phytoplankton calcification (Taylor and Brownlee, 2016). Coccolithophores, particularly the cosmopolitan species *Gephyrocapsa (Emiliania) huxleyi*, significantly contribute to PIC through coccolith production.

Coccoliths are thought to enhance the sinking of organic matter (i.e., the 'ballast effect'), facilitating the export of Particulate Organic Carbon (POC) to deeper waters (e.g., Klaas and Archer, 2002), thereby influencing carbon sequestration and nutrient cycling in the global pelagic ocean (Monteiro et al., 2016; Balch, 2018). Changes in PIC dynamics, driven by shifts in coccolithophore communities across spatial and temporal scales, feedback into the ocean-atmosphere system (Balch et al., 2016; Claxton et al., 2022). For example, ocean acidification (OA)–characterized by decreasing pH and carbonate ion concentration–can impair coccolithophore calcification (Riebesell et al., 2000; Barcelos e Ramos et al., 2010; von Dassow et al., 2018; Kottmeier et al., 2022; von Dassow, 2022). This could lead to diminished $CaCO_3$ fluxes to the seafloor, increasing organic matter respiration and potentially decreasing both organic carbon flux and mid-water oxygen levels (Hofmann and Schellnhuber, 2009; Zhang et al., 2023). In this context, understanding the relationships between coccolithophore PIC, pH/$p$CO$_2$, and oxygen in natural oxygen minimum zones (OMZs)–where pH and $O_2$ levels can reach extremely low levels– is of particular interest.

While PIC measurements are more extensive in the Atlantic Ocean and the Atlantic sector of the Southern Ocean, the Pacific Ocean remains underrepresented in PIC and coccolithophore data. However, studies on coccolithophore distributions are increasing in the Indian Ocean, the subpolar Pacific, and the Southern Ocean (Balch et al., 2018; Oliver et al., 2023). Early satellite imagery suggested that the Pacific hosts relatively lower surface PIC concentrations compared to the Atlantic (Brown and Yoder, 1994). However, higher subsurface PIC concentrations–at depths beyond the detection limits of satellites– have been linked to events of higher coccolithophore abundance in remote Southeast Pacific waters (Beaufort et al., 2008; Oliver et al., 2023). A particularly notable basin-scale coccolithophore feature, the Great Calcite Belt, spans the Southern Ocean and is primarily sustained by *G. huxleyi* growth (Balch et al., 2016; Balch and Mitchell, 2023). This feature advances southward each year, peaking in the austral summer (Hopkins et al., 2019). Temperature, competition with diatoms for nutrients, and Fe availability have been proposed as factors controlling coccolithophore distributions in the Southern Ocean (Oliver et al., 2023 and references therein).

Vast OMZs are persistent features of the tropical and subtropical Eastern Pacific (Schmidtko et al., 2017), where low pH and high $p$CO$_2$ levels are common (Torres et al., 2002, 2011; Beaufort et al., 2011; Vargas et al., 2017; Vargas et al. 2021). Although such conditions have been shown to inhibit coccolithophore growth and calcification in laboratory settings, field observations suggest more complex relationships (Beaufort et al., 2011; Müller et al., 2015; von Dassow et al., 2018). There is still very limited data on how PIC, POC, pH, and $O_2$ interact within OMZ systems. Here, we present PIC measurements alongside coccolithophore counts and estimates of coccolithophore-derived PIC calculated from them, based on sampling conducted in spring 2015 and summer 2018 off the Southeast Pacific margin (~ 20-35º S). We estimate the proportion of total PIC (PIC$_{Total}$) attributed to coccospheres and detached coccoliths pools and examine how coccolithophore-produced PIC (PIC$_{Cocco}$) correlates with various physical and chemical parameters, with a particular focus on low pH/high $p$CO$_2$ conditions associated with oxygen-deficient waters. Additionally, we discuss other *in situ* variables influencing coccolithophore growth and compare PIC concentrations in the Southeast Pacific with those reported in other oceanic regions.

## 2. Materials and Methods

### 2.1 Sampling

The sampling consisted of two cruises conducted in waters off the Southeast Pacific margin, off northern Chile (Fig. 1a). In spring 2015 (LowpHOx 1), we vertically screened six stations along an inshore-offshore transect off Iquique (~20º S; sts. T1-T6), along with six stations arranged latitudinally between 22º S and 29.5º S (sts. L1-L6). During summer 2018 (LowpHOx 2), three of the inshore-offshore stations were replicated (sts. T1, T3, T5), along with seven stations sampled southward down to ~34º S (sts. Lander and Hydro; Fig. 1a). In depth, the sampling crossed both the euphotic zone and OMZ core (Fig. 1b; Fig. 2c-e, Fig. 3 k-o). The predominant oceanographic conditions during these two cruises, as reported and discussed by Vargas et al., (2021), provide the context for the new data presented here.

At each station, discrete seawater samples were collected, filtered, stored, and moved to the lab for the determination of PIC (section 2.2) and coccolithophore standing stocks (section 2.3). Due to seawater availability onboard in 2018, sampling for coccolithophores only reached 100 m, except for station T5, which reached a depth of 2000 m. This contrasts with 2015 sampling reaching equal or greater 350 m depths (max. 1000 m in T3 and T4). Complete PIC measurements and coccolithophore abundances are provided in the Supplement (Tables S1-S2). As ancillary data, 1-m averaged SST, SSS, oxygen, and fluorescence continuous profiles, as well as, discrete profiles of Chl-$a$, nutrients, Particulate Organic Carbon (POC), and carbonate system were included (see methods in Vargas et al., 2021). These physical and chemical data were extracted from Vargas et al. (2023a, b, c). The euphotic depth ($Z_{eu}$) was estimated by calculating the depth at which the Photosynthetically Active Radiation (PAR) is reduced to 1 % of surface levels, using the attenuation coefficients ($K_d$490) provided by the Bio-Geo-Chemical products based on the Copernicus-GlobColour processor (Copernicus-GlobColour, 2023). The averaged $K$d(490) values (3 x 3 pixels and 3-day average) were converted into $K_d$(PAR) values following the model of Morel et al. (2007), and the 4.6 optical depth calculated as ln(100)/$K_d$(PAR) (Morel, 1988). Where available, a good fit was found between the euphotic zone depth derived from satellite observations and PAR obtained from a sensor attached to the CTD (Fig. S1). Lastly, the OMZ core was defined as the water layer where dissolved $O_2$ was below 20 µmol kg$^{-1}$, a commonly used threshold (e.g., Gilly et al., 2013), which we use here also to approximate the depth of the base of the upper oxycline.

### 2.2 Particulate inorganic carbon standing stocks

We produced synoptic maps of satellite-derived PIC in the Southeast Pacific margin (Balch and Mitchell, 2023). Monthly and weekly PIC climatologies (November-December 2015 and January-February 2018) were obtained from the MODIS-Aqua mission (NASA Ocean Biology Processing Group, 2023). The data were then converted from mol CaCO$_3$ m$^{-3}$ to µg C L$^{-1}$ by multiplying by 11910.69 and plotted using RStudio.

For *in situ* measurements of PIC$_{Total}$, we followed a slightly modified version of the procedure introduced by Poulton et al. (2006). In summary, we filtered between 0.1-1.5 L of seawater (increasing with sampled depth) onto 25 mm polycarbonate filters with a 0.4 $\mu$m pore size. Prior to removal from vacuum, filters were immediately rinsed with a squirt of potassium

tetraborate solution in 2015 and with seawater alkalinized with ammonium in 2018 (to maintain pH > 8.0 while in storage) and stored in metal-free Falcon tubes at −20º C until transit to the lab. Subsequently, $Ca^{2+}$ was extracted with nitric acid and quantified using Inductively Coupled Plasma Atomic Mass Spectrophotometry facilities at the Bigelow Laboratory for Ocean

Sciences. A correction was made to correct for potential excess of $Ca^{2+}$ due to $Na^+$ residues that might be left on the filters. This calculation indicated that residual seawater contributed on average 29.1 % ± 25.8 % of total $Ca^{2+}$ in LowpHOx 1 samples and on average 35.2 % ± 21.5 % of total $Ca^{2+}$ in LowpHOx 2 samples. The PIC concentrations were expressed in $\mu g\ C\ L^{-1}$. Additionally, we examined the association between calcification (PIC) and POC (from Vargas et al., 2021) using PIC:POC ratios. PIC:POC ratios were calculated using both $PIC_{Cocco}$ and $PIC_{Total}$ values recorded in 2015 and 2018, which mostly showed

similar trends, except for a significant discrepancy at station L4 in 2015 (see Fig. S2). These ratios were then categorized into two groups: above and within the OMZ core, to assess the influence of the OMZ on PIC and POC concentrations.

Data from this study were compared against those reported for other open ocean or coastal margins (see Balch et al., 2018). PIC and POC data were obtained from the SEABASS (Werdell et al., 2003) and BCO-DMO repositories (Balch, 2010). Depth intervals were chosen to balance of competing criteria aiming to detect broader ecological patterns robustly. We aimed to be

roughly comparable with the categories of above the oxycline (mostly corresponding to the euphotic zone, where coccolithophores are growing) and below the oxycline, where coccolithophores are presumed present entirely due to sinking from the surface. However, OMZ systems are highly stratified, and eukaryotic phytoplankton growth is excluded from below the lower half of the oxycline (which also corresponds to a strong pycnocline) even when sufficient light penetrates for photosynthesis (Wong et al., 2023). In contrast, in non-OMZ systems, the lower limits of growing phytoplankton are less

constrained, and coccolithophores are part of the "shade flora" (Balch, 2018). Therefore, data were binned over 0-100 m depths to represent the surface, and over 100-400 depths to represent the subsurface.

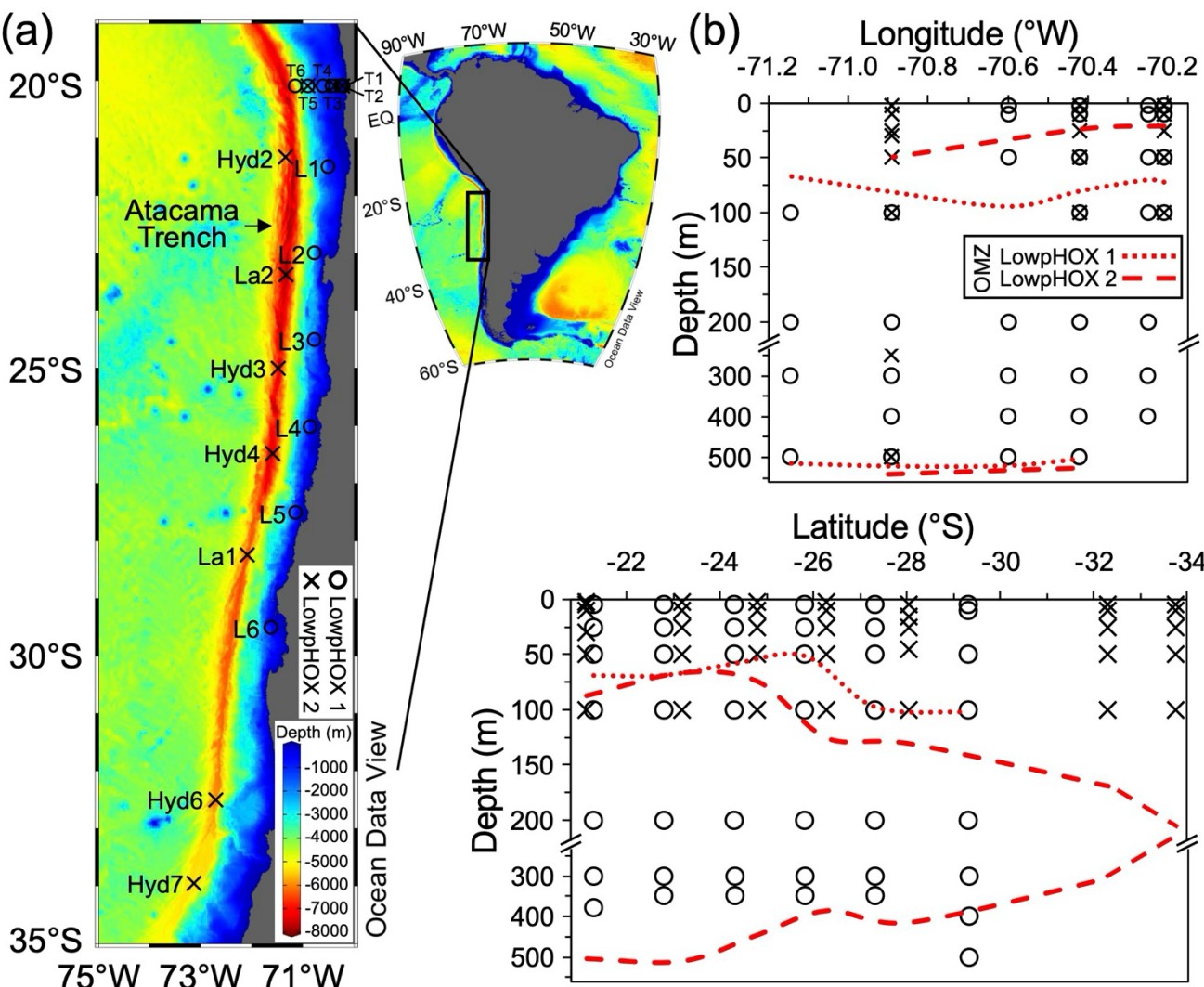

**Figure 1: (a) Map of the Southeast Pacific margin showing the study site and stations sampled during late-spring 2015 (circles) and mid-summer 2018 (crosses). (b) Sampling depth coverage for coccolithophores, highlighting areas crossing the OMZ core thresholds of 20 μmol kg⁻¹ (dotted and dashed red lines). Map produced by Ocean Data View (Schlitzer, 2024), with bathymetry based on the GEBCO chart (GEBCO, 2023).**

## 2.3 Coccospheres and detached coccoliths standing stocks

For enumeration of coccospheres and detached coccoliths, between 0.1 to 1.0 L of seawater (increasing with depth) were filtered onto 25 mm polycarbonate filters with a 0.8 $\mu$m pore size, left to dry at room temperature in Petri dishes, and stored with desiccant until microscopy analyses. Total coccosphere counts were conducted on filter slide preparations with oil immersion, using cross-polarized light microscopy (LM; Zeiss, Axioscope 5). The analysis of 20 fields of view at 400x magnification covered 5.1 mm² of the filter area, corresponding to a range of 1.9-16.3 mL of seawater analysed. For counts of

total detached coccoliths, 11 fields of view per filter were screened (224 x 165 $\mu$m per frame) at 630x magnification (oil immersion objective), covering 0.41 mm$^2$ of the filter area, corresponding to total volumes of 0.2-1.3 mL of seawater analysed. On average, $171 \pm 307$ coccospheres (range: 2-2099) and $708 \pm 664$ coccoliths (range: 5-4067) were analyzed per sample. An issue arose where some filters from inshore-offshore 2015 sampling (20$^{\circ}$ S; Stations T1-T6) exhibited excessive brightness under LM, for which counts were made through scanning electron microscopy (SEM) analysis (Quanta FEG 250) as described in Díaz-Rosas et al. (2021). For the quantification of coccosphere abundances (see equation in Díaz-Rosas et al., 2021), between 28-48 images taken at 800-1500x magnification were examined per filter, covering from 5 to 6 mm$^2$ of the filter area corresponding to a range of 2.1-18.4 mL of seawater analyzed. For total detached coccolith abundances, between 4-5 images were examined per filter, covering from 0.6 to 1.0 mm$^2$ of the filter area corresponding to a range of 0.2-2.8 mL of seawater analyzed. On average, $166 \pm 280$ coccospheres (range: 1-1141) and $861 \pm 879$ detached coccoliths (range: 58-3815) were counted per sample. Layers of coccoliths detached from *G. huxleyi* (Fig. S3a-d) were added to the detached-coccolith counts. Collapsed coccospheres were included when they remained mostly intact, but when more disintegrated could not be accurately counted, especially as they were often less reflective than intact coccospheres and coccoliths (Fig. S3e-h). In a subset of samples, collapsed coccospheres were estimated to contribute < 21 % (min. = 0 %, average = 7.1 %) of the total number of coccospheres. As expected, the standard error of the means (among the images obtained from the same sample) drops hyperbolically with the total number of coccospheres or coccoliths counted, whether with LM or SEM (Fig. S4), but remains higher in SEM due to the smaller size of SEM images. To check for differences between counts obtained through LM and SEM examination, five samples with varying coccolithophore abundances were analyzed with SEM as outlined above, revealing the slopes were highly linear with R$^2$ greater than 0.9, and were not significantly different from 1, while the intercepts were close to 0 (Fig. S5), allowing for counts from the two methods to be combined.

## 2.4 Diversity of coccospheres and detached coccoliths

The identification of coccolithophores and detached coccoliths by LM is sometimes limited, which may impact the estimation of coccolithophore-derived PIC relative to total PIC, as it relies on the estimated PIC quotas of coccospheres and coccoliths. To understand this effect, taxonomic classification by SEM was performed on samples from the LowpHOx 1 cruise (2015), focusing on samples from T1 to T6 as well as selected samples from L1 (at 5 and 25 m), L2 (at 5 and 50 m) and L3 (at 5 m). Between 6-11 and 4-5 images per filter, depending on magnification (ranging from 800x to 1500x), were examined for coccosphere and coccolith classification, respectively, following the classification in Young et al. (2003). To estimate the absolute abundances of coccospheres and detached coccoliths at the species or genus level, the relative abundance of each species or genus was multiplied by the absolute abundance of coccospheres and coccoliths counted by SEM (see section 2.3). Due to limitations in SEM time, it was not always possible to zoom to higher magnification to differentiate between *G. parvula* and *G. ericsonii* coccospheres, so they were merged into the *G. parvula/ericsonii* category. The only character distinguishing the coccoliths is a small bridge present in *G. ericsonii* and not *G. parvula*, which would represent a very minor effect on PIC quotas. Also, the grouping of the two species is phylogenetically supported (Bendif et al., 2016, 2019). Moreover, as some of

the small coccoliths detached from *G. parvula/ericsonii* might be overlooked (2.0 μm in length), the few coccoliths found in distal-shield view, and the coccoliths from *G. huxleyi* (3.6 μm in length) were merged into the *Gephyrocapsa* < 4 *μ*m category. Larger coccoliths (> 5 μm in length) were classified into species or genus, being subtracted from total counts of coccoliths

obtained in the same SEM image to obtain the number of coccoliths < 4 μm. Most rare detached coccoliths were grouped in the miscellaneous category that include *Syracosphaera* spp., *Acanthoica* spp., *Discosphaera tubifera*, *Umbellosphaera* spp., and *Umbilicosphaera* spp. Moreover, the counts of detached coccoliths from *Calcidiscus leptoporus* might contain a few coccoliths of *Oolithotus* spp. as it was not always possible to differentiate them. In all surface samples taken in 2018, as well as in samples L4 to L6 from 2015, species dominating the coccosphere and coccolith pools were identified using LM (40x and

63x objectives, Zeiss Axioscope 5) following Frada et al. (2010). Rarefaction and extrapolation analyses in RStudio's iNEXT package (Hsieh et al., 2024) assessed how sampling effort affects coccolithophore diversity using two indices: species richness and the exponential of Shannon entropy.

**2.5 Coccolithophore specific PIC quotas and estimation of coccolithophore-derived PIC**

To obtain the coccolith mass (in pg CaCO$_3$), the cube of the mean distal-shield length of coccoliths (= biovolume in $\mu$m$^3$)

measured in the SEM images (see Table S3) was multiplied by the respective taxonomic-specific shape factor $K_s$ (Young and Ziveri, 2000), and by the density of calcite (2.7 g cm$^{-3}$). The number of coccospheres was converted to coccoliths by using the compilation of coccoliths per coccosphere in Yang and Wei (2003). For *G. huxleyi*, the 17 coccoliths per coccosphere determined by Beaufort et al. (2011) in the study site was applied in agreement with those seen in SEM images. We did not account for variation in coccoliths per coccosphere or in mass content per coccolith among *G. huxleyi* morphotypes. The

PIC$_{Cocco}$ standing stock (in $\mu$g C L$^{-1}$) is then given by the sum of coccolith mass composed by coccospheres and detached coccoliths. Given the lack of taxonomic resolution for the 2018 samples, the *G. huxleyi* PIC quota (mass = 2.5 pg CaCO$_3$ per coccolith; see Table S3) was applied as the maximum threshold to all samples for consistency, a scenario justified as *G. huxleyi* or *G. huxleyi*-sized coccolithophores were dominant or co-dominant in both the 2015 and 2018 samples (see below). Notably, PIC$_{Cocco}$ quotas estimated using *G. huxleyi* as a uniform reference closely aligned with those derived from all-taxa coccolith

mass factors across the 2015 T1-T6 transect (Fig. S6).

**3 Results**

**3.1 Oceanographic conditions**

Surface (mixed) layer temperatures exceeded 21 °C to the north and offshore, while surface temperatures were below 17 °C near the coast, in the north and south of 21° and 24º S latitude in 2015, and south of 29º S in 2018 (Fig. 2a). The pycnocline

(Fig. 2b) and oxycline (Fig. 2c) roughly paralleled the thermocline in both years. In 2015, the oxycline was always in the upper 50 m. In 2018, the depth of the oxycline increased to between 50 and 75 m south of 26° S. The estimated Z$_{eu}$ was often near

the base of the oxycline (Fig. 2c). At all stations, the oxycline and $Z_{eu}$ were always shallower than 100 m. In these well-illuminated waters, peaks of phytoplankton biomass were observed in both 2015 and 2018 (Chl-$a$ > 3 mg m$^{-3}$ and fluorescence > 5 mg m$^{-3}$; Fig. 2d-e).

Nitrate and phosphate levels were low above the pycnocline (< 2 and < 1 µM, respectively), and increased below (to approximately 20 µM and 2.5 µM respectively; Fig. 2f-g). Both pH and $\Omega_{calcite}$ declined sharply while $p$CO$_2$ increased with depth through the oxycline (Fig. 2h-j). Increased nitrate coincided with higher $p$CO$_2$ and lower pH in the 100 m depth waters sampled in 2015 and 2018 (Fig. S9). In this upwelling zone, high nitrate levels are associated with lower pH (Fig. S9b). At depth, decomposition depletes oxygen while releasing nitrates and CO$_2$, and as nitrate-rich waters rise to the surface, they

bring low oxygen/low pH and high $p$CO$_2$ with them (Fig. S9c-e). As expected, dissolved O$_2$, pH and $\Omega_{calcite}$ decreased as $p$CO$_2$ and nitrate increased (Fig. S9f-i).

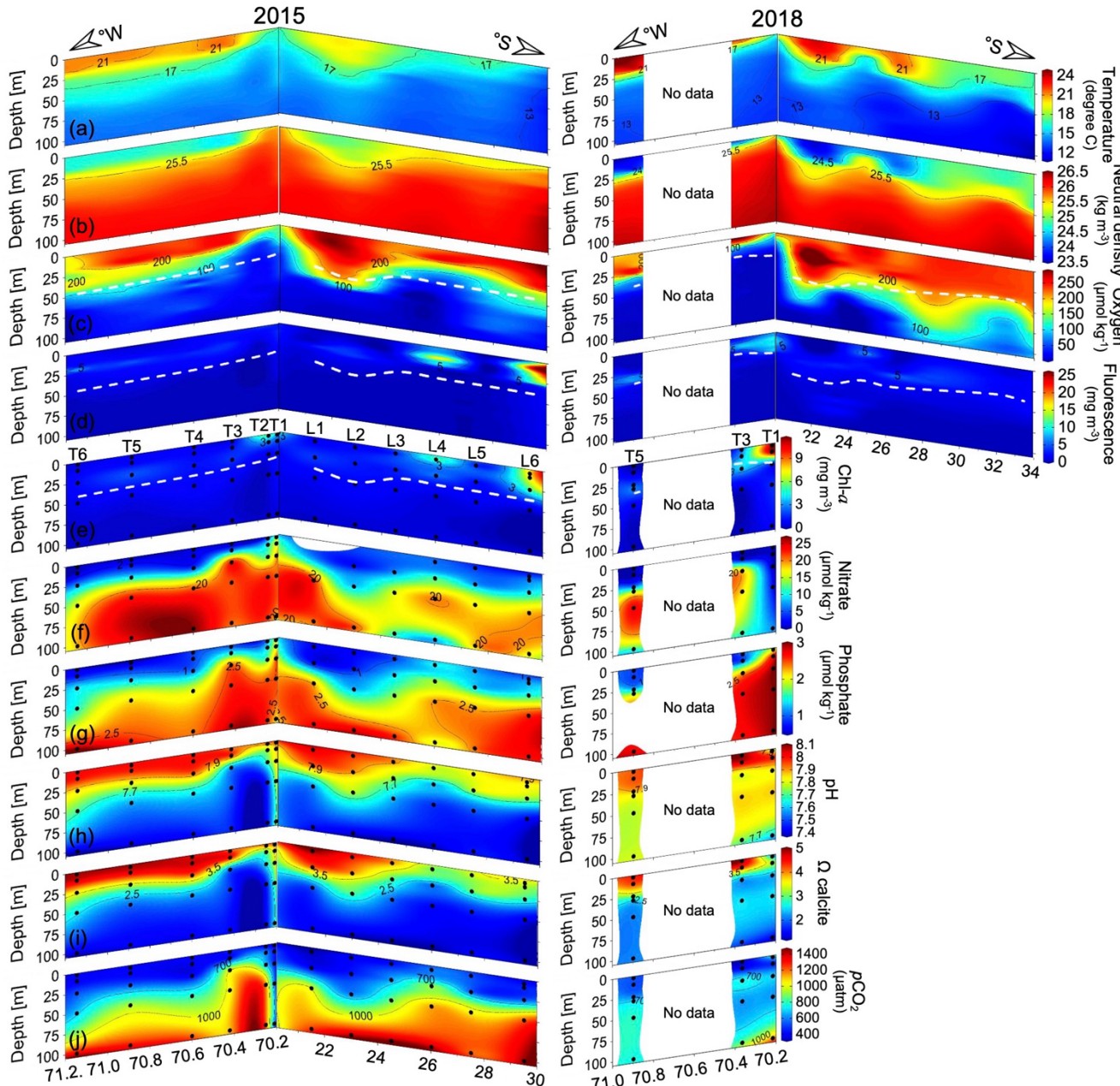

**Figure 2. Spatial variation in physical-chemical-biological parameters recorded during late-spring 2015 (left) and mid-summer 2018 (right). Temperature (a), neutral density (b), oxygen (c), fluorescence (d), Chl-*a* (e), nitrate (f), phosphate (g), pH (h), $\Omega_{calcite}$ (i), and *p*CO₂ sections (j) versus long-lat and depth with sample locations (solid black circles). Continuous profiles shown in (a-d) are 1-m binned. Dashed white lines indicate the estimated euphotic depth in plots c-e. Deeper profiles of the variable in plot e are provided in Supplementary Figures S7-S8.**

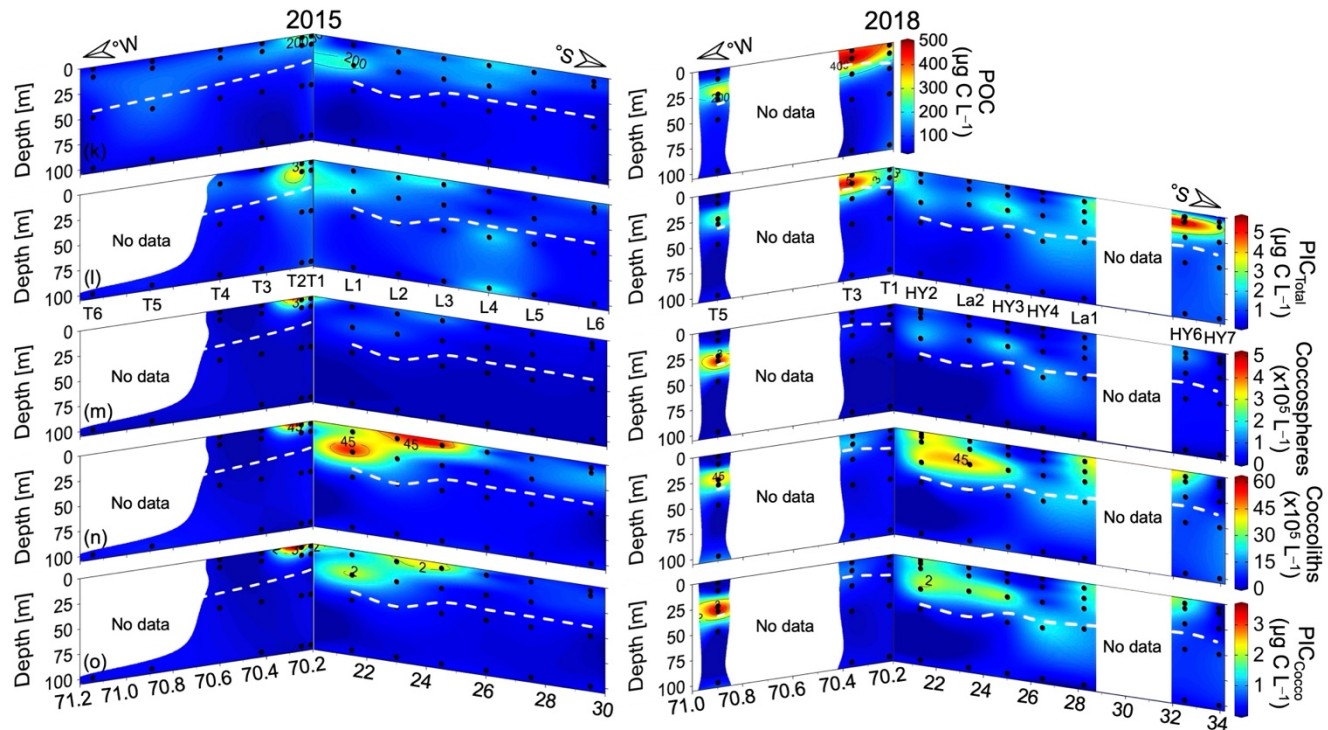

**Figure 3. Spatial variation in POC, PIC$_{Total}$ and coccolithophore standing stocks and their contribution to PIC (PIC$_{Cocco}$) recorded during late-spring 2015 (left) and mid-summer 2018 (right). POC (k), PIC$_{Total}$ (l), coccospheres (m), detached-coccoliths (n), and PIC$_{Cocco}$ (o) sections versus long-lat and depth with sample locations (solid black circles). Dashed white lines indicate the estimated euphotic depth. The PIC$_{Cocco}$ pool estimation shown in (o) is derived from abundances obtained through scanning electron microscopy and cross-polarized light microscopy. POC = Particulate Organic Carbon. PIC = Particulate Inorganic Carbon. Deeper profiles of the variables in plots l-o are provided in Supplementary Figures S7-S8.**

## 3.2 Standing stocks of coccolithophores, pools of PIC and POC, and PIC:POC ratios

In late-spring 2015, the highest abundances of coccospheres (> 3.0 x$10^5$ L$^{-1}$) and detached coccoliths (> 45 x$10^5$ L$^{-1}$), along with PIC$_{Total}$ (> 3 μg C L$^{-1}$) and PIC$_{Cocco}$ pools (> 2 μg C L$^{-1}$), were found in surface or near-surface waters (depths < 25 m) closer to the coast at ~20° S (stations T1-T2), extending south to ~24° S (stations L1-L3; Fig. 3l-o). Two years later, in mid-summer 2018, the highest surface stocks of coccolithophores (> 5 x$10^5$ coccospheres L$^{-1}$), PIC$_{Total}$ (5.3 μg C L$^{-1}$), and PIC$_{Cocco}$ (> 3 μg C L$^{-1}$) were observed at ~20° S (stations T3 and T5). Additionally, events of relatively higher coccolithophore abundance and elevated PIC values (> 2 μg C L$^{-1}$) were recorded farther south (stations Hyd2-La2 and La1-Hyd6-Hyd7; Fig. 3l-o). Notably, a surface peak in PIC$_{Total}$ was observed at the southern station Hyd6 (33° S), reaching 5.86 μg C L$^{-1}$. Across all samples, POC ranged from 22.7 to 250.0 μg C L$^{-1}$ in 2015 and from 26.1 to 501.6 μg C L$^{-1}$ in 2018 (Fig. 3k). POC correlated closely with Chl-*a* (Fig. S10a), and the two-orders-lower PIC$_{Total}$ varied significantly with POC (Fig. S10b). Considering all stations and depths, PIC$_{Total}$ ranged from 0.18 to 3.81 μg C L$^{-1}$ in 2015 and from 0.08 to 5.86 μg C L$^{-1}$ in 2018 (Fig. 3l). Coccosphere abundances reached maxima of 3.9 x$10^5$ and 5.2 x$10^5$ L$^{-1}$, and abundances of detached coccoliths reached maxima of 63 x$10^5$ and 49 x$10^5$ L$^{-1}$ in spring 2015 and summer 2018, respectively (Fig. 3m-n). Overall, coccosphere abundance tended

to vary directly with detached coccolith abundance, although with a high scatter ($y = 13.270(\pm1.123)x + 6.740(\pm0.874)$; $R^2_{adjusted} = 0.51$; Fig. S11a). Above the $Z_{eu}$, the average ratio of coccoliths to coccospheres was 40 in 2015 (range: 4-104; $n = 21$) and 33 in 2018 (range: 6-62; $n = 31$) (Fig. S11b). Only one sample had ratio higher than 100. The ratio of detached coccoliths to coccospheres did not relate either to coccosphere abundance or to $PIC_{Cocco}$, and the four samples with the highest coccosphere abundance and $PIC_{Cocco}$ all showed ratios less than 20 (Fig. S11c-d). The estimated PIC pools produced by coccolithophores ranged from 0.02 to 3.60 µg C L$^{-1}$ in 2015 and from 0.01 to 3.69 µg C L$^{-1}$ in 2018 (Fig. 3o). Coccosphere and coccolith abundances, as well as PIC pools, diminished substantially below 50 m depth ($\sim Z_{eu}$) (Fig. 3m-n, S7-S8). Surface PIC:POC ratios (depths < 30 m) ranged from 0.002 to 0.030 in 2015 (mean = 0.011, n = 18) and from 0.001 to 0.014 in 2018 (mean = 0.007, n = 10) (Fig. S2; Tables S1-S2).

### 3.3 Effectiveness of *in situ* sampling at capturing spatial variability in PIC indicated by satellite

During the 2015 sampling period (Fig. 4a-d), satellite-derived PIC concentrations exhibited notable spatial variability, with peaks (> 10 µg C L$^{-1}$) near 20° S effectively captured by sampling locations. Notably, in both years *in situ* and satellite-derived PIC levels were relatively high off ~20° S (Fig. 3-4), underscoring this sector as a local hotspot for coccolithophore PIC production. Patches of potentially higher PIC indicated by satellite between 26° S and 32° S (Fig. 4a-d) were clearly not reflected in *in situ* PIC data in 2015 (possibly due to low coverage and patchiness) (Fig. 3l-o). The patches of high PIC indicated by satellite in a similar latitudinal band in 2018 (Fig. 4e-h) were partially reflected by higher near surface *in situ* $PIC_{Total}$ and coccoliths at 28°S and 32°S (Fig. 3l-o). Deeper euphotic peaks in coccoliths or $PIC_{Total}$, such as those observed in 2018 near 23°-25°, might correspond to a more southerly and subsurface expression of the high PIC detected by satellite-derived PIC north of 24° during the same cruise.

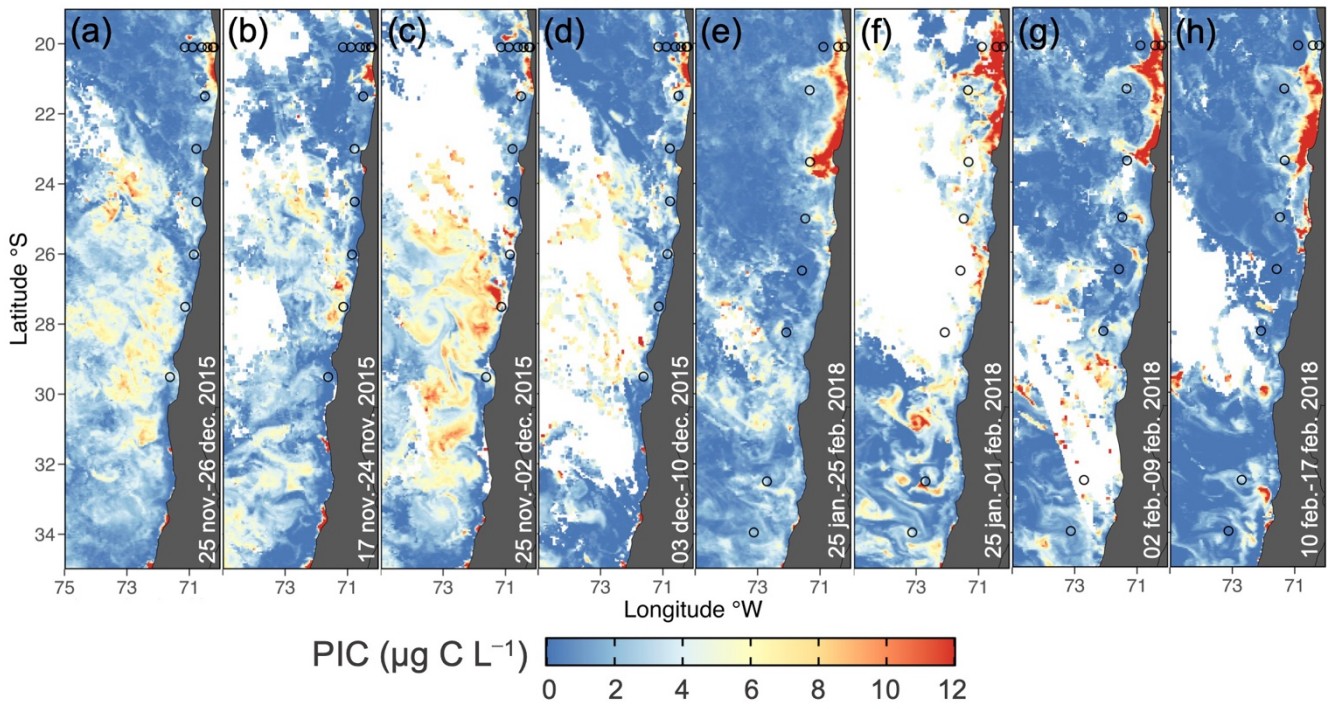

**Figure 4. Monthly (a) and weekly (b-d) satellite-PIC climatologies during the LowpHOx 1 sampling (27-28 nov./05-09 dec. 2015; open circles), along with monthly (e) and weekly (f-h) satellite-PIC climatologies for the LowpHOx 2 sampling (30 jan./03-09 feb./12-13 feb. 2018; open circles).**

### 3.4 Diversity of coccospheres and detached coccoliths

In 2015, *G. huxleyi* was the numerically dominant coccolithophore south of 20° S (Fig. S12-S13). It co-dominated with *G. parvula/ericsonii* the coccosphere standing stocks at 20° S, where there was no consistent variation in the relative abundance of these taxa (Fig. 5a). Despite the presence of *G. parvula/ericsonii*, its small coccoliths were underrepresented, reinforcing the overwhelming prevalence of *G. huxleyi* in the samples (Fig. 5b). The low diversity of both coccospheres and detached coccoliths is supported by the rapid saturation of rarefaction curves from samples at different stations, grouped by depth layers (Fig. S14). The relative abundance of larger taxa (principally *G. oceanica* in stations T1-T2, *Helicosphaera* spp. in T3, and *C. leptoporus* in T4) increased below the $Z_{eu}$, reaching 0-18 % and 0-15 %, respectively (Fig. 5), as total coccosphere and coccolith abundances declined sharply.

Only polarized LM data was available in 2018 (Fig. S13), and moderately-sized *Gephyrocapsa* spp. dominated coccolithophore communities in all 2018 samples, as in 2015. LM cannot reliably distinguish *G. huxleyi* from *G. parvula/ericsonii* or *G. muellerae* (the bridge in *G. ericsonii* is too small to reliably see in LM, and in both *G. muellerae* and *G. ericsonii* the bridge can be absent from some coccoliths) but qualitative LM observations suggested that coccospheres and coccoliths smaller than *G. huxleyi* were very rare.

**3.5 Distribution of standing stocks of coccolithophore-derived PIC**

After converting the coccospheres and detached coccoliths abundances recorded at ~20º S in 2015 to coccolith-PIC values allometrically, we observed that the majority $PIC_{Cocco}$ (> 2 µg C $L^{-1}$) was produced near the coast in depths shallower than 30 m (Fig. 6a). Below $Z_{eu}$, the sharp decline in the numerical abundances of coccospheres (< 0.1 x$10^5$ $L^{-1}$) and detached coccoliths (< 10 x$10^5$ $L^{-1}$) decreased coccolithophore PIC pools to below 1 µg C $L^{-1}$ (Fig. 6a).

Above the $Z_{eu}$, the coccospheres and detached coccoliths of *G. huxleyi* and *G. ericsonii/parvula* dominated the $PIC_{Cocco}$ pools (Fig. 6b). In contrast, detached coccoliths of numerically rarer taxa, such as *C. leptoporus*, *Helicosphaera* spp., and *G. oceanica*, contributed significantly to $PIC_{Cocco}$ below the $Z_{eu}$, often exceeding the contributions of the smaller *Gephyrocapsa* species at most stations (Fig. 6c).

Regarding the $PIC_{Cocco}$ dataset, a significant positive linear relationship was found between the $PIC_{Total}$ and $PIC_{Cocco}$ values both in the upper 100 m and across all sampled depths (Fig. 7a-b). In all but three cases, the calculated $PIC_{Cocco}$ values were greater than the measured $PIC_{Total}$. These exceptions corresponded to a high total numerical abundance of *G. huxleyi*. Another notable exception was station L1 at 25 m, where *C. leptoporus* and *Helicosphaera* spp. contributed 51 % and 20 % of the coccosphere PIC quota (2.23 µg C $L^{-1}$), and the estimated total $PIC_{Cocco}$ (5.1 µg C $L^{-1}$) was about twice the measured $PIC_{Total}$.

On average, coccospheres plus detached coccoliths were estimated to account for 30-48 % of the $PIC_{Total}$ (Fig. 7c). In the upper 100 m, detached coccoliths contributed more to $PIC_{Cocco}$ pools than coccospheres, but coccospheres still contributed 37 % of total accounted $PIC_{Total}$ (Fig. 7c). Below 100 m, the contribution of coccospheres was less than 10 % that of detached coccoliths (Fig. 7c).

In a subset of samples from T1-T6 in 2015 analysed by higher magnification SEM for better taxonomic resolution, $PIC_{Cocco}$ estimations using taxa-specific conversions generally showed little differences with $PIC_{Cocco}$ estimations made with the assumption that all coccospheres and coccoliths had PIC quotas similar to *G. huxleyi* (Fig. S6a). Substantial differences were only seen at depths of 5 m at station T1 and 300 m at station T5 (Fig. S6b, f), related to the high contribution of *C. leptoporus* and *Helicosphaera* spp. to $PIC_{Cocco}$ at those stations.

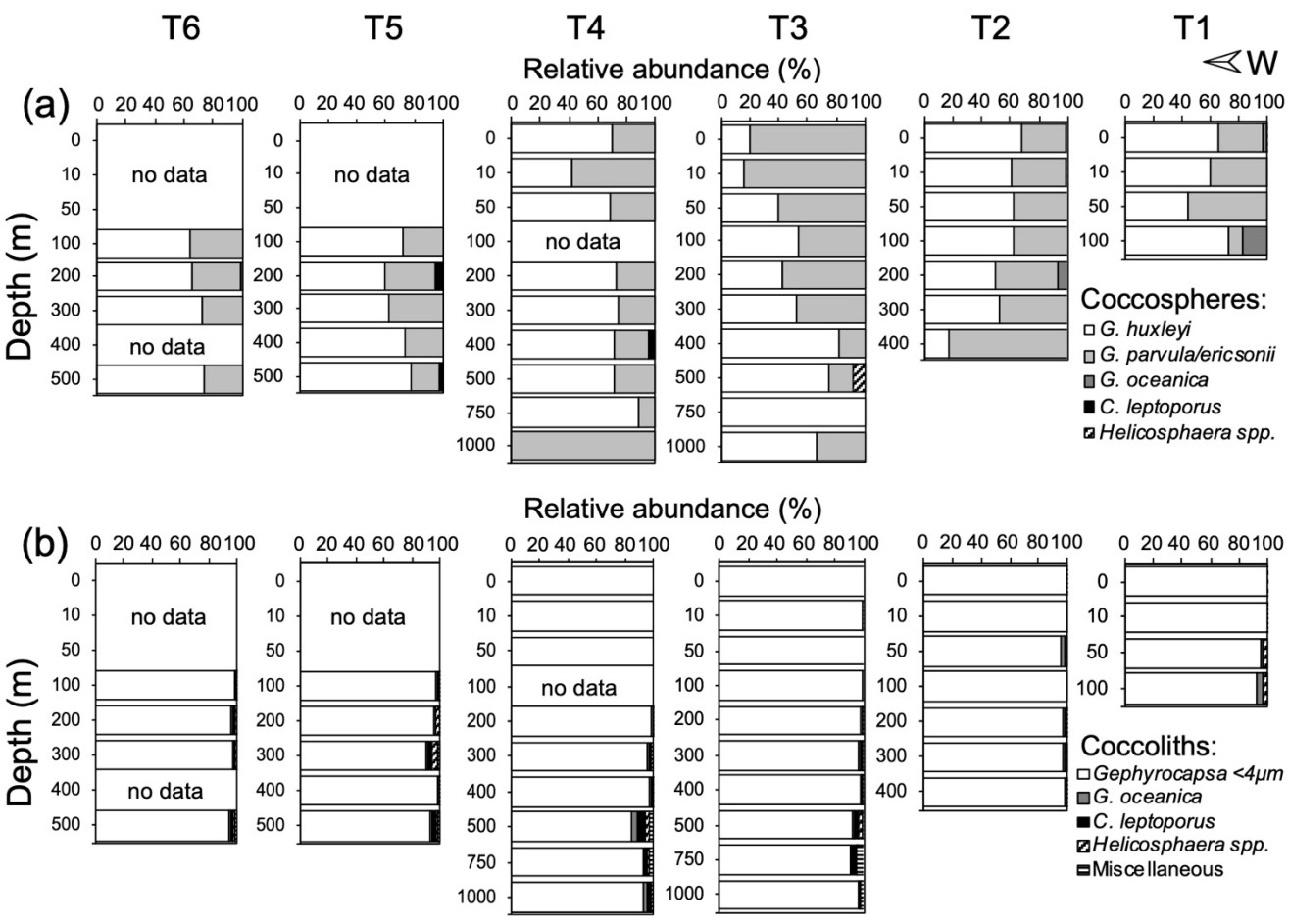

**Figure 5: Coccospheres (a) and detached coccoliths (b) relative abundances in waters off Iquique (~20º S) during late-spring 2015.**

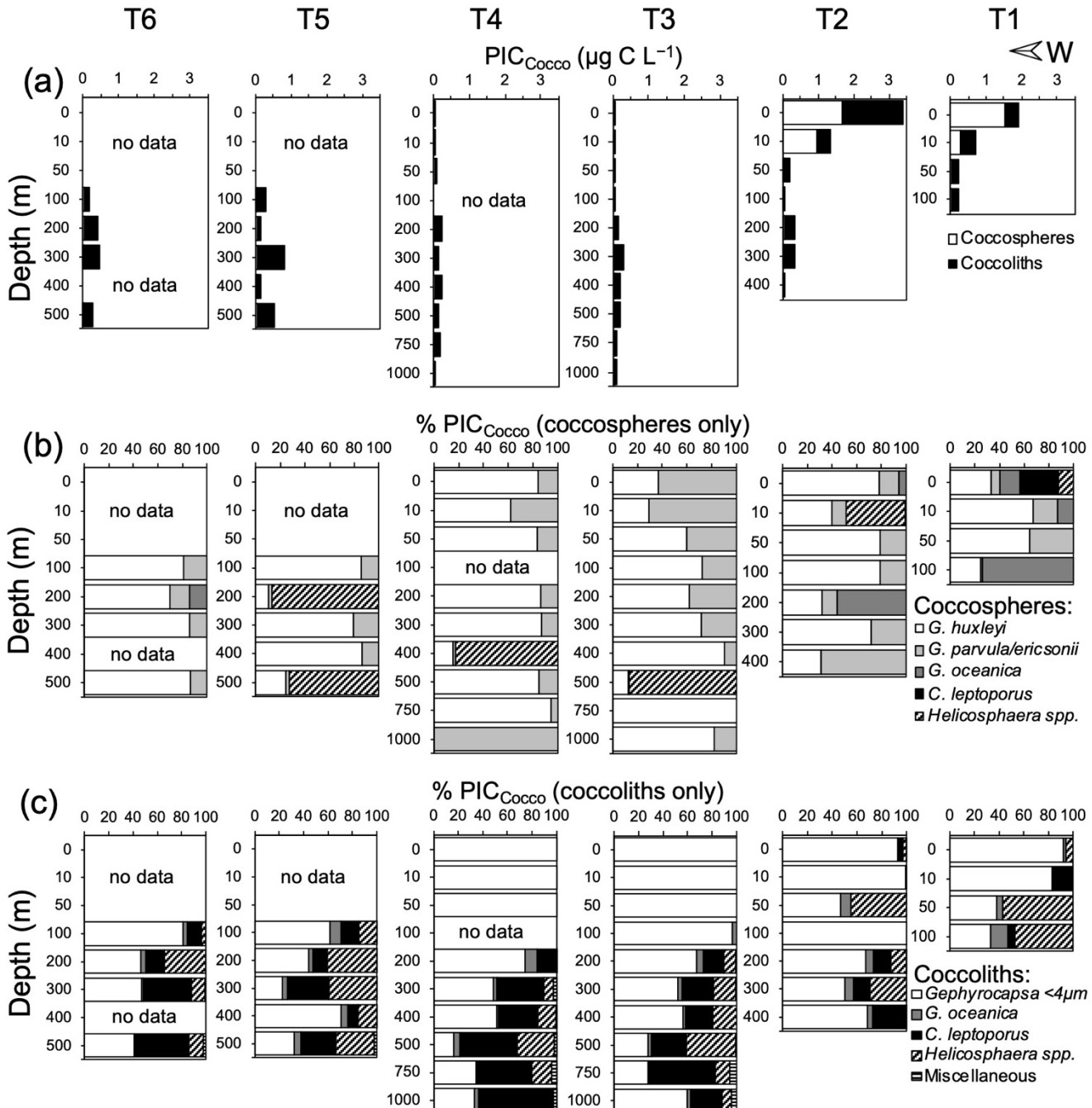

**Figure 6: Estimated PIC masses from coccospheres and detached coccoliths recorded in waters off Iquique (~ 20º S) during late-spring 2015. (a) Contribution of coccospheres and detached coccoliths to the total PIC$_{Cocco}$ pool. (b) Taxonomic breakdown of the relative contribution of coccospheres to PIC$_{Cocco}$ quotas, expressed as percentages of the total PIC$_{Cocco}$ pool. (c) Taxonomic breakdown of the relative contribution of detached coccoliths to PIC$_{Cocco}$ quotas, expressed as percentages of the total PIC$_{Cocco}$ pool. The PIC$_{Cocco}$ pool estimate is derived from abundances obtained through scanning electron microscopy.**

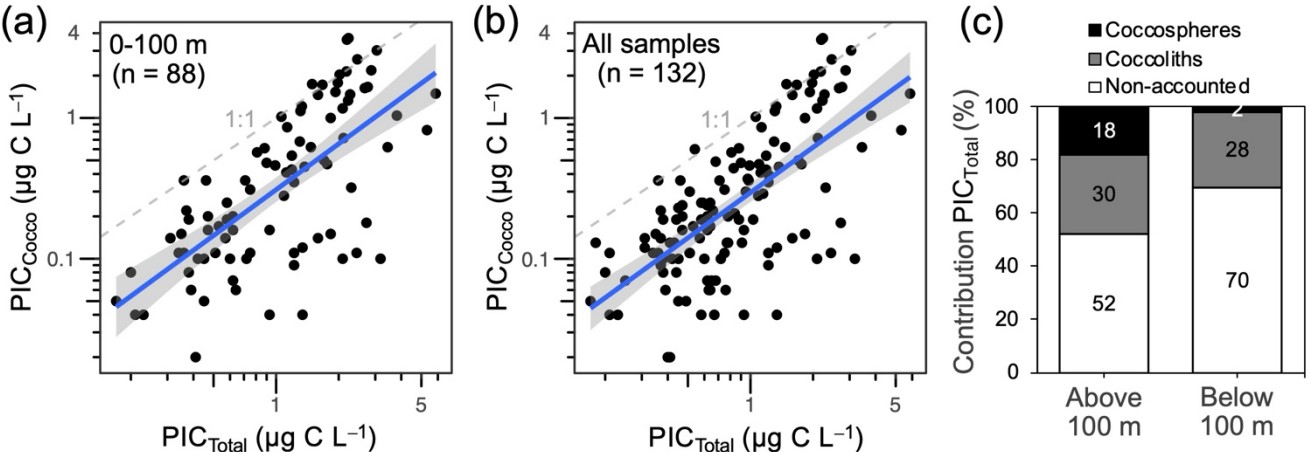

Figure 7: Linear dependence between the $PIC_{Total}$ and $PIC_{Cocco}$ (a-b) across 100 m depth and all samples, along with the bulk proportion of $PIC_{Total}$ accounted for by $PIC_{Cocco}$ (c) including data from above and below 100 m depth. $PIC_{Total}$ and $PIC_{Cocco}$ show a significant correlation (a): $Y = 1.082(\pm 0.126)X - 1.173(\pm 0.100)$; $R^2_{(adjusted)} = 0.45$; $p$-value$_{(slope, constant)} < 0.05$; n = 88 samples, as well as across all samples (b): $Y = 1.068(\pm 0.102)X - 1.220(\pm 0.077)$; $R^2_{(adjusted)} = 0.46$; $p$-value$_{(slope, constant)} < 0.05$; n = 132 samples. Solid blue line in (a-b) is the least squares fit, and grey areas depict 95 % confidence intervals. Dashed line in (a-b) represent the 1:1 relationship. Note the log-log axes enable assessment of $PIC_{Cocco}$ estimation performance over ~2 orders of magnitude of $PIC_{Total}$ concentration. PIC = Particulate Inorganic Carbon.

### 3.6 Coccoliths and PIC variation across environmental conditions

Coccospheres, detached coccoliths, PIC, and the estimated $PIC_{Cocco}$ stocks were not statistically different ($p < 0.05$) between the 2015 and 2018 cruises, although mid-summer 2018 exhibited higher average and maximal values compared to late-spring 2015 (Fig. 8). The relatively higher SST (> 21 ºC) recorded westward of the stronger upwelling band closer to the coast (Fig. 2a), may be associated with these enhanced $PIC_{Cocco}$ pools. A shallow $Z_{eu}$, averaging 36 m and ranging from 26 to 50 m across both cruises ($n = 22$ stations; Fig. 2c-e, Fig. 3k-o), was observed, below which there was a sharp decline in coccolithophores and derived PIC (Fig. 8a-c). Overall, the average $PIC_{Total}$ and $PIC_{Cocco}$ pools below the $Z_{eu}$ decreased by 50 % and 83 %, respectively (Fig. 8c-d).

On average, a notable unimodal thermal response peaking at ~18 ºC was observed for both $PIC_{Cocco}$ and $PIC_{Total}$ for depths above $Z_{eu}$ (Fig. S15a, S16a). The oxycline and pHcline (7.7-7.9), located near the base of $Z_{eu}$ (Fig. 2c), were associated with decreased PIC levels (Fig. S15c, g, S16c, g). Above a pH of ~7.7 ($\Omega_{calcite}$ ~ 2.5; these two variables were highly correlated), $PIC_{Cocco}$ displayed a clear unimodal relationship, peaking at pH ~7.9 ($\Omega_{calcite}$ 2.5-3.5; Fig. S15g, i), while $PIC_{Total}$ showed a monotonic increase (Fig. S16g, i). $p$CO$_2$ levels below 700 µatm corresponded to an average increase in PIC values (Fig. S15h, S16h). Low-to-moderate Chl-$a$ levels (< 4 mg m$^{-3}$) were associated with enhanced $PIC_{Cocco}$ (> 2 µg C L$^{-1}$) and $PIC_{Total}$ (> 3 µg C L$^{-1}$), although only $PIC_{Cocco}$ returned to background levels at higher Chl-$a$ concentrations (Fig. S15d, S16d).

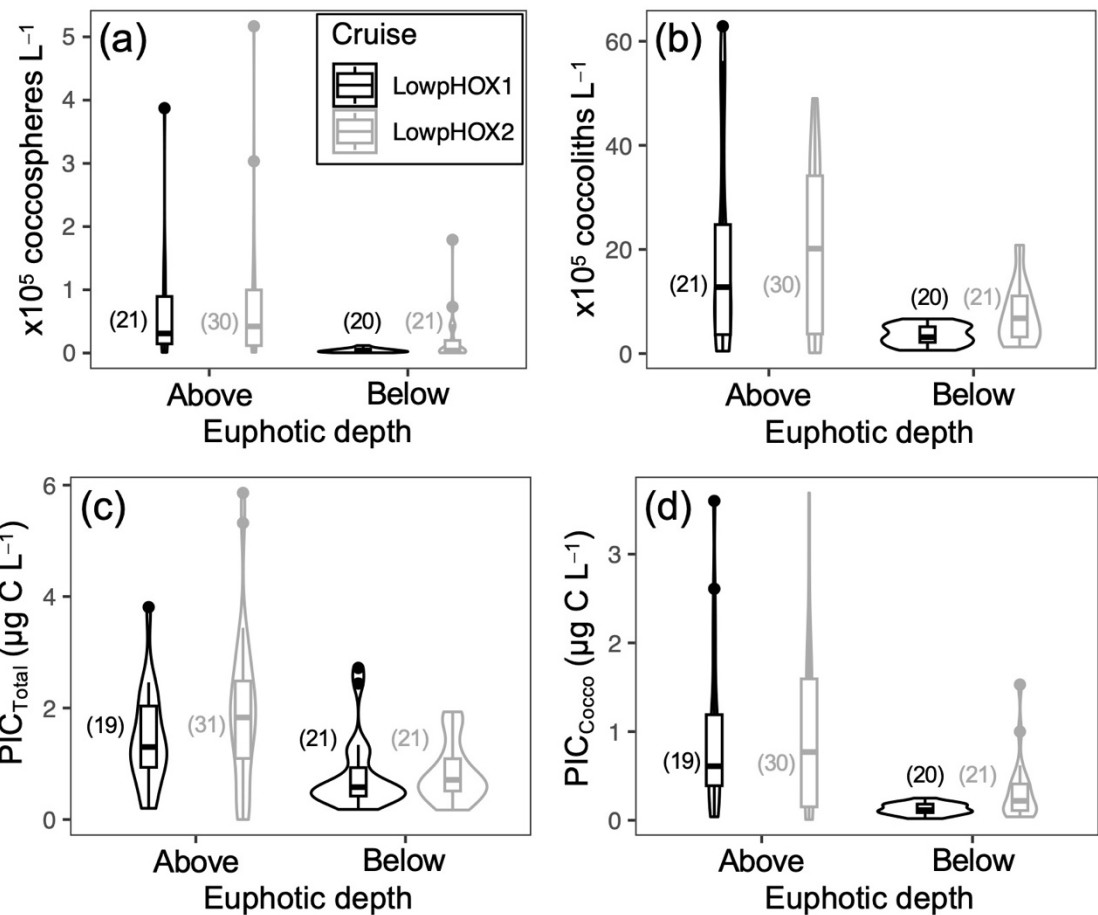

**Figure 8: Violin plots comparing the variation in coccospheres (a), detached coccoliths (b), PIC_Total (c), and PIC_Cocco (d) above and below the euphotic depth during late-spring 2015 and mid-summer 2018. Only samples taken in the upper 100 m depth are included. The number of samples tested per category is given in parentheses.**

## 4 Discussion

### 4.1 Coccolithophore species diversity and dominance off the Southeast Pacific margin

Coccospheres and detached coccoliths of *G. huxleyi* were dominant in 2018 at 20° S, as well as south of 20° S in both 2015 and 2018. This pattern is shared with other eastern boundary current systems, where *G. huxleyi* dominates numerically in waters off the Northeast Pacific (Ziveri et al., 1995; Venrick, 2012) and the Southeast Atlantic margins (Siegel et al., 2007; Henderiks et al., 2012). However, in addition to *G. huxleyi*, *G. parvula/ericsonii* were co-dominant members of the coccolithophore communities at ~20º S in 2015. The co-dominance of these three species was previously reported in winter samples from 2013 in the same zone (von Dassow et al., 2018; Díaz-Rosas et al., 2021). Likewise, "small *Reticulofenestra* complex" (presumably mostly *G. parvula*) and "small *Gephyrocapsa* complex" have been previously reported as important in

tropical and equatorial waters further offshore and to the north (Hagino and Okada, 2004), so these species are frequently important in these and neighbouring waters. To the south, co-dominance of *G. huxleyi* and *G. muellerae* has been reported previously (e.g., Díaz-Rosas et al., 2021). Thus, while coccolithophore richness remains low in the Eastern South Pacific margin, as indicated by the rapid saturation of rarefaction curves to total diversity across upper and deeper surface layers, this region exhibits a higher diversity within the genus *Gephyrocapsa* compared to other Eastern Boundary Currents (Henderiks et al., 2012; Venrick, 2012). Standing stocks of larger taxa contributed minimally to the total community in the $Z_{eu}$ (~50 m depth), becoming more prominent below this depth. It should be noted that each coccosphere or detached coccolith of *C. leptoporus* is equivalent to approximately 80 coccospheres or 45 coccoliths of *G. huxleyi* in terms of PIC. This suggests that the heavier coccoliths of these larger species may sink more efficiently (e.g., Menschel et al., 2016; Guerreiro et al., 2021).

## 4.2 Potential uncertainties in PIC measurements and contributions of coccolithophores to PIC

How much coccolithophores contribute to PIC remains an open question, as contributions from calcifying zooplankton (e.g., foraminifera, pteropods; Ziveri et al., 2023), lithogenic sources (Daniels et al., 2012), and processes like fragmentation and dissolution in the water column (Barrett et al., 2014; Subhas et al., 2022) complicate the relationship between PIC and coccolithophores. Additionally, PIC becomes increasingly difficult to measure as it decreases. A classic method is to measure total particulate carbon before and after acidification to remove PIC, but this method is relatively insensitive and problematic when the PIC:POC ratio is low (Balch and Kilpatrick, 1996). Measuring PIC by the acid soluble $Ca^{2+}$ extracted from particulate matter is much more sensitive, yet it also must be corrected for $Ca^{2+}$ from seawater which is retained on the filter by organic matter even after gentle rinsing (the Na+ correction; see Matson et al., 2019 and section 2.2 above). These complexities challenge remote sensing algorithms (Balch and Mitchell, 2023) or the use of PIC as a paleoproxy indicator (Beaufort et al., 2011). This emphasizes the value of microscopy counts, which have been found to be effective in quantifying PIC due to coccolithophores (D'Amario et al., 2018; Guerreiro et al., 2021; Ziveri et al., 2023; this study).

In the northern Chilean coast, potential lithogenic PIC sources should be negligible. The exceptionally arid Atacama Desert means fluvial inputs are negligible (Thiel et al., 2007) and the deep and steep topography of the Atacama Trench acts as a major depocenter effectively trapping sediments and limiting their resuspension into the upper water column (Xu et al., 2021). As sources of lithogenic PIC to these waters are limited, we consider most PIC is likely biogenic.

In addition to PIC production by other planktonic organisms, which would not be detected by the microscopy protocols used here, estimates of $PIC_{Cocco}$ by microscopy also include several sources of error which can cause underestimates of $PIC_{Cocco}$ relative to $PIC_{Total}$ due to difficulties in detection of smaller coccoliths, collapsed coccospheres, and fragmented coccoliths. Other sources of error relate to taxonomic and phenotypic variability in conversion factors, PIC quotas per coccolith, and estimates of the number of coccoliths per coccosphere. Young and Ziveri (2000) suggested that these considerations might result in errors of up to 50 % in the estimation of $PIC_{Cocco}$ using microscopic methods. However, we found that improved taxonomic resolution led only minor changes in $PIC_{Cocco}$ estimates in most T1-T6 samples due to the strong dominance of *G. huxleyi* and its close relatives in these waters. Despite these important limitations, $PIC_{Cocco}$ accounted for nearly half of direct

$PIC_{Total}$ measurements and $PIC_{Cocco}$ values calculated from coccosphere and coccolith abundances were linearly correlated with chemical measurements of $PIC_{Total}$.

The highest monthly and weekly satellite-PIC estimations, which sometimes exceeded 10 µg C $L^{-1}$ (average ~ 2.4-3.6 µg C $L^{-1}$; Fig. 4), exceeded maximum *in situ* PIC measurements by a factor of ~2. Similarly, satellite-PIC retrievals were reported to overestimate the $PIC_{Cocco}$ pools by a factor of 1 to 5 across the New Zealand and Drake Passage sectors of the Southern Ocean (Saavedra-Pellitero et al., 2023). Satellite data has high spatial resolution which can detect patchy distributions not effectively sampled *in situ*, but is affected by frequent cloud cover (as in this region, which forced us to depend on weekly and monthly averages rather than daily measures) and also misses events of higher PIC that only occur in the lower euphotic zone. While optical PIC proxies may need to be geographically adjusted (Balch and Mitchell, 2023), it is difficult to directly compare remote and *in situ* measurements because of these differences in spatial and temporal resolution. Overall, both *in situ* and satellite-derived PIC supported a local hotspot of PIC near 20° S, and *in situ* measures are consistent with the lower PIC in the region suggested by previous satellite-analyses (discussed below).

In summary, our results emphasize the importance of coccolithophores as contributors to total PIC pools in this OMZ. The results also indicated the importance of combining the two *in situ* methods to enhance confidence in PIC pools estimations within the OMZ system.

**4.3 Surface variation in coccolithophores and PIC pools**

Events of increased *G. huxleyi* coccospheres (> 3.0 x$10^5$ $L^{-1}$), along with detached coccoliths (> 30 x$10^5$ $L^{-1}$), were observed in the upper waters during both late-spring 2015 and mid-summer 2018. Consequently, the $PIC_{Total}$ and $PIC_{Cocco}$ pools rose to 3-5 µg C $L^{-1}$, representing approximately a threefold increase compared to background levels. Coccosphere and detached coccolith abundances were particularly high at 20º S, with inshore station T2 in 2015 showing 12 x$10^6$ coccoliths $L^{-1}$ at 2 m depth and offshore station T5 in 2018 showing 10 and 12 x$10^6$ coccoliths $L^{-1}$ at 25 and 30 m depth, respectively. On average, higher values were recorded during mid-summer 2018 compared to late-spring 2015, suggesting potential seasonality in PIC levels. This observation aligns qualitatively with satellite-PIC retrievals showing PIC concentrations in February 2018 extending further offshore and southward from the inshore 20º S sampled site (see Fig. 4) and is consistent with global satellite-PIC estimations showing higher inventories off the coasts of Chile from austral spring (October-November) peaking in summer (January-February), a pattern similar to what is indicated by satellite for the Benguela Current System in front of Namibia (Hopkins et al., 2019).

Coccospheres accounted for an average of 37 % of $PIC_{Cocco}$ in the surface mixed layer (Fig. 7c), and the ratios of detached coccoliths to coccospheres were mostly below 40 (median: 31), with the highest coccosphere abundances and $PIC_{Cocco}$ values associated to relatively low detached coccospheres (Fig. S11c-d), in contrast to the greater excess of detached-coccolith over coccosphere ratios reported for massive blooms (> 250; e.g., Balch et al., 1991; Holligan et al., 1993b). Lower ratios might

indicate populations in active physiological states rather than cells entering decreased growth as maximum bloom densities are reached and cell growth slows or ceases (Balch et al., 1991; Holligan et al., 1993b; Lessard et al., 2005).

Our dataset revealed relatively higher coccolithophore-produced PIC at the margin compared to the very low values in the Southeast Pacific central and eastern gyre (i.e., compared to the Atlantic Ocean; see Balch et al., 2018). The highest $PIC_{Cocco}$ standing stocks observed between 60 and 200 m depth during spring/summer in the remote Southeast Pacific (Beaufort et al., 2008; Oliver et al., 2023) likely reflect site-dependent biophysical constraints. Overall, these findings highlight the dynamic nature of coccolithophore populations and their PIC contributions, emphasizing the need for ongoing monitoring to understand

their ecological roles and responses to environmental changes. In that sense, combining remote sensing and *in situ* sampling is particularly helpful.

Coccolith PIC stocks varied significantly with SST, with higher PIC levels ($PIC_{Cocco}$ and $PIC_{Total}$) around 18 °C (Fig. S15a), consistent with a preference for nutrient-rich coastal upwelling and nutrient-poor offshore waters. Similar trends were observed off the Santa Barbara coast, where higher temperatures and stratification increased coccolithophore abundance and PIC pools

(Matson et al., 2019). High coccolithophore abundances also occur in upwelling regions like the equatorial Pacific (Balch and Kilpatrick, 1996) and Arabian Sea (Balch et al., 2000), and in semi-oligotrophic areas such as the northern fjords, North Sea and Atlantic Ocean, under high-light conditions (Tyrrell and Merico, 2004; Zondervan, 2007). Coccolithophore blooms and PIC production are boosted after spring diatom blooms in upwelling zones, aligning with Margalef's (1978) successional model, which links their blooms to preferences for lower nutrients and turbulence than diatoms. Strong stratification with

moderate nutrients favours higher $PIC_{Cocco}$ and $PIC_{Total}$, and indicating an inverse relationship between coccolithophores and diatoms under varying conditions in this region (Menschel et al., 2016; Díaz-Rosas et al., 2021), similar to what has been proposed the North Atlantic and other regions (Tyrrell and Merico, 2004).

Consistent with conditions that favored coccolithophores, the maximum coccolithophore abundances recorded in both years in this study were about twice maximum values previously reported in the entire Chilean sector of the Southeast Pacific

(Beaufort et al., 2008; Menschel et al., 2016; Díaz-Rosas et al., 2021) and the previously reported maximum for the entire Humboldt Current System (Hagino and Okada, 2006). Nevertheless, maximum standing coccolithophore stocks were still about an order of magnitude lower than the typical bloom abundances reported in other regions (Tyrrell and Merico, 2004), including the Southwest Atlantic (Poulton et al., 2013), the North Atlantic (Holligan et al., 1993a), and the Gulf of Maine (Balch et al., 1991), and so coccolithophore concentrations in the Humboldt Current still appear to be lower than reported for

other productive regions (Díaz-Rosas et al., 2021).

Two factors might limit coccolithophore populations from forming massive blooms like those seen in the Atlantic. First, these high productivity waters are typically dominated by diatoms, which may outcompete coccolithophores (Menschel et al., 2016). In the California Current, the most similar Eastern Boundary Upwelling system, a *G. huxleyi* bloom was reported to reach abundances like those seen in Atlantic and Bering Sea blooms (Matson et al., 2019), though such events appear less common

than in other regions.

Second, the low pH typical of upwelled water in this region inhibits the growth of many coccolithophore strains in culture (Sciandra et al., 2003; Bach et al., 2011; Meyer and Riebesell, 2015; Müller et al., 2015), even those isolated from the same region, which would presumably be best adapted to tolerate such conditions (von Dassow et al., 2018). Environmental factors such as pHcline and oxycline may limit coccolithophore growth by impacting calcification, respiration and perhaps also photorespiration even in non-upwelling conditions. Stocks of coccolithophores and coccoliths and PIC pools in the upper 100 m were indeed limited by both low pH and low dissolved $O_2$ (Fig. 9).

This study presents the first *in situ* $PIC_{Total}$ measurements from the OMZ waters off the Southeast Pacific margin, corresponding to late-spring 2015 and mid-summer of 2018. Similar to coccolithophore stocks, $PIC_{Total}$ values tended to be lower to values documented so far for most of the rest of the ocean (Table 1, Fig. 10b). The observed PIC levels were about one-order-lower those reported across the Great Calcite Belt during summer (Balch et al., 2018) or springtime waters off the Bay of Biscay (Daniels et al., 2012). Recently, enhanced coccolithophore and PIC pools were observed in February to the west in the central Pacific, with a maximum $PIC_{Cocco}$ of 15.5 µg C $L^{-1}$, representing about two-fold greater PIC pools to those estimated in this study (Oliver et al., 2023; refer to Table 1). As an upper threshold of $PIC_{Cocco}$ pools, the most densely recorded pool of ~400 $x10^6$ coccoliths $L^{-1}$ in the Gulf of Maine (Balch et al., 1991) and North Atlantic (Holligan et al., 1993a) correspond to approximately 120 µg C $L^{-1}$ via allometric mass conversion. There is very limited comparable PIC data from other Eastern Boundary upwelling systems. However, $PIC_{Total}$ values as high as 67.3 µg C $L^{-1}$ was reported in a California Current *G. huxleyi* bloom (Matson et al., 2019; data not included in Fig. 10 because POC values not available). With the prominent exception of the Western Arctic Chukchi Sea, perhaps reflecting that coccolithophores still have limited penetration into Arctic waters (Winter et al., 2014), mixed-layer $PIC_{Total}$ from the Southeast Pacific OMZ overlapped with the lower ranges of most other regions (Table 1, Fig. 10b).

In contrast to PIC, POC values strongly overlapped with those from other productive regions (Fig. 10c). As a result, PIC:POC ratios in the surface mixed layer were significantly lower in the OMZ region compared to all regions with available data, except for the Arctic (Fig. 10d).

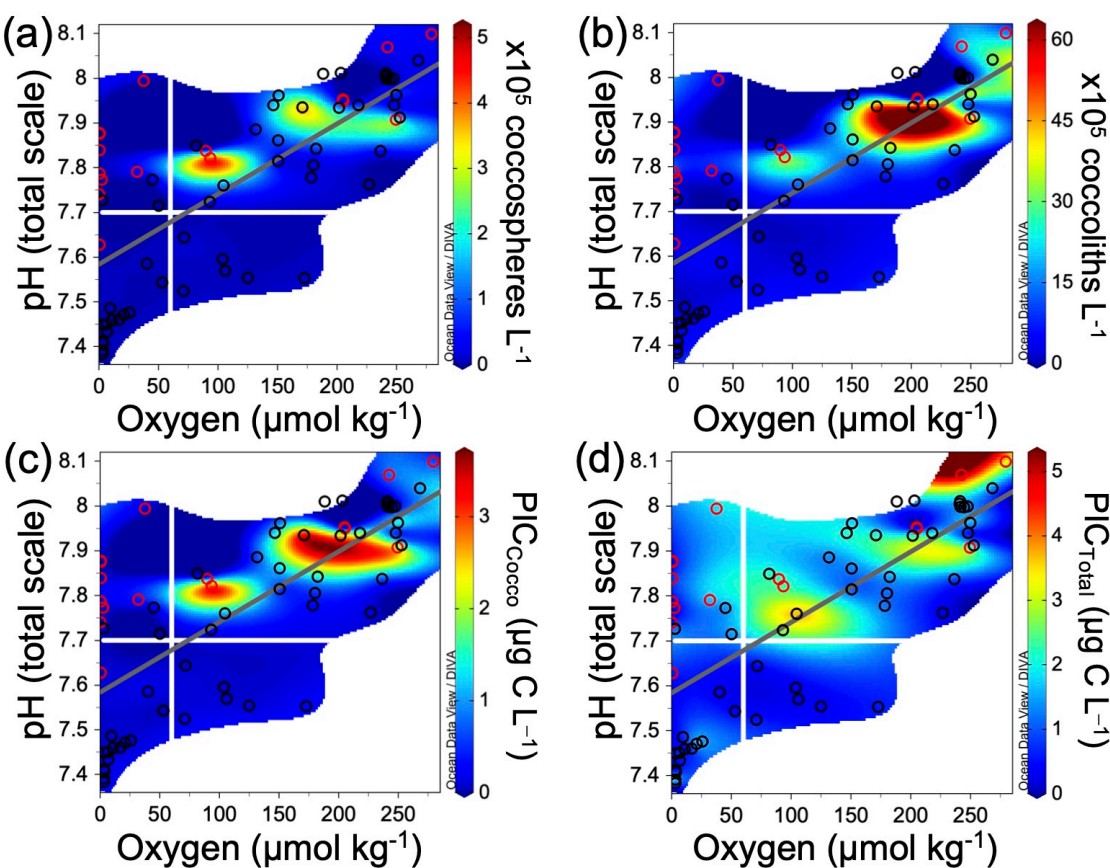

**Figure 9. Variation in coccosphere (a) and detached-coccolith standing stocks (b), as well as $PIC_{Cocco}$ (c) and $PIC_{Total}$ pools (d), across oxygen and pH levels recorded within upper 100 m during late-spring 2015 (open black dots) and mid-summer 2018 (open red dots). Horizontal and vertical white lines indicate the expected pH values for 2100 and the oxygen contour for the Eastern Equatorial Pacific Ocean proposed by Stramma et al. (2008), respectively. The grey line depicts the least-squares model fit curve.**

**Table 1: Comparison of Particulate Inorganic Carbon standing stocks reported in surface layers (0-100 m depth) of different oceanic areas or shelf/coastal margins. Unless otherwise indicated, the PIC was measured chemically using mass spectrophotometry. AMT – Atlantic Meridional Transect; SP – Spring; AU – Autumn; WI – Winter; SU – Summer.**

| Sampling domain (Cruise) | Season/Year | PIC ($\mu$g C L$^{-1}$) | # Stations | Source |
|---|---|---|---|---|
| North and South Atlantic (AMT14) | AU/2004 | 0.09 – 99.8[a] | 27 | 1 |
| Scotia Sea/Drake Passage (JR163) | SP/2006 | < 0.01 – 15.5[b] | 31 | 2 |
| North and South Atlantic (AMT17-22) | SP, AU/2005-12 | 0.05 – 49.8 | 481 | |
| South Atlantic/Patagonia (COPAS08) | SP/2008 | 0.02 – 51.7 | 33 | |
| Southern Ocean/Atlantic (G. Belt-I) | SU/2011 | 0.03 – 116.8 | 31 | 3 |
| Western Arctic (ICESCAPE'11) | SU/2011 | 0.004 – 8.9 | 16 | |
| Southern Ocean/Indian (G. Belt-II) | SU/2012 | 0.01 – 9.71 | 32 | |
| Northeast Atlantic margin | WI, SP/2009-10 | 0.84 – 140.5 | no data[e] | 4 |
| Northeast Atlantic margin (D381) | AU/2012 | 1.35 – 16.8 | 21 | 5 |
| Santa Barbara channel | SP/2015 | 3.6 – 67.3[c] | no data[e] | 6 |
| Southern Ocean/Pacific (RR2004) | SU/2021 | < 0.01 – 15.5[d] | no data[e] | 7 |
| Southeast Pacific margin (LowpHOx 1) | SP/2015 | 0.18 – 3.8 | 12 | This study |
| Southeast Pacific margin (LowpHOx 2) | SU/2018 | 0.08 – 5.9 | 10 | This study |

[a] A > 300 value identified as an outlier was not included. [b] Extracted manually from Figure 4c. [c] Extracted manually from Figure 3b. [d] Estimated from underway "acid-labile backscattering" measurements calibrated with PIC measurements obtained with the same methodology outlinedf above. [e] Restricted to the uppermost surface waters. 1 Poulton et al. (2006); 2 Holligan et al. (2010); 3 Balch et al. (2018); 4 Daniels et al. (2012); 5 Painter et al. (2016); 6 Matson et al. (2019); 7 Oliver et al. (2023).

## 4.4 Subsurface PIC and PIC:POC ratios

Below the $Z_{eu}$, coccolithophore PIC was predominantly composed of detached coccoliths, which accounted for almost all of the total PIC$_{Cocco}$ (Fig. 7c), similar to records during blooms in the North Atlantic (Fernández et al., 1993; van der Wal et al., 1995) and the North Pacific (Ziveri et al., 2023). While detached coccoliths of *G. huxleyi* and other small *Gephyrocapsa* remain important, larger and rare species with the heaviest coccoliths, such as *C. leptoporus* and *Helicosphaera* spp., became relatively more important to PIC quotas below the $Z_{eu}$, as seen previously in other regions (e.g., Ziveri et al., 2007; Guerreiro et al., 2021) and even in neighbouring waters (Menschel et al., 2016). This detrital material is expected to ballast POC to the deep ocean (e.g., Klaas and Archer, 2002). Additionally, a greater decrease in PIC$_{Cocco}$ than PIC$_{Total}$ below the $Z_{eu}$ (Fig. 8c-d) is consistent with aggregation and fragmentation of coccoliths contributing to the non-accounted PIC$_{Total}$, which could sink even more POC (Briggs et al., 2020).

Subsurface PIC values tended to overlap the lower ranges for other ocean regions for which data is available (with the prominent exception of the Western Arctic) (Fig. 10e). Because of the broad ranges in subsurface PIC values within each ocean region, subsurface PIC was not significantly different between the OMZ and other regions except for the comparison with the

Southern Ocean and Indian Ocean. Nevertheless, only the Western Artic showed subsurface PIC concentrations that were similarly low to those in the OMZ.

Within the OMZ-core, significantly elevated PIC:POC ratios compared to the surface layer (Fig. 10j; $p$-value $< 0.05$) suggest that sinking particles are enriched in denser inorganic carbon (PIC), enhancing their 'ballast effect' and causing them to sink faster (Lee et al., 2009; Iversen and Ploug, 2010). This rapid descent decreases the decomposition time in the upper water

column, facilitating the export of more POC to the OMZ. Despite low pH, $\Omega_{calcite}$ remained above 1 in the OMZ mesopelagic, and the higher PIC:POC ratio in the OMZ layer indicates that dissolution processes (such as those in microenvironment of lower $\Omega_{calcite}$) are not fully offsetting the export of PIC from the surface layer. Both PIC and POC decrease significantly within the OMZ core compared to above it ($p < 0.05$; Fig. 10h-i), indicating that while the 'ballast effect' may promote faster sinking, the overall POC pool remains lower, with anaerobic degradation processes continue to consume POC as $O_2$ levels vanish

(Vargas et al., 2021).

PIC:POC ratios in the subsurface OMZ were significantly lower than those observed in subsurface waters of most other open-ocean and coastal sites (Fig. 10g), consistent with a diminished role of PIC in POC export within OMZ systems. Interestingly, while PIC:POC did increase from the surface to the subsurface in OMZ waters (Fig. 10j), this increase was lower than in other regions with comparable data (Table 2). Despite evidence suggesting a decreased role of PIC as a ballast in OMZs, POC

transfer from the euphotic zone to deeper waters has been shown to be efficient in these systems (Cavan et al., 2017; Engel et al., 2017; Weber and Bianchi, 2020). This highlights that the limited role of PIC ballast is just one of several key functional differences influencing POC fluxes between OMZ and non-OMZ regions.

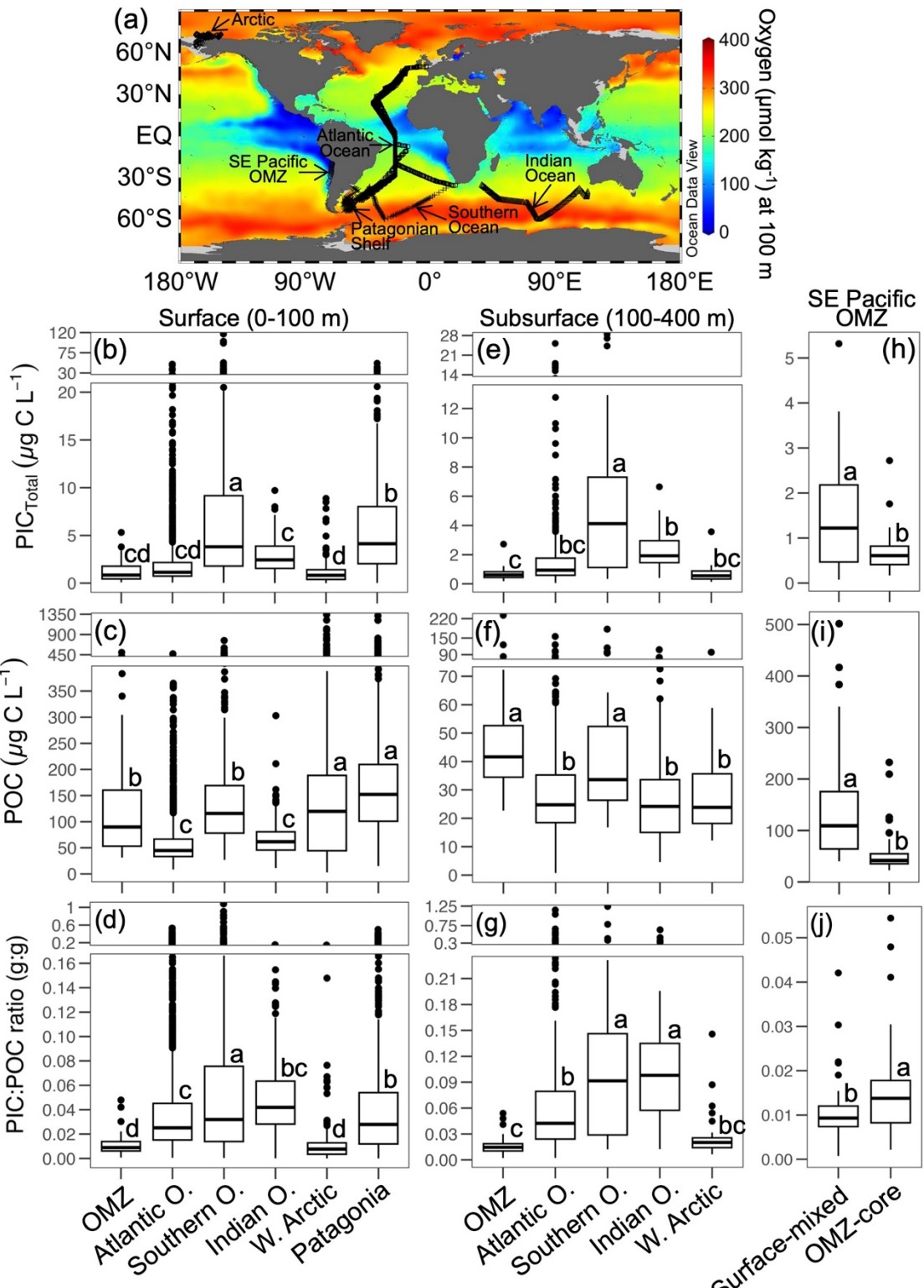

**Figure 10: (a) Global map showing the annual oxygen content at 100 m depth and the sampling locations for the well-mixed surface (0-100 m; panels b, c. d) and the stable subsurface (100-400 m; panels e, f, g), along with the surface-mixed and OMZ-core layers (panels h, i, j). The PIC$_{Total}$, POC, and PIC:POC ratio values during late-spring 2015 and mid-summer 2018 in the SE Pacific OMZ (this study), as well as in other open ocean and coastal margin regions, are shown (data from Balch et al., 2018). The Atlantic Ocean dataset includes samples from six cruises (AMT17-22). One-way ANOVA results indicate significant differences ($p < 0.05$) in PIC$_{Total}$, POC, and PIC:POC ratios among the OMZ, Atlantic Ocean, Southern Ocean, Indian Ocean, Western Arctic, and Patagonian Shelf, as well as between OMZ layer sample groups based on Tukey post-hoc comparisons, represented by lowercase letters above each boxplot. Only data points with both PIC and POC available were included in the analysis. The map was generated using Ocean Data View (Schlitzer, 2024), with annual oxygen climatology from the World Ocean Atlas 2018 (Boyer et al., 2018; García et al., 2018).**

**Table 2: Comparison of the fractional increase in PIC:POC ratios between the surface (0-100 m) and subsurface (0-400 m) layers in the SE Pacific OMZ (this study) and other open-ocean and coastal regions (data from Balch et al., 2018). Only data points with both surface and subsurface PIC:POC ratios were included in the analysis.**

| Zone | Mean Surface | Mean Subsurface | Fraction (%) |
|---|---|---|---|
| SE Pacific OMZ | 0.011 | 0.017 | $68 \pm 88$ |
| Atlantic Ocean | 0.037 | 0.066 | $116 \pm 186$ |
| Southern Ocean | 0.102 | 0.161 | $107 \pm 235$ |
| Indian Ocean | 0.046 | 0.094 | $145 \pm 234$ |
| Western Arctic | 0.014 | 0.025 | $133 \pm 114$ |

## 5 Conclusions

The examination of the coccolithophores, along with the PIC standing stocks between late-spring 2015 and mid-summer 2018 waters off the Southeast Pacific margin, yielded nine conclusions:

1. Coccolithophores-PIC pools are highest within the first 30 m depth, with slightly higher PIC pools observed in summer 2018 compared to spring 2015.

2. The cosmopolitan species *G. huxleyi* is the primary contributor to detached coccoliths and biogenic calcium carbonate.

3. Nevertheless, near 20° S *G. huxleyi* was found to be co-dominant with *G. parvula/ericsonii*, in late spring of 2015, as occurred in early winter of 2013, so this may be a repeated feature.

4. Coastal waters near 20° S were a hotspot for high coccolithophore stocks and PIC in different years and seasons.

5. Coccolithophores are responsible for roughly half of total suspended PIC in the surface layer of the OMZ (0-100 m depth), an estimate considered conservative due to challenges associated with estimating PIC by microscopy.

6. Both the PIC$_{Total}$ and PIC$_{Cocco}$ pools decreased by 50 % and 83 % below the euphotic depth, coinciding with the oxycline and pHcline.

7. Horizontal and vertical variation in PIC$_{Total}$ and PIC$_{Cocco}$ pools in the OMZ region strongly depends on temperature, along with oxygen and pH levels.

8. Mixed layer coccolithophore and PIC pools tended to be lower in the OMZ region compared to other open ocean and coastal margin regions of similar productivity (POC), with the exception of the Western Arctic. As a result, mixed layer PIC:POC ratios were significantly lower in the OMZ region compared to all but the Western Arctic.

9. Subsurface PIC and PIC:POC ratios were also lower than other regions for which data were available, with the exception of the Western Arctic. PIC:POC increases from the surface mixed layer to the OMZ core, but the increase is lower compared to the rest of the ocean. These considerations suggest that, in OMZ regions, PIC plays a diminished role as ballast to drive POC fluxes to the deep.

**Data availability.**

All data generated in this study are available upon request from the corresponding author. The scanning electron micrograph image datasets are accessible at https://doi.org/10.5281/zenodo.14048319 (Díaz-Rosas et al., 2024a). The cross-polarized light microscopy images shown in the Supplementary Material can be found at https://doi.org/10.5281/zenodo.14708540 (Díaz-Rosas et al., 2024b). The coccosphere and detached coccolith count data, along with the associated PIC and POC measurements from 2015 and 2018, are available at https://doi.pangaea.de/10.1594/PANGAEA.975783 (Díaz-Rosas et al., 2024c) and https://doi.pangaea.de/10.1594/PANGAEA.975784 (Díaz-Rosas et al., 2024d), respectively.

**Sample availability.**

**Supplement.**

**Author contributions.**

FDR (conceptualization, data curation, formal analysis, investigation, methodology, visualization, writing – original draft preparation, writing – review and editing) provided key proof-of-concept ideas, led the study, conducted polarized light and SEM microscopic analyses and taxonomic characterization of coccospheres and detached coccoliths, examined the relationships between PIC measurements, coccolithophore pools, and environmental/biological variables, compared PIC and PIC:POC values with existing repositories, and drafted the initial manuscript. PvD (conceptualization, funding acquisition, validation, visualization, writing – original draft preparation, writing – review and editing) defined the research goals, led the study, conducted sampling during the 2015 cruise, planned the 2018 sampling, and provided continuous insights into results interpretation and manuscript structure. CV (funding acquisition, validation, writing – review and editing) guided the interpretation of results and contributed to characterizing the physical and chemical environments. All co-authors provided critical feedback and contributed to the final editing of the manuscript.

**Competing interest.**

The contact author has declared that none of the authors has any competing interests.

**Acknowledgements.**

We thank Dr. William Balch, Dr. Catherine Mitchell, and Dr. David Drapeau for reading and providing valuable discussion on the manuscript.

**Financial support.**

This study was supported by the National Agency for Research and Development (ANID) of Chile through grants AIM23-
570 0003 and ICN12_019N for the Millennium Institute of Oceanography (IMO), as well as FONDECYT grant 1181614. Scanning electron microscopy analysis was performed at the Centro de Investigación en Nanotecnología y Materiales Avanzados (CIEN) of the Pontificia Universidad Católica de Chile using an SEM instrument purchased with FONDEQUIP grant EQM150101.

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
