# Peer review of "Particulate inorganic carbon pools by coccolithophores in low oxygen/low pH waters off the Southeast Pacific margin"

_EGUsphere, 2024_

## Author Comment (AC1)

In this study the authors investigate the impact of upwelled low O2/low pH waters on coccolithophore PIC quotas, coccolithophore PIC contribution to the total PIC pool and ultimately what impact this has on the flux of carbon to depth via the PIC:POC ratio in the oxygen minimum zone off central Chile. The general conclusion that this OMZ exhibits comparable PIC concentrations to selected open ocean areas but a greatly reduced PIC:POC ratio in the OMZ core is a nice observation that emerges from the study but the paper needs attention to clarify methodological ambiguities and the occasional lack of precision in reporting results.

**We thank the Reviewer for the supportive comments and the very thoughtful and thorough suggestions and corrections that we respond to below.**

*Major comments*

Methods: Aspects of the methodologies used need to be clarified. In particular, sections 2.3 and 2.4 are muddled presenting mixed LM and SEM methodologies and unclear reasoning. It is particularly unclear if bulk counting or species/genus level counting was followed consistently and the low number of SEM images examined (with a low magnification) may lead to underestimation of both coccosphere and coccolith counts (see comment below regarding relationship in Figure S4b). Improved description of the methods used is required as this directly leads to ambiguity over how PIC_cocco was estimated (section 2.5) and presented (results). In particular the authors should improve description of the LM and SEM methods used, the apparent bias due to LM counting and the implications of the methodological bias on PIC quotas.

**Thank you for highlighting this important point, which is further elaborated in the minor comments below and has been addressed in the relevant sections. We propose improving the corresponding Methods section to make clear that we mostly relied on LM for coccolithophore and coccolith counts, but the more expensive SEM was used in two separate ways. First, in some samples LM counts were not possible, so were replaced with SEM counts, and, to make sure the two methods were comparable, we counted a subset of samples with both LM and SEM and the two were statistically indistinguishable.  Second, we used a higher resolution SEM analysis to evaluate in the taxonomic composition. This is important to understand which taxa are contributing to PIC pools. It allows us to get an idea of how estimations of coccolithophore-derived PIC might be improved with better taxonomic resolution.**

**We also specify more details of both methods as well as how they were used and compared.  The proposed new section 2.3 makes this more clear:**

**2.3 Standing stocks of coccospheres and detached coccoliths**

For enumeration of coccospheres and detached coccoliths, between 0.1 to 1.0 L of seawater (increasing with depth) were filtered onto 25 mm polycarbonate filters with a 0.8 $\mu$m pore size, left to dry at room temperature in Petri dishes,

and stored with desiccant until microscopy analyses. Total coccosphere counts were conducted on filter slide preparations with oil immersion, using cross-polarized light microscopy (Zeiss, Axioscope 5). The analysis of twenty fields of view at 400x magnification covered 5.1 mm$^2$ of the filter area, corresponding to a range of 1.9-16.3 mL of seawater analysed. For counts of total detached coccoliths, eleven fields of view per filter were screened (224 x 165 $\mu$m per frame) at 630x magnification (oil immersion objective), covering 0.41 mm$^2$ of the filter area, corresponding to total volumes of 0.2-1.3 mL of seawater analysed. An issue arose where the filters from inshore-offshore 2015 sampling (20º S; Stations T1-T6) exhibited excessive brightness under cross-polarized light microscopy (LM), for which counts were made through scanning electron microscopy (SEM) analysis (Quanta FEG 250) as described in Díaz-Rosas et al. (2021). For the quantification of coccosphere abundances (see equation in Díaz-Rosas et al., 2021), between 28-48 images taken at 800-1500x magnification were examined per filter, covering from 5 to 6 mm$^2$ of the filter area corresponding to a range of 2.1-18.4 mL of seawater analyzed. For total detached coccolith abundances, between 4-5 images were examined per filter, covering from 0.6 to 1.0 mm$^2$ of the filter area corresponding to a range of 0.2-2.8 mL of seawater analyzed. Layers of coccoliths detached from *G. huxleyi* (Fig. S3a-d) were added to the detached-coccolith counts. Collapsed coccospheres were included when they remained mostly intact, but when more disintegrated could not be accurately counted, especially as they were often less reflective than intact coccospheres and coccoliths (Fig. S3e-h). In a subset of samples, collapsed coccospheres were estimated to contribute < 21 % (min. = 0 %, average = 7.1 %) of the total number of coccospheres. As expected, the standard error of the means (among images) drops hyperbolically with the total number of coccospheres or coccoliths counted, whether with LM or SEM (Fig. S4), but remains higher in SEM due to the smaller size of SEM images. To check for differences between counts obtained through cross-polarized light microscopy and SEM examination, five samples with varying coccolithophore abundances were analyzed with SEM as outlined above, revealing the slopes were highly linear with R$^2$ greater than 0.9, and were not significantly different from 1, while the intercepts were close to 0 (Fig. S5), allowing for counts from the two methods to be combined.

**We also propose to make similar adjustments to the next section of Methods**

"**2.4 Diversity of coccospheres and detached coccoliths**

Identification of coccolithophores and detached coccoliths by light microscopy is sometimes limited, which might impact the estimation of coccolithophore-derived PIC (PIC$_{Cocco}$) to total PIC via the estimations of coccolith and coccosphere PIC pools. To understand this effect, taxonomic classification by SEM was performed on samples from the LowpHOX 1 cruise (2015), focusing on samples from T1 to T6 as well as selected samples from L1 (at 5 and 25 m), L2 (at 5 and 50 m) and L3 (at 5 m)..."

**Similar minor adjustments to the Results as well as figure legends can also help make these points clear.**

**We also propose to convert section 4.2 of the Discussion into a concise consideration of the types of uncertainties and how these are dealt with. This also makes clear that we focus the central interpretation on the manuscript on messages that are robust to these uncertainties. Despite the uncertainties, we can feel we can conclude that coccolithophores are important contributors to the surface and subsurface PIC pools in this OMZ area, and the conclusion that both layers of these waters show relatively low PIC and PIC:POC ratios.**

"**4.2 Potential uncertainties in PIC measurements**

How much coccolithophores contribute to PIC remains an open question, as contributions from calcifying zooplankton (e.g., foraminifera, pteropods; Ziveri et al., 2023), lithogenic sources (Daniels et al., 2012), and processes like fragmentation and dissolution in the water column (e.g., Barrett et al., 2014; Subhas et al., 2022) complicate the relationship between PIC and coccolithophores. Additionally, PIC becomes increasingly difficult to measure as it becomes low. A classic method is to measure total particulate carbon before and after acidification to remove PIC, but this method is relatively insensitive and problematic when the PIC:POC ratio is low. Measuring PIC by the acid soluble Ca extracted from particulate matter is much more sensitive, yet it also must be corrected for Ca from seawater which is retained on the filter by organic matter even after gentle rinsing (the Na correction). These complexities challenge remote sensing algorithms (Balch and Mitchell, 2023) or the use of PIC as a paleoproxy indicator (Beaufort et al., 2011). This emphasizes the value of microscopy counts, which have been found to be effective in quantifying PIC due to coccolithophores (D'Amario et al., 2018; Guerreiro et al., 2021; Ziveri et al., 2023).

In the northern Chilean coast, potential lithogenic PIC sources should be negligible. The exceptionally arid Atacama desert means fluvial inputs are neglible ( ) and the deep and steep topography of the Atacama Trench acts as a major depocenter effectively trapping sediments and limiting their resuspension into the upper water column (Xu et al., 2021). As sources of lithogenic PIC to these coastal waters are limited, we consider most PIC is likely biogenic.

In addition to PIC production by other planktonic organisms, which would not be detected by the microscopy protocols used here, estimates of $PIC_{Cocco}$ by microscopy also include several sources of error which can cause underestimates of $PIC_{Cocco}$ relative to $PIC_{Total}$ due to difficulties in detection of smaller coccoliths, collapsed coccospheres, and fragmented coccoliths. Other sources of error relate to taxonomic and phenotypic variability in conversion factors, PIC quotas per coccolith, and estimates of the number of coccoliths per coccosphere. Young and Ziveri (2000) suggested that these considerations might result in up to 50% errors in the estimation of $PIC_{Cocco}$ using microscopic methods. Despite these important limitations, $PIC_{Cocco}$ accounted for nearly half of direct $PIC_{Total}$ measurements and $PIC_{Cocco}$ values calculated from coccosphere and coccolith abundances were linearly correlated with chemical measurements of $PIC_{Total}$. These results emphasize the importance of coccolithophores as contributors to total PIC pools in this OMZ and also mean that using the two methods together improves confidence in the estimation of PIC pools in the OMZ system.

The patterns of PIC detected *in situ* correlated spatially with satellite PIC estimates but there were some quantitative differences. Our dataset revealed relatively higher coccolithophore-produced PIC during November and February at the border of the Southeast Pacific "PIC-data desert" (i.e., compared to the Atlantic Ocean; see Balch et al., 2018). In this context, the weekly and monthly satellite-PIC estimations, which did not exceed 10 µg C L$^{-1}$ (average ~ 2.4-3.6 µg C L$^{-1}$; Fig. 3), were above the *in situ* PIC pools by a factor of 3-5. Satellite-PIC estimates have been reported to

overestimate the PIC$_{Cocco}$ by a factor of 2-5 across the New Zealand and Drake Passage sectors of the Southern Ocean (Saavedra-Pellitero, 2024). It has been recently stressed that these optical PIC proxies need to be geographically adjusted (Balch and Mitchell, 2023), however contrasts between satellite and *in situ* estimates of PIC can also reflect the distinct spatial and temporal scales and resolutions of the measurements, which would be reduced by increased *in situ* coverage. Given these constraints, both types of *in situ* measurements from near surface waters aligned qualitatively well with satellite measures."

POC data: Though PIC:POC ratios are a central aspect to this study the actual POC data is not presented. The absence of the POC data is a curious omission that is not explained and weakens the study. If at all possible this data should be included and not simply alluded to.

**We agree that the POC data is a critical component of our study. We had not presented it directly before as this data is previously published. However, the reviewer has convinced us that it is necessary to include presenting the data in the main text. In the updated manuscript, we incorporate the POC data into the relevant sections to address this concern.**

**These data, as well as PIC and POC data from other regions used to calculate the PIC:POC ratios, are publicly available and have been appropriately referenced in the manuscript to ensure transparency and accessibility.**

Results: Primarily section 3.1 (but see below). There is a sense that the description of where maximum values were found emphasises Transect T1-T6 and overlooks the broader spatial distributions of parameters along Transect L1-Hyd7, particularly for PIC and coccoliths. Movement of Figure S14 into the main text may help mitigate any uncertainty caused by the unfortunate gap in data along Transect T1-T6 by providing better spatial context and allowing the authors/readers to gauge the representativeness of the in-situ data (i.e. were key spatial features, such as regions with high PIC offshore, missed by the sampling?).

**We responded to this valuable suggestion by moving the satellite PIC maps into a figure in the main text, a new Fig. 3. We will now explicitly note in the Results where enhanced coccolithophore stocks and PIC pools coincided in satellite and in situ data during the 2015 and 2018 sampling, and we will add a paragraph to the Discussion to this effect (see above for proposed expansion of Section 4.2).**
**We agree that including Fig. S14 in the main text enhances the context of sampling for PIC and coccolithophores by situating it within the broader spatial framework provided by satellite-derived PIC.**
**The new Fig. 3 of satellite imagery has been refined from the original S14, and we can add details on data source and processing in Methods Section 2.2.**
"Lastly, we utilize the satellite PIC as proxy of coccolithophore standing stocks (Balch and Mitchell, 2023) to produce synoptic maps on the Southeast Pacific margin. To do this, the monthly and weekly PIC climatologies (November-December 2015 and January-February 2018) were obtained from the MODIS-Aqua mission (NASA Goddard Space

Flight Center, Ocean Ecology Laboratory, and Ocean Biology Processing Group, 2022)The data were then converted from mol CaCO$_3$ m$^{-3}$ to µg C L$^{-1}$ by multiplying by 12010 and plotted using RStudio.
"

**A description to be added in Result Section 3.1 will explain how it addresses gaps in sampling coverage:**

"During the 2015 sampling period (Fig. 3a-d), satellite PIC concentrations exhibited notable spatial variability, with peaks (> 10 µg L$^{-1}$) near 20° S effectively captured by sampling locations. However, a significant latitudinal gap was observed between 25° S and 30° S, where elevated offshore PIC concentrations were missed. In 2018 (Fig. 3e-h), PIC peaks at 20° S were again well-represented by sampling. South of 20° S (up to 24° S), a gap in coverage occurred as the sampling transect extended westward, away from the coastal band of elevated PIC concentrations. Likewise, the lowest satellite PIC corresponded to relatively low PIC in surface samples (Fig. 3). Overall, our discrete sampling captured snapshots of oceanographic processes along the Southeast Pacific. Notably, PIC levels remained relatively high off ~20° S (Fig. 3), highlighting this region as a potential hotspot for coccolithophore PIC production."

[Figure]

**Figure 3. Monthly (a) and weekly (b-d) satellite-PIC climatologies during the LowpHOX 1 sampling (27-28 nov./05-09 dec. 2015; open circles), along with monthly (e) and weekly (f-h) satellite-PIC climatologies for the LowpHOX 2 sampling (30 ene./03-09 feb./12-13 feb. 2018; open circles).**

**Finally, this information will be incorporated into the Discussion, particularly in the last paragraph of the proposed expansion of section 4.2 above.**

*Minor comments*

Page 1 Line 23: This statement appears at odds with the authors conclusion that upwelling of nutrients and generation of non-coccolithophore POC may be the cause of the low PIC:POC ratios. The link specifically to low O2/low pH waters is therefore not quite correct.

**We propose to replace these sentences of the Abstract to be clearer:**

"Our findings are consistent with the prediction that the presence of a shallow OMZ in an upwelling region promotes POC production by phytoplankton other than PIC-producing coccolithophores through the injection of nutrient-rich but low pH water, decreasing PIC:POC ratios, and that the role of PIC in POC sedimentation might be decreased in such conditions. We highlight that comparing PIC in diverse upwelling conditions will be valuable to unravel how its role in POC fluxes may be affected by low pH and low $O_2$ conditions."

P3 L79: Please specify the Copernicus product used for Kd490

**Thank you for pointing this out. The Copernicus specific product citation will be included in the updated manuscript.**

P3 L83: The definition of the OMZ core was not clear to me and use of the maximum O2 concentration seems counterintuitive when defining the core of the oxygen minimum feature. Please check definition and expand how it was defined.

**We appreciate the Referee's comments and the opportunity to clarify the definition of the OMZ core in our study.**

**We can replace the actual percentage-based approach with 20 µmol kg$^{-1}$ fixed threshold, which frequently used. This results in a very minor update of the figures.**

"Lastly, the OMZ core was defined as the water layer where oxygen concentrations fell below a threshold of 20 µmol kg$^{-1}$. The upper and lower boundaries delimiting the OMZ core (Fig. 1b) are consistent with those discussed by Vargas et al. (2021) during the same cruises."

P3 L91: Please quantify the magnitude of the applied correction made to PIC measurements for Na residues. It is unclear if this is significant.

**We appreciate the Referee's comment regarding the correction for Na residues in PIC measurements. In our study, Ca from residual seawater was estimated from Na and subtracted. This calculation indicated that residual seawater contributed on average 29.1% ± 25.8% of total Ca in LowpHOx 1 samples and on average 35.2% ± 21.5% of total Ca in LowpHOx 2 samples.**

**It is important to note that, to the best of our knowledge, corrections for Na residues in PIC measurements have not been explicitly reported in the previous studies we compare to. Nevertheless, applying this correction aligns with the general practice in carbonate chemistry to mitigate contamination and improve the reliability of the results (e.g., procedures described in the GEOTRACES program and other inorganic carbon measurement guidelines).**

**As mentioned above, we offer above a modified section 4.2 of the Discussion where we consider all the different sources of uncertainty in PIC measurements, whether by chemical or microscopic estimates.**

P3 L92: Typo, presumably this should be 'PIC concentrations' not 'calculations'

**Yes, thank you. We will correct it in the revised manuscript.**

P4 L94/95: The terms 'built' and 'building' seem inappropriate and the intention here is unclear. Please rephrase.

**Thank you for pointing this out! We will rephrase it in the revised manuscript.**

P4 L96: Direct reference to assessing the effect of this OMZ on POC concentrations suggests that the POC data are central to this study and therefore should be presented alongside the PIC data.

**We propose to include this now.**

P4 L97-99: What criteria were used to define the depth intervals used in this study? As the subsurface interval (5-100 m) presumably crosses the mixed layer there are strong gradients to consider in the distribution of both PIC and coccolithophore diversity which may be lost by the depth bins used.

**The original depth bins used in the dataset were 0–5 m, 5–100 m, and 100–500 m. These finer intervals were initially designed to capture vertical gradients, including those within the mixed layer and deeper subsurface layers. However, now we re-binned the data from the global comparison into broader intervals of 0–100 m and 100–400 m. These new intervals were specifically chosen to be comparable with the "above" and "within" categories relative to the OMZ. We recognize that this approach may result in the loss of some fine-scale gradients, particularly in the mixed layer (which typically ranged between 50 and 100 m depth in the other publicly available datasets). Nevertheless, the re-binned depth intervals allow for more simple comparisons of broader ecological patterns across OMZ and non-OMZ regions. We will explicitly outline the bin choice criteria at the end of Section 2.2, as well as provide a discussion of the potential trade-offs involved in this binning approach and its implications for interpreting PIC, POC, and PIC:POC ratios as follows:**

"These ratios were then categorized into two groups: above and within the OMZ core, to assess the influence of the OMZ on PIC and POC concentrations. These ratios were plotted against those reported for other open ocean or coastal margins (see Balch et al., 2018). To this end, PIC and POC data was obtained from the SEABASS (Werdell

et al., 2003) and BCO-DMO repositories (Balch, 2010), and PIC:POC ratios were binned for the well-mixed surface (0-100 m depth) and the stable sub-surface layer (100-400 m). These depth intervals were specifically chosen to be roughly comparable with the above and below categories relative to the OMZ, providing a consistent framework for direct comparisons of broader ecological patterns across OMZ and non-OMZ regions."

Figure 1: The figure legend needs a better description of what the black and grey lines actually represent (the odd placement of the plot legend to the right of panel b was initially overlooked). The black and grey lines need to be better distinguished either by changing the line style or line thickness.

**Thank you for pointing this out. We have updated the figure legend and ensure the threshold enclosing the OMZ core are most clearly delineated by using thicker white lines for improved visualization (see below).**

[Figure]

**Figure 1: (a) Map of the Southeast Pacific margin showing the study site and stations sampled during late-spring 2015 (circles) and mid-summer 2018 (crosses). (b) Sampling depth coverage for coccolithophores, highlighting areas crossing the**

thresholds enclosing the OMZ core (dotted and dashed red lines). Map produced by Ocean Data View (Schlitzer, 2024), with bathymetry based on the GEBCO chart (GEBCO, 2023).

P5 L116/L126: Figure S4b implies a near constant bias exists between light microscopy and SEM methods when estimating coccolith abundances. As the identity of the two axes in Figure S4 are unclear I am presuming that the SEM counts are on the y-axis (and LM counts on the x-axis) in which case SEM coccolith counts are higher than LM counts, even though the general relationship between the two methods is linear. How significant is this bias and what does it mean for the results of this study? (particularly coccolith counts for stations T1-T6 given the reliance upon SEM; Line 116)

**We fix Fig. S4 (now S5) so that the axes are labelled to make clear which are SEM counts, and which are light microscopy counts. We also now report the standard errors of the slopes and the intercepts. In all cases, the slopes were highly linear with $R^2$ greater than 0.9, and were not significantly different from 1, while the intercepts were close to 0. In the case of detached coccoliths, there was a significant difference in the intercept, but it was very minor and would not result in any significant changes to PIC estimates.**

**We relied on polarized light LM counts for almost all absolute counts, because one can cover much more of each sample for a lower cost in time as well as minimizing expensive SEM time. However, the comparison of LM to SEM counts was only necessary because a small number of samples could not be counted by LM, and we could only count with SEM, due to filters that were very bright in polarized light, for reasons we could not determine.**

**We also complement with a new analysis of the error from counting effort. For both LM and SEM, we compute the standard error of the mean (SE in %) among different fields of view vs the total number of coccospheres or coccoliths counted (considering all fields). As expected, the SE drops hyperbolically with the total number of coccospheres or coccoliths counted. The number or fields counted was constant at 20 and 11 for coccospheres and detached coccoliths in LM and between 27-48 and 2-5 for coccospheres and detached coccoliths in SEM depending on magnification, respectively, allowing us to estimate the expected error of counts within each sample, and we report this error as a percentage in the new Supplementary Figure S4, with actual Fig. S4 becoming S5.**

[Figure]

Relationship between the counted coccospheres and detached coccoliths with the standard error of the mean (SE in %) of counts performed using cross-polarized light microscopy (a-b) and scanning electron microscopy (c-d). Inset histograms show the frequency distribution of SE values. The solid black line represents the fitted hyperbolic curve, with shaded grey areas indicating the 95% confidence intervals. Counts and SEs showed a significant hyperbolic relationship for coccospheres (a): equation; $R^2_{adjusted} = 0.89$; $p$-value$_{slope, constant} < 0.05$, and detached coccoliths obtained with cross-polarized light microscopy (b): equation; $R^2_{adjusted} = 0.59$; $p$-value$_{slope, constant} < 0.05$, as well as, coccospheres (c): equation; $R^2_{adjusted} = 0.89$; $p$-value$_{slope, constant} < 0.05$, and detached coccoliths obtained with scanning electron microscopy (d): equation; $R^2_{adjusted} = 0.42$; $p$-value$_{slope} < 0.05$.

[Figure]

**Figure S4. Linear relationships for coccosphere counts (a) and detached coccoliths counts (b) obtained through scanning electron microscopy (SEM) and cross-polarized light microscopy (LM). The dotted line represents the 1:1 ratio, while the solid black line represents the lineal trend. SEM and LM approaches showed a significant correlation for coccospheres (a): equation Y=0.899(±0.143)X+3.121(±13.038; $R^2_{adjusted}$ = 0.91; _p_-value$_{slope}$ < 0.05, as well as for detached coccoliths counts (b): equation Y=0.979(±0.134)X+5.760(±5.478); $R^2_{adjusted}$ = 0.93; _p_-value$_{slope}$ < 0.05.**

P5 L129-133: The approach used to estimate absolute abundances of species/genus coccospheres and coccoliths from SEM images is a little unclear. Based on counts from a low number of images (magnification not reported) I do not understand the rational for multiplying the counts by the total coccosphere or coccolith abundances to obtain total species counts particularly if there are biases between LM and SEM coccolith counts (mainly applicable to stations L1,L2 and L3). Should not the same approach as described in section 2.3 be used i.e. the equation in Diaz-Rosas et al 2021 thereby accounting for volumetric factors? I am concerned that there could be a scaling error here resulting from the use of mixed methodologies. Please clarify.

**While SEM provides higher-resolution imaging, its operational costs are substantial, so total counts (for absolute abundances of total coccolithophores and coccoliths) are more efficiently made by LM. Therefore, the main use of SEM was to obtain relative abundances of the principal taxa with higher resolution, exactly the strategy followed in Diaz-Rosas et al. 2021 (again, a separate protocol for the SEM was used for counting a small number of samples where the filters were too bright under cross-polarized light to permit LM counts – we could not determine the source of that high brightness as they looked the same as other samples under SEM).**

**The principal difference with Diaz-Rosas et al. 2021, where we only analyzed epipelagic samples, was that here we analyzed mesopelagic samples, which are much more dilute in phytoplankton. Even with the maximum volumes of water we could filter before filters clogged, filters of deep samples had much sparser coccospheres and coccoliths compared to surface samples. As a result, we had to**

be much more careful with SEM time, and this obligated us to work to obtain a minimum sample coverage rather than a minimum number of coccospheres analyzed for the SEM analysis. On average, 165 coccospheres per sample were analyzed by SEM, but in some deeper samples, the numbers were much lower (the range was 1 to 1140 coccospheres). When samples from different stations are grouped by depth layer, rarefaction analysis shows that we still had a decent view of total diversity from the subsurface layers. The main conclusions of this paper are based on estimating the pools of PIC and coccolithophore-derived PIC in the surface and subsurface layers of an OMZ region, because spatial variability in the subsurface waters is more difficult to sample for these reasons.

For smaller detached coccoliths, we assumed their origin to be from the G. parvula/ericsonii assemblage, based on prior studies in the same area (e.g., see Beaufort et al., 2008; von Dassow et al., 2018; Díaz-Rosas et al. 2021). The dominance of G. huxleyi coccospheres in absolute coccospheres counts allowed us to reduce SEM image re-analysis for species/genus classification by approximately 25%, while maintaining the same effort for coccolith counts.

[Figure]

**Rarefaction-extrapolation analysis for late-spring 2015 and mid-summer 2018, showing species richness (a-b), the exponential of Shannon entropy (c-d), and sample completeness (e-f) for coccospheres and detached coccoliths observed at 2-5, 10, 50, and 100 m. Each curve includes 95 % confidence intervals.**

P5 L137: There is ambiguity in the methodology over whether species/genus level counts were obtained from SEM images and individual species/genus PIC quotas calculated or whether a size-class based approach was used with a mean size based conversion factor.

**We can make more clear that the larger spatial and temporal study including samples from both 2015 and 2018 was based on LM images, whereas the focused approach only using 2015 samples used SEM to be able to better assess taxonomic composition and in particular whether the taxa contributing to PIC pools in the surface layer differed from the taxa contributing to PIC pools in the subsurface OMZ core.**

Figure 2: This is a complex figure to understand which is made harder by the splitting of this figure across 2 pages. The separation of station names (along top of panel 2e) and the lat/lon (along bottom of panel 2n) is unfortunate and this information really needs to be present on both pages to help the reader. Other issues are the difficulty in seeing the white dashed line (euphotic depth) in all panels. Given the reliance upon POC data, why is there no POC section in Figure 2? Key contours or features cited in text should be more clearly visible.

**Thank you for your feedback. To enhance visualization, we offer a modified figure (now Fig. 2, which extends across two pages) where we ensure that both parts include latitude/longitude coordinates and station labels. The euphotic zone is made more prominent by using white thicker lines, and additional panels for POC will be included in the second part to enable direct comparison with PIC. The contours will be simplified to those most discussed in the text. However, we are unable to adjust the contours line width, as the functionality does not appear to be fully supported by Ocean Data View (version 5.8.0).**

[Figure]

[Figure]

P9 L184-196: There are several unclear statements here that can be clarified. For example i) L185 near surface waters <25m vs L192 surface waters 0-30m; ii) L187 highest PICtotal in 2018 was 5.3 mmol/m3 vs L189 highest PIC in 2018 was 5.86 mmol/m3; iii) apparent bias in emphasising max PICtotal in 2018 as being along Line T1-T6 when station Hyd6 had a higher concentration. Overall, this paragraph was a little muddled and unclear and lacking detail or precision in the reporting of results.

**Thank you for your feedback. We will update the paragraph to clarify the depth ranges, address the discrepancy in the reported PIC$_{Total}$ values, and highlight the peak in PIC$_{Total}$ observed at station Hyd6 during 2018.**

P10 L220: It is not clear where the value of 67% was derived from. I do not see this in Figure 5c?

**We will correct the text to clarify that the detached coccolith fraction represents 63% of the total accounted PIC$_{Total}$ within 100 m depth, based on the updated calculation (30 x 100 / 48 = 63%). The previously mentioned value of 67%, which referred to all samples, has been replaced with the value specific to the 100 m depth.**

Figure 3: Panel b gives the impression of monospecific coccolith distributions at many sampled depths due to the approach used of grouping all liths <4um together. This is limitation that needs to be briefly addressed in the discussion.

The dominance of the detached coccoliths pool by G. huxleyi was confirmed during the analysis of lower magnification SEM images and was also confirmed in selected higher magnification images. The inclusion of smaller coccoliths into the <4 μm category reflects this observation, although their contribution is likely negligible, and the category could essentially be referred to as 'G. huxleyi'. To address this more explicitly, we will add the following sentence to the Results Section *Diversity of coccospheres and detached coccoliths*:

"Despite the presence of *G. parvula/ericsonii*, its small coccoliths were underrepresented, reinforcing the overwhelming prevalence of *G. huxleyi* in the samples"

Figure 4: Figure 4a arguably reproduces some of the data contoured in Figure 2k-n, and presented in Figure S5 so the figures could be simplified. Also, I understand PICcocco to be a bulk term representing the total contribution to PIC by both coccosphere and coccolith PIC, thus it seems wrong to present the contribution of spheres and liths to PICcocco in two subplots with axes reaching 100% in both (fig 4b & 4c). How can both axes be correct when panel 4a indicates a changing contribution by both coccospheres and liths to PICcocco? (evident for station T1,T2 but an unclear contribution by coccospheres for T3-T6).

We appreciate the Referee's concern regarding potential redundancy between the $PIC_{Cocco}$ values presented in Figures 4a, 2k-n, and S5, as well as the decomposition of the $PIC_{Cocco}$ pool in Figures 4b and 4c. First, it's important to note that the $PIC_{Cocco}$ values in Figures 4b and 4c are based on species/genus-level conversion factors, whereas the $PIC_{Cocco}$ values in Figures 2k-n and S5 (as well as related Figures 5, 6d and 7c) are derived using the G. huxleyi conversion factor as a maximum threshold. This distinction is outlined at the end of the Methods. Second, we clarify that Figures 4b and 4c presents the relative contributions of coccospheres and detached coccoliths to the $PIC_{Cocco}$ pool, expressed as percentages of the total $PIC_{Cocco}$ pool. These percentages are calculated within the $PIC_{Cocco}$ fraction and do not represent absolute contributions to the $PIC_{Total}$ pool. We will update the figure caption to reflect this distinction:

"Figure 4: Estimated PIC masses from coccospheres and detached coccoliths recorded in waters off Iquique (~ 20⁰ S) during late-spring 2015. (a) Contribution of coccospheres and detached coccoliths to the total $PIC_{Cocco}$ pool. (b) Taxonomic breakdown of the relative contribution of coccospheres (b) and detached coccoliths to PICCocco quotas (c), expressed as percentages of the total $PIC_{Cocco}$ pool."

It is important to highlight that the figures in question are primarily aimed at describing the diversity of coccospheres and coccoliths allocated to the total $PIC_{Cocco}$ pool. These figures provide essential context for understanding the taxonomic composition of the coccolithophore community and their contributions to PIC. However, to ensure consistency across the dataset, the PIC estimation was extended to all samples using the conversion factor of the most abundant species (G. huxleyi). While some samples have species/genus-level resolution, this generalized approach allowed for a robust estimation of the maximum potential PIC contribution.

P13 L235: Typo in legend of figure 5 (concentsdaration)

**This will be correct in the updated manuscript.**

P13 L238: The phrase 'marginally higher' is ambiguous without a quantified value or statistical support. Is the difference significant?

**We can be more clear. There were no statistical differences between the coccospheres, detached coccoliths, $PIC_{Total}$, and $PIC_{Cocco}$ across the 2015 and 2018 cruises although peaks are notable in 2018. We will revise the text as follows: "Coccospheres, detached coccoliths, PIC pools, and the estimated $PIC_{Cocco}$ quotas were not statistically different ($p > 0.05$) between the 2015 and 2018 cruises, although there is a slight observable difference in the maximum values, with mid-summer 2018 showing higher peaks compared to late-spring 2015 (Fig. 6)."**

P13 L245: Typo, panel 2c,g ? not 2h?

**Thank you for your comment. We will double-check these panels references are accurate as per the panels shown.**

P13 L249: Typo, panel 2e-f, not 2f-g?

**We appreciate your careful attention. The missing "j" for $pCO_2$ panel will be add in the updated manuscript.**

P15 L82: Can remove approximation by stating actual results (45-48%)

**We appreciate the suggestion and agree that providing exact values can enhance clarity. However, since these values are explicitly detailed in the Results section, we believe that rounding to approximate percentages in the Discussion helps to convey a more concise and accessible message. This approach aligns with the purpose of the Discussion section, which is to synthetize the key findings.**

P16 L303: Would be useful to state the ratios from Balch et al 1991, Holligan et al 1993b that were used in this comparison.

**We agree. We will include a detached-coccolith-to-coccosphere ratio of > 250, which encompasses values from both studies.**

P16 L303/L306/L343/L411: No need to abbreviate maximum to max.

**We will replace 'max' with 'maximum' throughout the text.**

P16 L305: It is not clear where the stated values of cell-attached coccolith contribution to PICcocco (51-72%) come from. Please clarify and highlight in the results.

**Thank you for raising this point. To clarify, the stated values for the cell-attached coccolith contribution to $PIC_{Cocco}$ were calculated from the same samples used for the detached-coccolith-to-coccosphere ratio analysis mentioned in the preceding sentence. This represents the percentage of the total $PIC_{Cocco}$ quota accounted for by coccospheres. However, since this analysis provides an interpretative perspective rather than presenting raw observational data, we chose to include it in the Discussion rather than in Results.**

**To enhance clarity and directly address your concern, we will add "for these specific samples" in the Discussion to emphasize that these values correspond to the same samples used in the detached-coccolith-to-coccosphere ratio calculations. This ensures transparency while maintaining the logical flow of the manuscript.**

P17 L332: Missing appropriate references (for Calcite Belt, Bay of Biscay)

**We will add the appropriate references for the studies conducted in the Calcite Belt and the Bay of Biscay.**

P19 L373: From Figure 5c I do not see how the statement that up to two-thirds of the PIC_cococ quota comes from detached coccoliths can be correct? Please clarify

**Thank you for your concern, which is similar to a previous point raised. The value provided represents an approximation of the percentage of the total accounted $PIC_{Total}$ (48%) contributed by the detached coccolith fraction (30%) within the 100 m depth range. Thus, 30 x 100 / 48 = 63%, which accounts approximately two-thirds of the total $PIC_{Cocco}$. We will clarify this in the updated manuscript.**

P20 L388: The observation that the PIC:POC ratio (Figure 8c) is greatly reduced compared to other areas is intriguing despite the comparable PIC standing stocks (Figure 8b). Without more detail on the coincident POC dataset however it is difficult to rationalise this observation beyond the suggestion put forward by the authors that upwelling stimulates non-calcareous phytoplankton. For this reason, the authors should consider including the POC dataset in this study. It may be particularly important to ascertain the similarity or differences in POC concentrations between the various studies/sites used for comparison to validate the conclusions reached. Also, the PIC:POC results appear most comparable to results from the W. Arctic, which is not an upwelling zone. This point needs to be highlighted. What could be the cause of this similarity?

**We appreciate your concern regarding the accuracy of the PIC:POC ratio analysis. During the preparation of the POC data for plotting, we identified an error in the PIC values for the OMZ dataset. Specifically, the PIC values were incorrectly expressed in mmol C m$^{-3}$ instead of µg C L$^{-1}$, resulting in disproportionately high PIC levels. This issue has now been corrected, and the POC data have also been added. We present an updated version of Figure 8, which includes the corrected POC data, the newly added POC data, and revised binning of the external datasets (see below). The findings derived from Figure 8 will be updated in the respective sections of the manuscript.**

[Figure]

**Figure 8: (a) Global map showing the annual oxygen content at 100 m depth and the sampling locations for (b) PIC$_{Total}$, (c) POC, and (d) PIC:POC ratios above and within the OMZ-core during late-spring 2015 and mid-summer 2018 in the SE Pacific (this study). Additional data represent the well-mixed surface (0-100 m) and the stable sub-surface layer (100-400 m) from other open ocean or coastal margin regions (data from Balch et al., 2018). The Atlantic Ocean dataset includes samples from six cruises (AMT17-22). One-way ANOVA results indicate significant differences (p < 0.05) in PIC$_{Total}$, POC, and PIC:POC ratios among SE Pacific OMZ-core depths, as well as across sample groups from the Atlantic Ocean, Southern Ocean, Indian Ocean, Western Arctic, and Patagonian Shelf (Atlantic). Tukey post-hoc comparisons are represented by lowercase letters above each boxplot. Only sample groups with more than 20 data points for both PIC and POC were included in the analysis. The map was generated using Ocean Data View (Schlitzer, 2024), with oxygen climatology based on the World Ocean Atlas 2018 (Boyer et al., 2018; García et al., 2018).**

**Regarding the comparable PIC:POC ratios, and the now observed low PIC and high POC levels between the Western Arctic and SE Pacific OMZ (Fig. 8), upwelling events in the Western Arctic significantly influence the distribution and composition of POC and PIC. These upwelling events, primarily driven by strong northeasterly winds, facilitate the vertical movement of nutrient-rich waters from deeper layers to the surface, particularly in regions like Barrow Canyon and the Chukchi Sea (see Li et al., 2022, and references therein). These events support high biological productivity, including diatom-dominated phytoplankton communities. At the same time, coccolithophore penetration of the Arctic is known to be limited (e.g. Winter et al. 2014). As a result, the low PIC and high POC levels observed in the Western Arctic are notably lower in PIC and higher in POC compared to other open ocean and coastal margin regions.**

**We propose to address this in the Discussion.  Proposed text to a revised subsection "4.3 Surface variation in coccolithophores and PIC pools":**

"Although the periods sampled in this study were characterized by stratified summer water column conditions which may be conducive to coccolithophore growth (e.g., Matson et al., 2019), it is noteworthy that this zone, characterized by exceptionally low pH in sub-surface waters frequently brought to the surface by upwelling, does exhibit a tendency to lower PIC compared other ocean regions (Fig. 9a-b). The prominent exception was the Western Arctic Chukchi Sea, which also was reported to have lower PIC pools. Coccolithophores are known to have limited penetration of Arctic waters (e.g., Winter et al. 2014), and the Chukchi Sea is a site of high diatom productivity due to upwelling (Li et al. 2022). With the caveat that data from upwelling regions are still limited globally, these results are consistent with the prediction that the intrusion of nutrient-rich but low pH waters into the surface stimulates POC production from other phytoplankton (e.g., diatoms) while repressing coccolithophores. "

**Proposed text to a revised subsection "4.4 Subsurface variation in PIC and PIC:POC ratios":**

"Nevertheless, the PIC:POC ratios were significantly lower than those observed in other open-ocean and coastal sites in both surface and subsurface waters (Fig. 9d). The prominent exception was the Western Arctic Chukchi Sea, as discussed above. These findings, while emphasize that comparing diverse upwelling systems will be essential to resolving the role of coccolithophore PIC in POC fluxes and potential interactions with pH/low $O_2$ waters, are consistent with the prediction that PIC may play a lower role in POC fluxes in OMZ conditions."

P22 L413: Conclusion 6 seems to contradict statements on P19 L379. On P19 is the statement that the PIC:POC ratio was significantly elevated in the OMZ core (due to a ballast effect), whereas on L413 the (relatively) low PIC:POC ratio of the OMZ core is highlighted for its difference to other coastal margin areas. Whilst the overall conclusion that this OMZ exhibits lower PIC:POC ratios compared to other locations is valid, it also seems that when examined in detail the core of this OMZ is associated with elevated PIC:POC ratios (Figure 8c), thus the broader significance of this could be addressed in section 4.4.

**Thank you for your feedback. We will update Conclusion 6 to incorporate the new findings on PIC:POC ratios as detailed in the minor comment above.**

Table S1: Are the units for PIC concentration incorrect? (i.e. uM not mM?)

**Yes, thank you. We have corrected the units to µg C L$^{-1}$.**

Figure S10: The SEM images have reproduced poorly in my copy. Maybe upload the SEM image files separately to allow greater accessibility. Missing word in Figure legend.

**All scanning electron microscopy images used in this study have been stored in the Zenodo public repository. This dataset is referenced as:**

**Díaz-Rosas, F., Vargas, C. A., and von Dassow, P.: Scanning Electron Microscopy Datasets – Coccospheres and detached coccoliths in waters off the Southeast Pacific margin, https://doi.org/10.5281/zenodo.14048319, 2024.**

**It is publicly available and so we consider it most efficient to share these images by that method. We also upload here separately a high resolution version specifically of what is now Fig. S11.**

**The legend has been corrected to:**
"**Figure S11. Example scanning electron microscopy images showing diverse coccolithophore and diatom assemblages in 2015 station T1 at 2 m depth (a), the dominance of the coccolithophore component by coccospheres and detached-coccoliths of *G. huxleyi* during a diatom bloom in 2015 station L2 at 5 m depth (b), and example low biomass conditions in 2015 station L3 at 5 m depth (c). Each 800x frame corresponds to 0.2 mm$^2$.**"

Figure S11: These LM images also reproduced poorly. Maybe upload the image files as well.

**All cross-polarized light microscopy images shown in this figure have been stored in the Zenodo public repository. This dataset is referenced as:**

**Díaz-Rosas, F.: Cross-polarized light microscopy images – Coccospheres and detached coccoliths in waters off the Southeast Pacific margin, https://doi.org/10.5281/zenodo.14708540, 2025.**

**It is publicly available. We also upload here separately a high resolution version specifically of what is now Fig. S12.**

Figure S14: I find this figure to be a useful means of assessing the spatial variability of PIC during both cruise periods. I would encourage the authors to consider moving this figure into the main text as it provides useful context.

**Thank you for the feedback, which has been addressed in the major comment above.**

---

## Author Comment (AC2)

The article by Díaz-Rosas et al. presents an interesting examination of coccolithophore impacts on PIC standing stocks in relation to upwelling and the oxygen minimum off the coast of Chile. The authors compare coccolithophore based estimates of PIC and discrete PIC chemical measurements and show that only 50% of the PIC is sourced from coccolithophores in these waters. There is also a strong gradient in species contributions which align with the base of the euphotic zone and upper part of the oxygen minimum zone. The authors explore these observations in terms of the context of the fate of sinking organic carbon and how reduced PIC ballasting of sinking organic carbon could strengthen low oxygen conditions.

The article is well written and contains interesting observations and measurements, however several issues need to be further clarified. Firstly, the link between PIC production, ocean acidification (pH) and PIC standing stocks, sinking POC ballasting by PIC, and the oxygen minimum zone needs to be better explained in the introduction – at the moment there is no mention of PIC ballasting so that the link between pH conditions and $O_2$ concentrations is not clear. Secondly, a lot of material is found in the supplementary material rather than the main article and this makes following the article difficult – this is especially in the case where the authors compare their coccolithophore and PIC dynamics with environmental conditions. This is wholly in the supplementary material and if the authors consider this an important element of the paper, this statistical analysis should be in the main article. Lastly, can the authors confirm that their conversions from coccolith $CaCO_3$ (their Table S3) to the values reported in the article (mmol C m$^{-3}$) took into account the molecular weight of $CaCO_3$ – for most of the article this appears to be so, but then there is a calculation in the discussion which gives very high PIC concentrations (Ln 339) and its not clear how these values were achieved.

**We thank Dr. Poulton for the supportive and careful comments and suggestions. We confirm that we checked the conversions and corrected an error in one part. We respond to the detailed comments below.**

SPECIFIC COMMENTS

Ln 12, Consider whether the term 'pools' or quotas or standing stocks (as used elsewhere in the paper) would be better.

**We appreciate your concern about the consistency of pools, quotas, and standing stocks terminology and offer to clarify them according to the following criteria:**

**- "Pools" will be used to describe integrated or total quantities, such as the cumulative PIC across a specific depth range or area, which is essential for understanding large-scale patterns. For example: "Below the Zeu, the sharp decline in the numerical abundances of coccospheres (< 0.1 x10$^5$ L$^{-1}$) and detached coccoliths (< 10 x10$^5$ L$^{-1}$) decreased coccolithophore PIC pools to below 1 µg L$^{-1}$." or "These results support the conclusion that coccolithophores are important contributors to total PIC pools in this OMZ.".**

- Quotas refer to per-cell metrics, such as PIC per coccolithophore cell, which provide insights into species-specific contributions to the PIC pools. For example, "The PIC quotas of individual coccoliths and coccospheres were estimated following Young and Ziveri (2000), and used to calculate coccolithophore-derived PIC pools (PICCocco) from abundances of coccospheres and detached coccoliths"

- The term "Standing stocks" will be used in the sense of organism stocks (such as fish stock assessments). Here it will be employed to capture instantaneous measurements of total coccolithophore abundances at specific depths or times, helping to characterize temporal and spatial variability. For example: "Standing stocks of larger taxa were only noticeable below the $Z_{eu}$ (~50 m depth)." or "This could represent the maximum coccolithophore standing stocks and PIC pools attainable in the Southeast Pacific margin".

Each of these terms corresponds to a distinct analytical perspective reported in the manuscript, and using only one term would oversimplify the multifaceted nature of our findings. We will correct the manuscript to use them consistently in the appropriate contexts. For example, the header of the first section of Results will be corrected to

"3.1 Oceanographic conditions. Standing stocks of coccolithophores, pools of PIC and POC, and PIC:POC ratios"

Lns 12-13, The links between ocean acidification, PIC concentrations and production, the efficient ballasting of sinking POC, and the impact of remineralisation depth need to be further expanded on in the first couple of lines of the abstract; the links are not obvious without further explanation.

Thank you for pointing this out. We updated the first lines of the Abstract as follow:

"A predicted consequence of ocean acidification is to reduce coccolithophore-produced Particulate Inorganic Carbon (PIC) pools. PIC is thought to promote the sinking of Particulate Organic Carbon (POC) to deeper waters, potentially influencing the depth of organic matter remineralization and subsurface oxygen levels."

Ln 17, What do the authors mean by 'countable coccoliths'? Do they mean detached coccoliths, or are they hinting at a portion of coccoliths that is not countable?

Thank you for pointing out the ambiguity of "countable coccoliths". It refers specifically to coccospheres and coccoliths observed and enumerated under the microscope. This excludes any coccoliths that were too small, fragmented, or otherwise indistinguishable for reliable counting. The term does not imply the

existence of a portion of coccoliths that is inherently uncountable but rather reflects the limitations of our counting method. We will update the sentence as follows:

**"On average, about half of the PIC was attributed to reliably enumerated coccospheres and detached coccoliths, with diminished proportions observed below the euphotic depth."**

Ln 27, What about the important contributions from other calcifying plankton? As the paper concludes that coccolithophore PIC is only around 50% of PIC standing stocks, would it not be good to expand on other potential sources of biological PIC?

**We appreciate the suggestion to expand on other potential sources of biological PIC. We fully agree that other calcifiers play significant roles in the biological PIC pool. However, in the Abstract and Introduction, we specifically limit the scope of our study to phytoplankton calcification, excluding foraminifera and pteropods zooplankton from our analysis. Nevertheless, in response to another comment, we will acknowledge the roles of other calcifying plankton in the revised discussion. For example "How much coccolithophores contribute to PIC remains an open question, as contributions from calcifying zooplankton (e.g., foraminifera, pteropods; Ziveri et al., 2023), lithogenic sources (Daniels et al., 2012), and processes like fragmentation and dissolution in the water column (e.g., Barrett et al., 2014; Subhas et al., 2022) complicate the relationship between PIC and coccolithophores" and "In addition to PIC production by other planktonic organisms, which would not be detected by the microscopy protocols used here,..."**

Ln 34, There is no mention of the ballast effect on POC sinking in the first paragraph of the introduction so that the link between reduced PIC production and the 'favouring of the respiration of more organic material' is not obvious. Suggest adding introduction of the ballast effect earlier in the introduction to make this link obvious.

**Thank you for your feedback. We will update the first part of the Introduction to explicitly describe the potential role of coccoliths in the sinking process or 'ballast effect' as follows:**

**"The Particulate Inorganic Carbon (PIC) pool, a significant component influencing marine C cycles and atmospheric reservoirs (Ridgwell and Zeebe, 2005), originates from various sources, including land transport to coastal margins (Cai, 2011) and biological processes such as phytoplankton calcification (Taylor and Brownlee, 2016). Coccolithophores, particularly the cosmopolitan species *Gephyrocapsa (Emiliania) huxleyi*, significantly contribute to PIC through coccolith production. These coccoliths are thought to promote the sinking of organic matter (i.e., the 'ballast effect'), facilitating the export of particulate organic carbon (POC) to deeper waters (e.g., Klaas and Archer, 2002), thereby influencing carbon sequestration and nutrient cycling in the global pelagic ocean (Monteiro et al., 2016; Balch, 2018). In turn, changes in PIC dynamics driven by alterations in coccolithophore communities at different spatial and temporal scales feedback into the ocean-atmosphere system (Balch et al., 2016; Claxton et al., 2022)."**

Ln 43, Consider the use of the term 'coccolithophore bloom' in the context of the references given – Beaufort et al. (2008) is based in the subtropical S Pacific waters and did no observe what many would consider a 'bloom'.

**Thank you for pointing this out. You're right, and we will replace 'bloom' with 'high-density events'. We will also revise its use throughout the manuscript.**

Ln 92 and 105, Can the authors please confirm that all the PIC concentrations presented in the paper are in mmol C m$^{-3}$ and that their conversions from the values in Table S3 have taken into account the molecular weight of calcite? Its not obvious that this is a problem or that a mistake has been made but a calculation in the discussion (Lns 337-339) gives much higher PIC concentrations that expected in units of mmol C m$^{-3}$.

**Thank you for your feedback. You are correct that there was an erroneous use of units, where mmol C m$^{-3}$ corresponds to µg C L$^{-1}$. We will correct the units in the figures and tables. This adjustment refines the overall narrative, particularly in relation to Fig. 8 and Table 1, as it impacts the PIC and PIC:POC ratio comparisons with other studies or sampling sites. The updated findings will be incorporated accordingly.**

Ln 102, Fig. 1 – What are the solid lines on (b) and (c)? As these lines do not line up with the discrete samples, could the legend explain more?

**Thank you for your feedback. We agree that these lines were not appropriately presented to highlight areas crossing the thresholds enclosing the OMZ core. For clarity, we will update the figure legend and ensure the OMZ core threshold is more clearly delineated by using thicker red lines for improved visualization. See response to Reviewer 1 where we include a corrected figure.**

Lns 112 and 118, Having given the proportion of volume counted from the filters for SEM and light microscope analysis (<20 mL or <2% of the volume filtered), it is a surprise that no comment is made on the potential impact this may have on the observations made – specifically the species diversity and the counts of large, often numerically rare species.

**Thank you for pointing this out. While SEM provides high-resolution imaging, its operational costs are substantial. To optimize resources, filters were scanned at lower magnifications (1000-1500x) with sufficient resolution to allow for post-scan zooming, enabling the detection of both coccospheres and detached coccoliths. For smaller detached coccoliths, we assumed their origin to be from the G. parvula/ericsonii assemblage, based on prior studies in the same area (e.g., see Beaufort et al., 2008; von Dassow et al., 2018). The dominance of G. huxleyi coccospheres in absolute coccosphere counts allowed us to reduce SEM image re-**

analysis for species/genus classification by approximately 25%, while maintaining the same effort for detached coccolith counts.

Please see our detailed comments to Reviewer 1 about these points.  Also, we provide below rarefaction curves for different layers showing how we expect that we effectively captured rare species, especially when different samples are grouped by layers. As on average over 100 coccospheres were counted per sample, we should have been able to capture rare species in all but the most dilute (deepest) samples.

[Figure]

**Rarefaction and extrapolation curves for late-spring 2015 and mid-summer 2018, showing species richness (a-b), the exponential of Shannon entropy (c-d), and sample completeness (e-f) for coccospheres and detached coccoliths observed at depth of 2-5, 10, 50 and 100 m. Each curve includes 95 % confidence intervals.**

Ln 126, What is meant by a 'good fit'? The statistics of the relationship should be presented in the main article rather than the supplementary material.

**Thank you for pointing this out. This aligns closely with Referee #1's concern regarding the potential bias introduced by the use of different microscopic methods. To address this concern, we will present a revised version of Fig. S4 (see below), specifying the counting methods used on each axis (abbreviated as SEM and LM), along with a more detailed statistical analysis to assess the agreement between them.**

Ln 162, What is the oxycline? This needs to be defined somewhere as to what the authors consider either the absolute threshold, relative change or gradient of this feature.

**Thank you for your comment. We now define the OMZ core based 20 µmol kg$^{-1}$ threshold (see response to Reviewer 1). With this change, it should become clear that the oxycline is where vertical transition between the surface oxygen levels and this threshold is steepest, with the greatest relative change between 50-150 µmol kg$^{-1}$ O$_2$.**

Ln 165, Are the authors referring to fluorescence or Chl-*a* peaks when discussing 'peaks of phytoplankton'? Please clarify.

**Thank you for your comment. We will include a threshold for higher fluorescence and Chl-a levels to clarify that we are referring to both proxies for phytoplankton biomass.**

Ln 166, In Fig. 2f the surface (<25 m) nitrate concentrations look much lower (<5 µM) than the 10 µM quoted in this line. As 10 µM is not generally considered 'low' in terms of biologically-limiting, this line should be changed to better reflect (biologically) 'low' concentrations (e.g., <2 µM).

**Thank you for your feedback. We will adjust the nitrate according to a biologically relevant threshold.**

Ln 176, Fig. 2 – ODV is notorious for problems with plotting discontinuous (discrete) data in that it often 'creates' patterns not supported by the data. The authors should review the patterns shown in the patchier data (e.g., fluorescence, Chl-*a*, PIC, coccospheres, coccoliths) as to whether they are confidence in the lateral (horizontal) patterns shown

where there is no data. The use of the colour scheme should also be reviewed as it is not possible to see the patterns discussed at low relative concentrations in the surface. Also, are the PIC concentrations shown in mmol C m$^{-3}$?

**Thank you for your comment. We will revise the ODV sections to confirm the patterns shown. For fluorescence and Chl-a the colour scheme will be linearized to be consistent with other panels. The $PIC_{Cocco}$ and $PIC_{Total}$ unit will we corrected to µg C L$^{-1}$ (see commentary above).**

Ln 191-192, What does 'over these abundance ranges, coccospheres and detached coccoliths varied in direct proportion' mean? Please report the statistics in the main paper rather than the supplementary material.

**Thank you for pointing this out. We will recreate the plot in R and include the relevant statistics in the main text. The updated sentence will be "Overall, coccosphere abundance tended to vary directly with detached coccolith abundances, although with a high scatter (Fig. S10; y = 13.27x + 6.74, $R^2$=0.51)."**

Ln 195-196, What are figures in the supplementary material (Fig. S5-S6) referenced to support the decline below the euphotic zone in coccosphere and coccolith abundances rather than Fig. 2 in the main text?

**Thank you for the detailed review. We will include as Fig. 2m-n as well in an extended Fig. 2 (the new Fig. 2 is included in response to Reviewer 1).**

Ln 221, Do the authors mean 'coccoliths were estimated to account for 48% of the total PIC' or do they mean coccolithophores? It seems from Fig. 5c that the sum of coccospheres and coccoliths is about 50% of the total PIC concentrations rather than just coccoliths.

**Thank you for your feedback. We will clarify the sentence adding the coccospheres as the all-samples percentage as follows:**

**"On average, coccospheres plus coccoliths were estimated to account for 45-48 % of the $PIC_{Total}$ (Fig. 5c)."**

Ln 237, Section 3.4 – All the examination of coccolithophore and PIC patterns with environmental conditions are in the supplementary material and not the main article. This should be changed as it makes it difficult to see this analysis as part of the main article. Further, Ln 249-250, what do the authors mean by 'phosphate [being] more limiting than nitrate'? Are the relatively high concentrations of nitrate and phosphate likely to be growth limiting to the cell densities of coccolithophores observed? Also, what about light availability – this doesn't seem to be a considered environmental condition in this section.

**Thank you for your feedback. As stated in the introduction, this study focuses on evaluating the relationship between PIC and coccolithophore-produced PIC with oxygen and carbonate chemistry conditions, as summarized in Fig. 7. However, to provide additional context, we included the analysis of these ancillary data in the supplementary material for readers interested in exploring this aspect further.**

**We acknowledge your concern that placing the examination of coccolithophore and PIC patterns with environmental conditions exclusively in the supplementary material may make it harder to see this analysis as part of the main article. In response, we will revise the manuscript to better integrate these findings into the main text while retaining detailed analyses in the supplementary material for thoroughness.**

**Regarding the Referee's concern about nitrate and phosphate limitation, we correct the sentence:**

**"**Higher $PIC_{Cocco}$ showed some tendency to be related to lower phosphate above $Z_{eu}$ (Fig. S13f)**"**

**Unfortunately, stations were sampled at different times of the day, so direct light levels are difficult to interpret. To address this limitation, we defined the euphotic zone depth using a Copernicus product and applied it to distinguish between samples collected above and below the euphotic zone (the CTD included a PAR sensor, and for mid-day the two depths correlated sufficiently for our purposes). However, this means that we do not feel confident in including light levels in this analysis.**

Ln 269-270, Could the higher *Gephyrocapsa* diversity observed in this study relative to the older studies referenced relate to the methods for species analysis in the different studies quoted (i.e. Hendricks et al., 2012, Venrick, 2012). Specifically, would they have been able to differentiate the weaker calcified species of the genus?

**Thank you for pointing this out. Of the two studies, Hendericks et al. (2012) employed methodologies similar to ours, combining cross-polarized light microscopy with scanning electron microscopy and a comparable sampling effort. In contrast, Venrick et al. (2012) utilized inverted microscopy and will therefore be excluded. Instead, we will include the study by Guerreiro et al. (2013), conducted off the coast of Portugal, which employed methodologies and sampling efforts comparable to those in our study.**

Ln 270, The phrasing of 'standing stocks of larger taxa were only noticeable below the euphotic zone' is rather vague and could be interpreted in two ways – either these larger species were absent in the upper ocean and only found at depth, or they made such a small contribution to the total community in the euphotic zone that they were not noticeable. Please rephrase to clarify the meaning.

**Thank you for pointing this out. We understand how the phrasing could lead to ambiguity and have revised the sentence for clarity:**

**"Standing stocks of larger taxa contributed minimally to the total community in the euphotic zone, becoming more prominent below this depth."**

Ln 275, Do the authors mean PIC contributions from other calcifying phytoplankton (such as?) or other plankton (i.e. foraminifera, pteropods). Reference to the recent study by Ziveri et al. (2023) would seem to fit here.

**Thank you for pointing this out. We will incorporate the contribution from other planktonic groups to the PIC pool in the text as follows:**

**"How much coccolithophores contribute to PIC remains an open question, as contributions from calcifying zooplankton (e.g., foraminifera, pteropods; Ziveri et al., 2023), lithogenic sources (Daniels et al., 2012), and processes like fragmentation and dissolution in the water column (e.g., Barrett et al., 2014; Subhas et al., 2022) complicate the relationship between PIC and coccolithophores. These complexities challenge remote sensing algorithms (Balch and Mitchell, 2023) and the use of PIC as a paleoproxy indicator (Beaufort et al., 2011)."**

Ln 283, Daniels et al. (2012) was based in the Bay of Biscay, not coastal waters of Chile. This should be clarified in this use of the citation, currently it could be interpreted as supporting little resuspension of biogenic minerals or river discharge of lithogenic material in coastal waters of Chile. What evidence do the authors have of this negligible input?

**Thank you for your comment. We recognize that Daniels et al. (2012) is based in the Bay of Biscay, not the coastal waters of Chile. It is cited to refer to the Bay of Biscay "or springtime waters off the Bay of Biscay (Daniels et al., 2012)" or as an example of resuspended sediments as sources of PIC "lithogenic sources (Daniels et al., 2012)," Regarding the evidence for negligible input of resuspended biogenic minerals or river discharge of lithogenic material in the coastal waters of north-central Chile, we base this conclusion on the unique oceanographic and geological characteristics of the region. The relatively narrow shelf and the deep, steep topography of the Atacama Trench, located just offshore, prevent significant sediment resuspension from reaching the upper layers, thus limiting the contribution of lithogenic material to the coastal/shelf waters. Furthermore, available studies evaluating the resuspension of sediments and river discharges in coastal Chilean waters focus on shallower regions, often at the river mouth or in nearshore environments (i.e., where pollutants are presents), and not at the broader coastal/shelf scale studied here, where the dynamics differ. We will expand the argument as follows:**

"In the northern Chilean coast, potential lithogenic PIC sources should be negligible. The exceptionally arid Atacama

desert means fluvial inputs are neglible (Thiel et al. 2007) and the deep and steep topography of the Atacama Trench

acts as a major depocenter effectively trapping sediments and limiting their resuspension into the upper water column (Xu et al., 2021). As sources of lithogenic PIC to these coastal waters are limited, we consider most PIC is likely biogenic. **"**

Ln 299, What not put the satellite PIC images into the main article (e.g., as part of Fig. 1)?

**Thank you for your feedback. This concern was also raised by Referee #1, and we propose including this as the new Fig. 3 to enhance the context and address gaps in sampling coverage for PIC.**

Ln 304, What do the authors mean by 'younger populations' in the context used here? Do they mean in terms of stages of bloom growth and senescence when considering coccolith to coccosphere ratios?

**Thank you for your comment. We acknowledge that the term "younger populations" might be confusing without the presence of a bloom. In this context, we are referring to coccolithophore populations in early stages of development or in a more active physiological state, rather than specifically to bloom dynamics. We will moderate the statement to better align with the interpretations in these studies, emphasizing the active physiological state of the populations rather than specifically referring to bloom dynamics as follows:**

**"While these lower ratios might indicate populations in earlier stages of development or more active physiological states (Balch et al., 1991; Holligan et al., 1993b; Lessard et al., 2005), cell-attached coccoliths comprised 51-72% of the PIC$_{Cocco}$ quotas.".**

Ln 306, The use of abbreviations (max.) here is confusing and unnecessary. Please use full word.

**Thank you for pointing this out. We will replace 'max' with 'maximum' throughout the text.**

Ln 310, Suggest adding 'in culture' to 'inhibits the growth of many coccolithophore strains' to emphasise that these are not field observations.

**Thank you for your feedback. We will incorporate this into the text accordingly.**

Ln 325, What about light availability – surely the 1% of surface irradiance at the base of the euphotic zone is an important limitation for growth?

**Thank you for pointing this out. We will update the text as follows:**

"At the base of the euphotic zone, in addition to low light impacting photosynthesis, $O_2$-deficient, acidic subsurface waters show pH and carbonate saturation shift ($\Omega$ calcite) to levels expected by 2100 (below 7.7 and 1.0, respectively) (Fig. 8; also see Vargas et al., 2021). The carbonate chemistry conditions may limit coccolithophore growth and calcification (Meyer and Riebesell, Müller et al. 2016, von Dassow et al. 2018, Kottmeier et al. 2022), and low $O_2$ conditions may also inhibit phytoplankton more generally (Wong et al. 2023). "

Lns 337-339, The estimated PIC concentrations of 120 mmol C m$^{-3}$ from 400 x10$^6$ coccoliths L$^{-1}$ sounds too high. Can the authors check their calculations – using a PIC content of one *G huxleyi* coccolith of 0.025 pmol C (their coccolith value; see Ln 156 and Table S3 converted from PIC to C) only leads to an estimate of 10 mmol C m$^{-3}$. To get to 120 mmol C m$^{-3}$ the authors would need a coccolith C content of 0.3 pmol C coccolith$^{-1}$ (or 3.6 pg C coccolith$^{-1}$), which is much higher than the (carbon values) given by Young and Ziveri (2000).

**Thank you for pointing this out. We have reviewed our calculations and identified that the originally reported value of 120 mmol C m$^{-3}$ was mistakenly expressed in the wrong units. The correct units should be µg C L$^{-1}$, not mmol C m$^{-3}$. We will update the text to reflect the corrected units and ensure consistency throughout the manuscript.**

Ln 369, What depth is meant by 'subsurface' when discussing coccolithophore PIC and detached coccoliths? Euphotic zone or deeper?

**Thank you for pointing this out. This is below the euphotic zone. We will clarify this updating the text as follows:**

**"Below the euphotic zone, coccolithophore PIC is mostly due to detached coccoliths."**

Ln 404-405, Is 'the majority of PIC contributed by coccolithophores is dominated by *G. huxleyi* coccoliths shed during blooms' a conclusion of the present study? What evidence do the authors present for 'blooms' preceding the observations presented?

**Thank you for pointing this out. We will replace 'blooms' by 'high-density events'**

Ln 411, Please reconsider the use of 'max.' in the main text.

**Thank you for pointing this out. We will replace 'max' with 'maximum' throughout the text.**

---

## Author Response (AR1)

**Responses to Referee #1:**

In this study the authors investigate the impact of upwelled low O2/low pH waters on coccolithophore PIC quotas, coccolithophore PIC contribution to the total PIC pool and ultimately what impact this has on the flux of carbon to depth via the PIC:POC ratio in the oxygen minimum zone off central Chile. The general conclusion that this OMZ exhibits comparable PIC concentrations to selected open ocean areas but a greatly reduced PIC:POC ratio in the OMZ core is a nice observation that emerges from the study but the paper needs attention to clarify methodological ambiguities and the occasional lack of precision in reporting results.

**We thank the Reviewer for the supportive comments and the very thoughtful and thorough suggestions and corrections.**
**First of all, we made a refinement to the general conclusion. Re-running the statistical tests separating the surface (euphotic) layer from the subsurface layer, we find that the PIC data overlapped the lower ranges of PIC from other regions and was statistically significantly different from the Southern and Indian Ocean data. So, we conclude: "8. Mixed layer coccolithophore and PIC pools tended to be lower in the OMZ region compared to other open ocean and coastal margin regions of similar productivity (POC), with the exception of the Western Arctic. As a result, mixed layer PIC:POC ratios were significantly lower in the OMZ region compared to all but the Western Arctic.". We hope this makes the message clearer.**
**We respond below point-by-point to the other comments and corrections.**

*Major comments*

Methods: Aspects of the methodologies used need to be clarified. In particular, sections 2.3 and 2.4 are muddled presenting mixed LM and SEM methodologies and unclear reasoning. It is particularly unclear if bulk counting or species/genus level counting was followed consistently and the low number of SEM images examined (with a low magnification) may lead to underestimation of both coccosphere and coccolith counts (see comment below regarding relationship in Figure S4b). Improved description of the methods used is required as this directly leads to ambiguity over how PIC_cocco was estimated (section 2.5) and presented (results). In particular the authors should improve description of the LM and SEM methods used, the apparent bias due to LM counting and the implications of the methodological bias on PIC quotas.

**Thank you for highlighting this important point, which is further elaborated in the minor comments below and has been addressed in the relevant sections. We have improved the corresponding Methods section to make clear that we mostly relied on LM for coccolithophore and coccolith counts, but the more expensive SEM was used in two separate ways.**

**First, in some samples LM counts were not possible, so were replaced with SEM counts, and to make sure the two methods were comparable, we counted a subset of samples with both LM and SEM and the two were statistically indistinguishable.**

Second, we used a higher resolution SEM analysis to evaluate the taxonomic composition. This is important to understand which taxa are contributing to PIC pools. It allows us to get an idea of how estimations of coccolithophore-derived PIC might be improved with better taxonomic resolution.

We also specify more details of both methods as well as how they were used and compared. The new section 2.3 makes this clearer:

**2.3 Coccospheres and detached coccoliths standing stocks**

For enumeration of coccospheres and detached coccoliths, between 0.1 to 1.0 L of seawater (increasing with depth) were filtered onto 25 mm polycarbonate filters with a 0.8 $\mu$m pore size, left to dry at room temperature in Petri dishes, and stored with desiccant until microscopy analyses. Total coccosphere counts were conducted on filter slide preparations with oil immersion, using cross-polarized light microscopy (LM; Zeiss, Axioscope 5). The analysis of 20 fields of view at 400x magnification covered 5.1 mm$^2$ of the filter area, corresponding to a range of 1.9-16.3 mL of seawater analysed. For counts of total detached coccoliths, 11 fields of view per filter were screened (224 x 165 $\mu$m per frame) at 630x magnification (oil immersion objective), covering 0.41 mm$^2$ of the filter area, corresponding to total volumes of 0.2-1.3 mL of seawater analysed. On average, 171 ± 307 coccospheres (range: 2-2099) and 708 ± 664 coccoliths (range: 5-4067) were analyzed per sample. An issue arose where some filters from inshore-offshore 2015 sampling (20º S; Stations T1-T6) exhibited excessive brightness under LM, for which counts were made through scanning electron microscopy (SEM) analysis (Quanta FEG 250) as described in Díaz-Rosas et al. (2021). For the quantification of coccosphere abundances (see equation in Díaz-Rosas et al., 2021), between 28-48 images taken at 800-1500x magnification were examined per filter, covering from 5 to 6 mm$^2$ of the filter area corresponding to a range of 2.1-18.4 mL of seawater analyzed. For total detached coccolith abundances, between 4-5 images were examined per filter, covering from 0.6 to 1.0 mm$^2$ of the filter area corresponding to a range of 0.2-2.8 mL of seawater analyzed. On average, 166 ± 280 coccospheres (range: 1-1141) and 861 ± 879 detached coccoliths (range: 58-3815) were counted per sample. Layers of coccoliths detached from *G. huxleyi* (Fig. S3a-d) were added to the detached-coccolith counts. Collapsed coccospheres were included when they remained mostly intact, but when more disintegrated could not be accurately counted, especially as they were often less reflective than intact coccospheres and coccoliths (Fig. S3e-h). In a subset of samples, collapsed coccospheres were estimated to contribute < 21 % (min. = 0 %, average = 7.1 %) of the total number of coccospheres. As expected, the standard error of the means (among the images obtained from the same sample) drops hyperbolically with the total number of coccospheres or coccoliths counted, whether with LM or SEM (Fig. S4), but remains higher in SEM due to the smaller size of SEM images. To check for differences between counts obtained through LM and SEM examination, five samples with varying coccolithophore abundances were analyzed with SEM as outlined above, revealing the slopes were highly linear with R$^2$ greater than 0.9, and were not significantly different from 1, while the intercepts were close to 0 (Fig. S5), allowing for counts from the two methods to be combined.

We also propose to make similar adjustments to the next section of Methods:

**"2.4 Diversity of coccospheres and detached coccoliths**

**The identification of coccolithophores and detached coccoliths by LM is sometimes limited, which may impact the estimation of coccolithophore-derived PIC relative to total PIC, as it relies on the estimated PIC quotas of coccolith and coccospheres. To understand this effect, taxonomic classification by SEM was performed on samples from the LowpHOx 1 cruise (2015), focusing on samples from T1 to T6 as well as selected samples from L1 (at 5 and 25 m), L2 (at 5 and 50 m) and L3 (at 5 m)..."**

**We believe that similar minor adjustments to the Results as well as figure captions also help make these points clear.**

**We also have converted section 4.2 of the Discussion into a concise consideration of the types of uncertainties related to different methods of estimating PIC and how these are dealt with. Our central interpretation of the results is guided by these considerations to be robust to these uncertainties. Despite the uncertainties, we can conclude that coccolithophores are important contributors to the surface and subsurface PIC pools in this OMZ area. Also, we can be confident in the conclusion that both the surface (euphotic/above oxycline) and subsurface layers of these waters show PIC and PIC:POC ratios that are lower than from most other areas for which comparable PIC and POC data are available.**

POC data: Though PIC:POC ratios are a central aspect to this study the actual POC data is not presented. The absence of the POC data is a curious omission that is not explained and weakens the study. If at all possible this data should be included and not simply alluded to.

**We agree that the POC data is a critical component of our study. We had not presented it directly before as this data is previously published. However, the reviewer has convinced us that it is necessary to present the POC data in the main text, as well as store it along with PIC in the Pangaea repository. In the updated manuscript, we incorporate the POC data into the relevant sections to address this concern.**

**These data, as well as PIC and POC data from other regions used to calculate the PIC:POC ratios, are publicly available and have been appropriately referenced in the manuscript to ensure transparency and accessibility.**

Results: Primarily section 3.1 (but see below). There is a sense that the description of where maximum values were found emphasises Transect T1-T6 and overlooks the broader spatial distributions of parameters along Transect L1-Hyd7, particularly for PIC and coccoliths. Movement of Figure S14 into the main text may help mitigate any uncertainty caused by the unfortunate gap in data along Transect T1-T6 by providing better spatial context and allowing the authors/readers to gauge the representativeness of the in-situ data

(i.e. were key spatial features, such as regions with high PIC offshore, missed by the sampling?).

**We responded to this valuable suggestion by moving the satellite PIC maps (originally Fig. S14) into a figure in the main text, which becomes a new Fig. 4. The new Fig. 4 of satellite imagery has been refined from the original S14, and we add details on data source and processing in Methods Section 2.2:**
**"We produced synoptic maps of satellite-derived PIC in the Southeast Pacific margin (Balch and Mitchell, 2023). Monthly and weekly PIC climatologies (November-December 2015 and January-February 2018) were obtained from the MODIS-Aqua mission (NASA Ocean Biology Processing Group, 2023). The data were then converted from mol $CaCO_3$ m$^{-3}$ to μg C L$^{-1}$ by multiplying by 11910.69 and plotted using RStudio."**

**The *in situ* sampling for PIC and coccolithophores is now situated within the broader spatial framework provided by satellite-derived PIC: In the new Results section 3.3, we now explicitly highlight where enhanced coccolithophore stocks and PIC pools coincided in both satellite and in situ data during the 2015 and 2018 sampling. Additionally, we have expanded the Discussion section 4.2 by adding a paragraph that uses satellite data to assess the effectiveness of our *in situ* sampling in capturing variability in the studied region.**

*Minor comments*

Page 1 Line 23: This statement appears at odds with the authors conclusion that upwelling of nutrients and generation of non-coccolithophore POC may be the cause of the low PIC:POC ratios. The link specifically to low O2/low pH waters is therefore not quite correct.

**We replace these sentences of the Abstract to be clearer:**
**"Our findings support the prediction that in upwelling regions with a shallow oxygen minimum zone, POC production is promoted by phytoplankton other than PIC-producing coccolithophores due to the injection of nutrient-rich but low-pH water. This process decreases PIC:POC ratios, suggesting that the role of PIC in POC sedimentation might be decreased under such conditions. We emphasize that comparing PIC dynamics across diverse upwelling systems will be valuable for understanding how low pH and O$_2$ conditions influence its role in POC fluxes."**

P3 L79: Please specify the Copernicus product used for Kd490

**Thank you for pointing this out. The Copernicus specific product citation is now included in the updated manuscript.**

P3 L83: The definition of the OMZ core was not clear to me and use of the maximum O2 concentration seems counterintuitive when defining the core of the oxygen minimum feature. Please check definition and expand how it was defined.

**We appreciate the Referee's comments and the opportunity to clarify the definition of the OMZ core in our study.**

**We have replaced the actual percentage-based approach with 20 µmol kg$^{-1}$ fixed threshold. Although there is variability in the threshold used in the literature, a threshold of 20 µmol kg$^{-1}$ (or values very close to that in µmol L$^{-1}$) are frequently used. This results in a very minor update of the figures. This is the corrected text in the Methods:**

**"Lastly, the OMZ core was defined as the water layer where dissolved O$_2$ was below 20 µmol kg$^{-1}$, a commonly used threshold (e.g., Gilly et al., 2013), which we use here also to approximate the depth of the base of the upper oxycline."**

P3 L91: Please quantify the magnitude of the applied correction made to PIC measurements for Na residues. It is unclear if this is significant.

**We appreciate the Referee's comment regarding the correction for Na residues in PIC measurements. In our study, Ca from residual seawater was estimated from Na and subtracted. This calculation indicated that residual seawater contributed on average 29.1% ± 25.8% of total Ca in LowpHOx 1 samples and on average 35.2% ± 21.5% of total Ca in LowpHOx 2 samples.**

**As mentioned above, we offer a modified section 4.2 of the Discussion where we consider all the different sources of uncertainty in PIC measurements, whether by chemical or microscopic estimates.**

P3 L92: Typo, presumably this should be 'PIC concentrations' not 'calculations'

**Yes, thank you. We have corrected it in the revised manuscript (Line 102 in the revised manuscript).**

P4 L94/95: The terms 'built' and 'building' seem inappropriate and the intention here is unclear. Please rephrase.

**Thank you for pointing this out! We rephrased it in the revised manuscript (Lines 104-105).**

P4 L96: Direct reference to assessing the effect of this OMZ on POC concentrations suggests that the POC data are central to this study and therefore should be presented alongside the PIC data.

**As mentioned above, we include these data in the revised manuscript.**

P4 L97-99: What criteria were used to define the depth intervals used in this study? As the subsurface interval (5-100 m) presumably crosses the mixed layer there are strong gradients to consider in the distribution of both PIC and coccolithophore diversity which may be lost by the depth bins used.

The original depth bins used in the dataset were 0–5 m, 5–100 m, and 100–500 m. These finer intervals were initially designed to capture vertical gradients, including those within the mixed layer and deeper subsurface layers. However, now we re-binned the data from the global comparison into broader intervals of 0–100 m and 100–400 m. These new intervals were specifically chosen to be comparable with the "above" and "within" categories relative to the OMZ. We recognize that this approach may result in the loss of some fine-scale gradients, particularly in the mixed layer (which typically ranged between 50 and 100 m depth in the other publicly available datasets). Nevertheless, the re-binned depth intervals allow for more simple comparisons of broader ecological patterns across OMZ and non-OMZ regions. We have explicitly outlined the bin choice criteria, and its implications for interpreting PIC, POC, and PIC:POC ratios at the end of Section 2.2:

"These ratios were then categorized into two groups: above and within the OMZ core, to assess the influence of the OMZ on PIC and POC concentrations.
Data from this study were compared against those reported for other open ocean or coastal margins (see Balch et al., 2018). PIC and POC data was obtained from the SEABASS (Werdell et al., 2003) and BCO-DMO repositories (Balch, 2010). Depth intervals were chosen to balance of competing criteria aiming to detect broader ecological patterns robustly. We aimed to be roughly comparable with the categories of above the oxycline (mostly corresponding to the euphotic zone, where coccolithophores are growing) and below the oxycline, where coccolithophores are presumed present entirely due to sinking from the surface. However, OMZ systems are highly stratified, and eukaryotic phytoplankton growth is excluded from below the lower half of the oxycline (which also corresponds to a strong pycnocline) even when sufficient light penetrates for photosynthesis (Wong et al., 2023). In contrast, in non-OMZ systems, the lower limits of growing phytoplankton are less constrained, and coccolithophores are part of the "shade flora" (Balch, 2018). Therefore, data were binned over 0-100 m depths to represent the surface, and over 100-400 depths to represent the subsurface."

Figure 1: The figure legend needs a better description of what the black and grey lines actually represent (the odd placement of the plot legend to the right of panel b was initially overlooked). The black and grey lines need to be better distinguished either by changing the line style or line thickness.

Thank you for pointing this out. We have updated the figure legend and ensure the threshold enclosing the OMZ core are most clearly delineated by using thicker red lines for improved visualization (see below).

[Figure]

**Figure 1: (a) Map of the Southeast Pacific margin showing the study site and stations sampled during late-spring 2015 (circles) and mid-summer 2018 (crosses). (b) Sampling depth coverage for coccolithophores, highlighting areas crossing the OMZ core thresholds of 20 μmol kg⁻¹ (dotted and dashed red lines). Map produced by Ocean Data View (Schlitzer, 2024), with bathymetry based on the GEBCO chart (GEBCO, 2023).**

P5 L116/L126: Figure S4b implies a near constant bias exists between light microscopy and SEM methods when estimating coccolith abundances. As the identity of the two axes in Figure S4 are unclear I am presuming that the SEM counts are on the y-axis (and LM counts on the x-axis) in which case SEM coccolith counts are higher than LM counts, even though the general relationship between the two methods is linear. How significant is this bias and what does it mean for the results of this study? (particularly coccolith counts for stations T1-T6 given the reliance upon SEM; Line 116)

**We fix Fig. S4 (now S5) so that the axes are labelled to make clear which are SEM counts, and which are light microscopy counts. We also now report the standard errors of the slopes and the intercepts. In all cases, the slopes were highly linear with R² greater than 0.9, and were not significantly different from 1, while the intercepts**

**were close to 0.  In the case of detached coccoliths, there was a significant difference in the intercept, but it was very minor and would not result in any significant changes to PIC estimates.**

**We relied on cross-polarized LM counts for almost all absolute counts, because one can cover much more of each sample for a lower cost in time as well as minimizing expensive SEM time.  However, the comparison of LM to SEM counts was only necessary because a small number of samples could not be counted by LM, and we could only count with SEM, due to filters that were very bright in polarized light, for reasons we could not determine.**

**We also include a new analysis of the error from counting effort. For both LM and SEM, we compute the standard error of the mean (SE in %) among different fields of view (images from the same sample) vs the total number of coccospheres or coccoliths counted (considering all fields). As expected, the SE drops hyperbolically with the total number of coccospheres or coccoliths counted. The number or fields counted was constant at 20 and 11 for coccospheres and detached coccoliths in LM and between 28-48 and 4-5 for coccospheres and detached coccoliths in SEM depending on magnification, respectively, allowing us to estimate the expected error of counts within each sample, and we report this error as a percentage in the new Supplementary Figure S4, with actual Fig. S4 becoming S5 (see also lines 143-148 in the revised manuscript).**

P5 L129-133: The approach used to estimate absolute abundances of species/genus coccospheres and coccoliths from SEM images is a little unclear. Based on counts from a low number of images (magnification not reported) I do not understand the rational for multiplying the counts by the total coccosphere or coccolith abundances to obtain total species counts particularly if there are biases between LM and SEM coccolith counts (mainly applicable to stations L1,L2 and L3). Should not the same approach as described in section 2.3 be used i.e. the equation in Diaz-Rosas et al 2021 thereby accounting for volumetric factors? I am concerned that there could be a scaling error here resulting from the use of mixed methodologies. Please clarify.

**While SEM provides higher-resolution imaging, its operational costs are substantial, so total counts (for absolute abundances of total coccolithophores and coccoliths) are more efficiently made by LM. Therefore, the main use of SEM was to obtain relative abundances of the principal taxa with higher resolution, exactly the strategy followed in Diaz-Rosas et al. 2021 (again, a separate protocol for the SEM was used for counting a small number of samples where the filters were too bright under cross-polarized light to permit LM counts – we could not determine the source of that high brightness as they looked the same as other samples under SEM).**

**The principal difference with Diaz-Rosas et al. 2021, where we only analyzed epipelagic samples, was that here we analyzed mesopelagic samples, which are much more dilute in phytoplankton. Even with the maximum volumes of water we could filter before filters clogged, filters of deep samples had much sparser coccospheres and coccoliths compared to surface samples. As a result, we had to be much more careful with SEM time, and this obligated us to work to obtain a minimum**

sample coverage rather than a minimum number of coccospheres analyzed for the SEM analysis. On average, 166 coccospheres per sample were analyzed by SEM, but in some deeper samples, the numbers were much lower (the range was 1 to 1141 coccospheres). When samples from different stations are grouped by depth layer, rarefaction analysis shows that we still had a decent view of total diversity from the deeper surface layers (see new Fig. S14 in the revised supplementary materials, which we copy below for convenience). The main conclusions of this paper are based on estimating the pools of PIC and coccolithophore-derived PIC in the surface and subsurface layers of an OMZ region, because spatial variability in the subsurface waters is more difficult to sample for these reasons.

[Figure]

**Figure S14: Rarefaction-extrapolation analysis for late-spring 2015 and mid-summer 2018, showing species richness (a-b), the exponential of Shannon entropy (c-d), and sample completeness (e-f) for coccospheres and detached coccoliths observed at 2-5, 10, 50, and 100 m. Each curve includes 95 % confidence intervals.**

**Moreover, we have refined the description of how absolute abundances of species/genus were obtained: "To estimate the absolute abundances of coccospheres and detached coccoliths at the species or genus level, the relative abundance of each species or genus was multiplied by the absolute abundance of coccospheres and coccoliths counted by SEM (see section 2.3)."**

P5 L137: There is ambiguity in the methodology over whether species/genus level counts were obtained from SEM images and individual species/genus PIC quotas calculated or whether a size-class based approach was used with a mean size based conversion factor.

**We clarify that the larger spatial and temporal study including samples from both 2015 and 2018 was based on LM analysis, whereas the more focused approach, using only 2015 samples, relied on SEM to better assess taxonomic composition. In particular, this allowed us to determine whether the taxa contributing to PIC pools in the surface layer differed from those contributing to PIC pools in the subsurface OMZ core (refer to section 2.4, lines 150-153 in the revised manuscript).**

Figure 2: This is a complex figure to understand which is made harder by the splitting of this figure across 2 pages. The separation of station names (along top of panel 2e) and the lat/lon (along bottom of panel 2n) is unfortunate and this information really needs to be present on both pages to help the reader. Other issues are the difficulty in seeing the white dashed line (euphotic depth) in all panels. Given the reliance upon POC data, why is there no POC section in Figure 2? Key contours or features cited in text should be more clearly visible.

**Thank you for your feedback. To enhance visualization, we split Fig. 2 into Fig. 2 and Fig. 3. Both figures include latitude/longitude coordinates and station labels. The euphotic zone is made more prominent by using white thicker lines, and additional panels for POC are included to enable direct comparison with PIC. The contours have been simplified to those most discussed in the text. However, we are unable to adjust the contours line width, as the functionality does not appear to be fully supported by Ocean Data View (version 5.8.0).**

P9 L184-196: There are several unclear statements here that can be clarified. For example i) L185 near surface waters <25m vs L192 surface waters 0-30m; ii) L187 highest PICtotal in 2018 was 5.3 mmol/m3 vs L189 highest PIC in 2018 was 5.86 mmol/m3; iii) apparent bias in emphasising max PICtotal in 2018 as being along Line T1-T6 when station Hyd6 had a higher concentration. Overall, this paragraph was a little muddled and unclear and lacking detail or precision in the reporting of results.

**Thank you for your feedback. We have updated the corresponding paragraph to clarify the depth ranges, address the discrepancy in the reported PIC$_{Total}$ values, and highlight the peak in PIC$_{Total}$ observed at station Hyd6 during 2018 (section 3.2, lines 216-236 in the revised manuscript).**

P10 L220: It is not clear where the value of 67% was derived from. I do not see this in Figure 5c?

**We have corrected the text to more clearly compare the PIC$_{Cocco}$ fractions that coccospheres and detached coccolith represents of the total accounted PIC$_{Total}$ within and below 100 m depth, as follows: "In the upper 100 m, detached coccoliths contributed more to PIC$_{Cocco}$ pools than coccospheres, but coccospheres still contributed 37 % of total accounted PIC$_{Total}$ (Fig. 7c). Below 100 m, the contribution of coccospheres was less than 10 % that of detached coccoliths (Fig. 7c)."**

**Here, the original Fig. 5c (now Fig. 7c) compared the contribution of PIC$_{Cocco}$ fractions from coccospheres and detached coccoliths to the total PIC within 100 m depth and across all samples. To align with the subsequent analysis of the subsurface layer, we replaced the 'across all-samples' category with 'below 100 m depth'.**

Figure 3: Panel b gives the impression of monospecific coccolith distributions at many sampled depths due to the approach used of grouping all liths <4um together. This is limitation that needs to be briefly addressed in the discussion.

**The dominance of the detached coccoliths pool by *G. huxleyi* was confirmed during the analysis of lower magnification SEM images and was also confirmed in selected higher magnification images. The inclusion of smaller coccoliths into the <4 μm category reflects this observation, although their contribution is likely negligible, and the category could essentially be referred to as '*G. huxleyi*'. To address this more explicitly, we have added the following sentence to the Results Section 3.4:**

**"Despite the presence of *G. parvula/ericsonii*, its small coccoliths were underrepresented, reinforcing the overwhelming prevalence of *G. huxleyi* in the samples (Fig. 5b)."**

Figure 4: Figure 4a arguably reproduces some of the data contoured in Figure 2k-n, and presented in Figure S5 so the figures could be simplified. Also, I understand PICcocco to be a bulk term representing the total contribution to PIC by both coccosphere and coccolith PIC, thus it seems wrong to present the contribution of spheres and liths to PICcocco in two subplots with axes reaching 100% in both (fig 4b & 4c). How can both axes be correct when panel 4a indicates a changing contribution by both coccospheres and liths to PICcocco? (evident for station T1,T2 but an unclear contribution by coccospheres for T3-T6).

**We appreciate the Referee's concern regarding potential redundancy between the PIC$_{Cocco}$ values presented in Figures 4a (now Fig. 6a), 2k-n (now Fig. 3 l-o) and S5 (now Fig. S7), as well as the decomposition of the PIC$_{Cocco}$ pool in Figures 4b and 4c (now Fig. 6b and 6c). First, it's important to note that the PIC$_{Cocco}$ values in Figures 6b and 6c are based on species/genus-level conversion factors, whereas the PIC$_{Cocco}$ values in Figures 3l-o and S7 are derived using the *G. huxleyi* conversion factor as a maximum threshold. This distinction is outlined at the end of the Methods. Second, we clarify that Figures 6b and 6c presents the relative contributions of coccospheres and detached coccoliths to the PIC$_{Cocco}$ pool, expressed as percentages of the total PIC$_{Cocco}$ pool. These percentages are calculated within the PIC$_{Cocco}$ fraction and do**

**not represent absolute contributions to the PIC$_{Total}$ pool. We have updated the figure caption to reflect this distinction:**

"**Figure 6: Estimated PIC masses from coccospheres and detached coccoliths recorded in waters off Iquique (~ 20° S) during late-spring 2015. (a) Contribution of coccospheres and detached coccoliths to the total PIC$_{Cocco}$ pool. (b) Taxonomic breakdown of the relative contribution of coccospheres to PIC$_{Cocco}$ quotas, expressed as percentages of the total PIC$_{Cocco}$ pool. (c) Taxonomic breakdown of the relative contribution of detached coccoliths to PIC$_{Cocco}$ quotas, expressed as percentages of the total PIC$_{Cocco}$ pool. The PIC$_{Cocco}$ pool estimate is derived from abundances obtained through scanning electron microscopy.**"

**Moreover, we have labeled the horizontal axis of Fig. 6b and 6c with "coccospheres only" and "coccoliths only" to clarify that these represent the contributions of coccospheres and detached coccoliths to the total PIC, accounted for separately by these two components.**

**We also specifically include an analysis of the error that occurs in this region when using only LM vs using SEM to enumerate coccospheres and coccoliths (see Fig. S6 in revised Supplementary Materials):**

"**In a subset of samples from T1-T6 in 2015 analysed by higher magnification SEM for better taxonomic resolution, PIC$_{Cocco}$ estimations using taxa-specific conversions generally showed little differences with PIC$_{Cocco}$ estimations made with the assumption that all coccospheres and coccoliths had PIC quotas similar to *G. huxleyi* (Fig. S6a). Substantial differences were only seen at depths of 5 m at station T1 and 300 m at station T5 (Fig. S6b, f), related to the high contribution of *C. leptoporus* and *Helicosphaera* spp. to PIC$_{Cocco}$ at those stations.**"

P13 L235: Typo in legend of figure 5 (concentsdaration)

**Corrected in the updated manuscript.**

P13 L238: The phrase 'marginally higher' is ambiguous without a quantified value or statistical support. Is the difference significant?

**We are now clearer. There were no statistical differences between the coccospheres, detached coccoliths, PIC$_{Total}$, and PIC$_{Cocco}$ across the 2015 and 2018 cruises although peaks are notable in 2018. We have revised the text as follows (lines 307-309):**

"**Coccospheres, detached coccoliths, PIC, and the estimated PIC$_{Cocco}$ stocks were not statistically different ($p < 0.05$) between the 2015 and 2018 cruises, although mid-summer 2018 exhibited higher average and maximal values compared to late-spring 2015 (Fig. 8).**"

P13 L245: Typo, panel 2c,g ? not 2h?

**Thank you for your comment. We have double-checked these panels references are accurate as per the panels shown.**

P13 L249: Typo, panel 2e-f, not 2f-g?

**We appreciate your careful attention. The missing "j" for pCO$_2$ panel was added in the updated manuscript.**

P15 L82: Can remove approximation by stating actual results (45-48%)

**We appreciate the suggestion and agree that providing exact values can enhance clarity. However, since these values are explicitly detailed in the Results section, we believe that rounding to approximate percentages in the Discussion helps to convey a more concise and accessible message. This approach aligns with the purpose of the Discussion section, which is to synthetize the key findings.**

P16 L303: Would be useful to state the ratios from Balch et al 1991, Holligan et al 1993b that were used in this comparison.

**We agree. We have included a detached-coccolith-to-coccosphere ratio of > 250, which encompasses values from both studies.**

P16 L303/L306/L343/L411: No need to abbreviate maximum to max.

**We have replaced 'max.' with 'maximum' throughout the text.**

P16 L305: It is not clear where the stated values of cell-attached coccolith contribution to PICcocco (51-72%) come from. Please clarify and highlight in the results.

**Thank you for raising this point. To improve clarity, we now present a histogram of the coccolith-to-coccosphere ratios, as well as scatterplots comparing this ratio with coccospheres and PIC$_{Cocco}$ (see new panels of Fig. S11). This information then supports a clearer discussion:**

**"Coccospheres accounted for an average of 37 % of PIC$_{Cocco}$ in the surface mixed layer (Fig. 7c), and the ratios of detached coccoliths to coccospheres were mostly below 40 (median: 31), with the highest coccosphere abundances and PIC$_{Cocco}$ values associated to relatively low detached coccospheres (Fig. S11c-d), in contrast to the greater excess of detached-coccolith over coccosphere ratios reported for massive blooms (> 250; e.g., Balch et al., 1991; Holligan et al., 1993b). Lower ratios might indicate populations in active physiological states rather than cells entering decreased growth as maximum bloom densities are reached and cell growth slows or ceases (Balch et al., 1991; Holligan et al., 1993b; Lessard et al., 2005)."**

P17 L332: Missing appropriate references (for Calcite Belt, Bay of Biscay)

**We have added the references for the studies conducted in the Calcite Belt (Balch et al., 2018) and the Bay of Biscay (Daniels et al., 2012), lines 440-441 of present manuscript.**

P19 L373: From Figure 5c I do not see how the statement that up to two-thirds of the PIC_cococ quota comes from detached coccoliths can be correct? Please clarify

**Thank you for your concern, which is similar to a previous point raised. The value provided represents an approximation of the percentage of the total accounted $PIC_{Total}$ (48% and 30%) contributed by the coccospheres (18% and 2%) and detached coccolith fractions (30% and 28%) within and below 100 m depth layers. Thus, for detached coccoliths 30 x 100 / 48 = 63% and 28 x 100 / 30 = 93%, which accounts for almost all of the total $PIC_{Cocco}$. We have updated the Results (lines 280-283): "On average, coccospheres plus detached coccoliths were estimated to account for 30-48 % of the $PIC_{Total}$ (Fig. 7c). In the upper 100 m, detached coccoliths contributed more to $PIC_{Cocco}$ pools than coccospheres, but coccospheres still contributed 37 % of total accounted $PIC_{Total}$ (Fig. 7c). Below 100 m, the contribution of coccospheres was less than 10 % that of detached coccoliths (Fig. 7c)." Also the Discussion (lines 478-479): "Below the $Z_{eu}$, coccolithophore PIC was predominantly composed of detached coccoliths, which accounted for almost all of the total $PIC_{Cocco}$ (Fig. 7c),…"**

P20 L388: The observation that the PIC:POC ratio (Figure 8c) is greatly reduced compared to other areas is intriguing despite the comparable PIC standing stocks (Figure 8b). Without more detail on the coincident POC dataset however it is difficult to rationalise this observation beyond the suggestion put forward by the authors that upwelling stimulates non-calcareous phytoplankton. For this reason, the authors should consider including the POC dataset in this study. It may be particularly important to ascertain the similarity or differences in POC concentrations between the various studies/sites used for comparison to validate the conclusions reached. Also, the PIC:POC results appear most comparable to results from the W. Arctic, which is not an upwelling zone. This point needs to be highlighted. What could be the cause of this similarity?

**We appreciate your concern regarding the accuracy of the PIC:POC ratio analysis. During the preparation of the POC data for plotting, we identified an error presenting the PIC values. Specifically, the PIC values were incorrectly expressed in mmol C m$^{-3}$ instead of µg C L$^{-1}$, resulting in disproportionately high PIC levels for these samples. This issue has now been corrected, and the POC data have also been added. We present an updated version of Figure 8 (now Fig. 10), which includes the corrected PIC data, the newly added POC data, and revised binning of the external datasets (see below). The findings derived from Figure 10 has been updated in the respective sections of the manuscript.
We have divided the Discussion sections to discuss separately coccoliths, coccospheres, PIC pools, and PIC:POC ratios in the surface mixed layer in one sub-section, and in the subsurface layer in the next sub-section, and in both sub-sections we specifically mention the Western Arctic. Coccolithophores still penetrate polar waters much less than temperate and tropical waters, so this is probably a major reason for the lower values in the Arctic.**

[Figure]

**Figure 10: (a) Global map showing the annual oxygen content at 100 m depth and the sampling locations for the well-mixed surface (0-100 m; panels b, c. d) and the stable subsurface (100-400 m; panels e, f, g), along with the surface-mixed and OMZ-core layers (panels h, i, j). The**

**PIC$_{Total}$, POC, and PIC:POC ratio values during late-spring 2015 and mid-summer 2018 in the SE Pacific OMZ (this study), as well as in other open ocean and coastal margin regions, are shown (data from Balch et al., 2018). The Atlantic Ocean dataset includes samples from six cruises (AMT17-22). One-way ANOVA results indicate significant differences ($p < 0.05$) in PIC$_{Total}$, POC, and PIC:POC ratios among the OMZ, Atlantic Ocean, Southern Ocean, Indian Ocean, Western Arctic, and Patagonian Shelf, as well as between OMZ layer sample groups based on Tukey post-hoc comparisons, represented by lowercase letters above each boxplot. Only data points with both PIC and POC available were included in the analysis. The map was generated using Ocean Data View (Schlitzer, 2024), with annual oxygen climatology from the World Ocean Atlas 2018 (Boyer et al., 2018; García et al., 2018).**

P22 L413: Conclusion 6 seems to contradict statements on P19 L379. On P19 is the statement that the PIC:POC ratio was significantly elevated in the OMZ core (due to a ballast effect), whereas on L413 the (relatively) low PIC:POC ratio of the OMZ core is highlighted for its difference to other coastal margin areas. Whilst the overall conclusion that this OMZ exhibits lower PIC:POC ratios compared to other locations is valid, it also seems that when examined in detail the core of this OMZ is associated with elevated PIC:POC ratios (Figure 8c), thus the broader significance of this could be addressed in section 4.4.

**Thank you for your feedback. We have updated conclusions. Now conclusions 8 and 9 are:**
**"8. Mixed layer coccolithophore and PIC pools tended to be lower in the OMZ region compared to other open ocean and coastal margin regions of similar productivity (POC), with the exception of the Western Arctic. As a result, mixed layer PIC:POC ratios were significantly lower in the OMZ region compared to all but the Western Atlantic.**

**9. Subsurface PIC and PIC:POC ratios were also lower than other regions for which data were available, with the exception of the Western Arctic. PIC:POC increases from the surface mixed layer to the OMZ core, but the increase is lower compared to the rest of the ocean. These considerations suggest that, in OMZ regions, PIC plays a diminished role as ballast to drive POC fluxes to the deep."**

Table S1: Are the units for PIC concentration incorrect? (i.e. uM not mM?)

**Yes, thank you. We have corrected the units to µg C L$^{-1}$.**

Figure S10: The SEM images have reproduced poorly in my copy. Maybe upload the SEM image files separately to allow greater accessibility. Missing word in Figure legend.

**All scanning electron microscopy images used in this study have been stored in the Zenodo public repository. This dataset is referenced as:**

**Díaz-Rosas, F., Vargas, C. A., and von Dassow, P.: Scanning Electron Microscopy Datasets – Coccospheres and detached coccoliths in waters off the Southeast Pacific margin, https://doi.org/10.5281/zenodo.14048319, 2024.**

**It is publicly available and so we consider it most efficient to share these images by that method. The legend has also been corrected.**

Figure S11: These LM images also reproduced poorly. Maybe upload the image files as well.

**All cross-polarized light microscopy images shown in what is now Fig. S13 have been stored in the Zenodo public repository. This dataset is referenced as:**

**Díaz-Rosas, F.: Cross-polarized light microscopy images – Coccospheres and detached coccoliths in waters off the Southeast Pacific margin, https://doi.org/10.5281/zenodo.14708540, 2025.**

Figure S14: I find this figure to be a useful means of assessing the spatial variability of PIC during both cruise periods. I would encourage the authors to consider moving this figure into the main text as it provides useful context.

**Thank you for the feedback, which has been addressed in the major comment above.**

**Responses to Referee #2:**

The article by Díaz-Rosas et al. presents an interesting examination of coccolithophore impacts on PIC standing stocks in relation to upwelling and the oxygen minimum off the coast of Chile. The authors compare coccolithophore based estimates of PIC and discrete PIC chemical measurements and show that only 50% of the PIC is sourced from coccolithophores in these waters. There is also a strong gradient in species contributions which align with the base of the euphotic zone and upper part of the oxygen minimum zone. The authors explore these observations in terms of the context of the fate of sinking organic carbon and how reduced PIC ballasting of sinking organic carbon could strengthen low oxygen conditions.

The article is well written and contains interesting observations and measurements, however several issues need to be further clarified. Firstly, the link between PIC production, ocean acidification (pH) and PIC standing stocks, sinking POC ballasting by PIC, and the oxygen minimum zone needs to be better explained in the introduction – at the moment there is no mention of PIC ballasting so that the link between pH conditions and $O_2$ concentrations is not clear. Secondly, a lot of material is found in the supplementary material rather than the main article and this makes following the article difficult – this is especially in the case where the authors compare their coccolithophore and PIC dynamics with environmental conditions. This is wholly in the supplementary material and if the authors consider this an important element of the paper, this statistical analysis should be in the main article. Lastly, can the authors confirm that their conversions from coccolith $CaCO_3$ (their Table S3) to the values reported in the article (mmol C m$^{-3}$) took into account the molecular weight of $CaCO_3$ – for most of the article this appears to be so, but then there is a calculation in the discussion which gives very high PIC concentrations (Ln 339) and its not clear how these values were achieved.

**We thank Dr. Poulton for the supportive and careful comments and suggestions. We confirm that we checked the conversions and corrected an error in one part. We respond to the detailed comments below.**

SPECIFIC COMMENTS

Ln 12, Consider whether the term 'pools' or quotas or standing stocks (as used elsewhere in the paper) would be better.

**We appreciate your concern about the consistency of pools, quotas, and standing stocks terminology and offer to clarify them according to the following criteria:**

**- "Pools" is used to describe integrated or total quantities, such as the cumulative PIC across a specific depth range or area, which is essential for understanding large-scale patterns. For example: "Below the Zeu, the sharp decline in the numerical abundances of coccospheres (< 0.1 x10$^5$ L$^{-1}$) and detached coccoliths (< 10 x10$^5$ L$^{-1}$) decreased coccolithophore PIC pools to below 1 μg L$^{-1}$." or "In summary, our results emphasize the importance of coccolithophores as contributors to total PIC pools in this OMZ.".**

- Quotas refer to per-cell metrics, such as PIC per coccolithophore cell, which provide insights into species-specific contributions to the PIC pools. For example, "The PIC quotas of individual coccoliths and coccospheres were estimated following Young and Ziveri (2000), and used to calculate coccolithophore-derived PIC stocks from abundances of coccospheres and detached coccoliths."

- The term "Standing stocks" is used in the sense of organism stocks (such as fish stock assessments). Here it is employed to capture instantaneous measurements of total coccolithophore abundances at specific depths or times, helping to characterize temporal and spatial variability. For example: "Nevertheless, maximum standing coccolithophore stocks were still about an order of magnitude lower than the typical bloom abundances reported in other regions (Tyrrell and Merico, 2004)".

Each of these terms corresponds to a distinct analytical perspective reported in the manuscript, and using only one term would oversimplify the multifaceted nature of our findings. We have corrected the manuscript to use them consistently in the appropriate contexts.  For example, the header of the second section of Results was corrected to:

**"3.2 Standing stocks of coccolithophores, pools of PIC and POC, and PIC:POC ratios"**

Lns 12-13, The links between ocean acidification, PIC concentrations and production, the efficient ballasting of sinking POC, and the impact of remineralisation depth need to be further expanded on in the first couple of lines of the abstract; the links are not obvious without further explanation.

**Thank you for pointing this out. We updated the first lines of the Abstract as follow:**

"A predicted consequence of ocean acidification is decreasing coccolithophore-produced Particulate Inorganic Carbon (PIC) pools. PIC is thought to enhance the sinking of Particulate Organic Carbon (POC) to deeper waters, potentially influencing the depth of organic matter remineralization and subsurface oxygen levels."

Ln 17, What do the authors mean by 'countable coccoliths'? Do they mean detached coccoliths, or are they hinting at a portion of coccoliths that is not countable?

**Thank you for pointing out the ambiguity of 'countable coccoliths'. It refers specifically to coccospheres and coccoliths observed and enumerated under the microscope. This excludes any coccoliths that were too small, fragmented, or otherwise indistinguishable for reliable counting. The term does not imply the existence of a portion of coccoliths that is inherently uncountable but rather reflects the limitations of our counting method. We have updated the sentence as follows:**

"On average, about half of the PIC was attributed to reliably enumerated coccospheres and detached coccoliths, with significantly diminished pools below the euphotic zone."

Ln 27, What about the important contributions from other calcifying plankton? As the paper concludes that coccolithophore PIC is only around 50% of PIC standing stocks, would it not be good to expand on other potential sources of biological PIC?

**We appreciate the suggestion to expand on other potential sources of biological PIC. We fully agree that other calcifiers play significant roles in the biological PIC pool. In response to another comment, we have now acknowledged the roles of other calcifying plankton in the revised Discussion:**

**"How much coccolithophores contribute to PIC remains an open question, as contributions from calcifying zooplankton (e.g., foraminifera, pteropods; Ziveri et al., 2023), lithogenic sources (Daniels et al., 2012), and processes like fragmentation and dissolution in the water column (Barrett et al., 2014; Subhas et al., 2022) complicate the relationship between PIC and coccolithophores." and "In addition to PIC production by other planktonic organisms, which would not be detected by the microscopy protocols used here,..."**

Ln 34, There is no mention of the ballast effect on POC sinking in the first paragraph of the introduction so that the link between reduced PIC production and the 'favouring of the respiration of more organic material' is not obvious. Suggest adding introduction of the ballast effect earlier in the introduction to make this link obvious.

**Thank you for your feedback. We have updated the first part of the Introduction to explicitly describe the potential role of coccoliths in the sinking process or 'ballast effect' as follows:**

**"The Particulate Inorganic Carbon (PIC) pool is a key component of marine carbon cycles and atmospheric reservoirs (Ridgwell and Zeebe, 2005). It originates from various sources, including land-derived inputs to coastal margins (Cai, 2011) and biological processes such as phytoplankton calcification (Taylor and Brownlee, 2016). Coccolithophores, particularly the cosmopolitan species *Gephyrocapsa (Emiliania) huxleyi*, significantly contribute to PIC through coccolith production. Coccoliths are thought to enhance the sinking of organic matter (i.e., the 'ballast effect'), facilitating the export of Particulate Organic Carbon (POC) to deeper waters (e.g., Klaas and Archer, 2002), thereby influencing carbon sequestration and nutrient cycling in the global pelagic ocean (Monteiro et al., 2016; Balch, 2018). Changes in PIC dynamics, driven by shifts in coccolithophore communities across spatial and temporal scales, feedback into the ocean-atmosphere system (Balch et al., 2016; Claxton et al., 2022)."**

Ln 43, Consider the use of the term 'coccolithophore bloom' in the context of the references given – Beaufort et al. (2008) is based in the subtropical S Pacific waters and did no observe what many would consider a 'bloom'.

**Thank you for pointing this out. You're right, and we have replaced 'bloom' with 'high-density events'. We have also revised its use throughout the manuscript.**

Ln 92 and 105, Can the authors please confirm that all the PIC concentrations presented in the paper are in mmol C m$^{-3}$ and that their conversions from the values in Table S3 have taken into account the molecular weight of calcite? Its not obvious that this is a problem or that a mistake has been made but a calculation in the discussion (Lns 337-339) gives much higher PIC concentrations that expected in units of mmol C m$^{-3}$.

**Thank you for your feedback. You are correct that there was an erroneous use of units, where mmol C m$^{-3}$ corresponds to µg C L$^{-1}$. We have corrected the units in the text, figures and tables. This adjustment refines the overall narrative, particularly in relation to Fig. 8 (now Fig. 10) and Table 1, as it impacts the PIC and PIC:POC ratio comparisons with other studies or sampling sites. The updated findings have been incorporated accordingly.**

Ln 102, Fig. 1 – What are the solid lines on (b) and (c)? As these lines do not line up with the discrete samples, could the legend explain more?

**Thank you for your feedback. We agree that these lines were not appropriately presented to highlight areas crossing the thresholds enclosing the OMZ core. For clarity, we have updated the figure legend and ensure the OMZ core threshold is more clearly delineated by using thicker red lines for improved visualization. See response to Reviewer 1 where we include a corrected figure.**

Lns 112 and 118, Having given the proportion of volume counted from the filters for SEM and light microscope analysis (<20 mL or <2% of the volume filtered), it is a surprise that no comment is made on the potential impact this may have on the observations made – specifically the species diversity and the counts of large, often numerically rare species.

**Thank you for pointing this out. While SEM provides high-resolution imaging, its operational costs are substantial. To optimize resources, filters were scanned at lower magnifications (800x-1500x) with sufficient resolution to allow for post-scan zooming, enabling the detection of both coccospheres and detached coccoliths. For smaller detached coccoliths, we assumed their origin to be from the G. parvula/ericsonii assemblage, based on prior studies in the same area (e.g., see Beaufort et al., 2008; von Dassow et al., 2018). The dominance of *G. huxleyi* coccospheres in absolute coccosphere counts allowed us to reduce SEM image re-analysis for species/genus classification by approximately 25%, while maintaining the same effort for detached coccolith counts.**

**Please see our detailed comments to Reviewer 1 about these points. Also, we provide below rarefaction curves for 2015 and 2018 samples showing how we expect that we effectively captured rare species, especially when different samples are grouped by layers (new Fig. S14). As on average over 100 coccospheres were counted per sample, we should have been able to capture rare species in all but the most dilute (deepest) samples. These new inputs are presented in Results and Discussion sections accordingly.**

[Figure]

**Figure S14: Rarefaction-extrapolation analysis for late-spring 2015 and mid-summer 2018, showing species richness (a-b), the exponential of Shannon entropy (c-d), and sample completeness (e-f) for coccospheres and detached coccoliths observed at 2-5, 10, 50, and 100 m. Each curve includes 95 % confidence intervals.**

Ln 126, What is meant by a 'good fit'? The statistics of the relationship should be presented in the main article rather than the supplementary material.

**Thank you for pointing this out. This aligns closely with Referee 1 concern regarding the potential bias introduced by the use of different microscopic methods. To address this concern, we have present a revised version of Fig. S4 (now Fig. S5), specifying the counting methods used on each axis (abbreviated as SEM and LM), along with a more detailed statistical analysis to assess the agreement between them.**

Ln 162, What is the oxycline? This needs to be defined somewhere as to what the authors consider either the absolute threshold, relative change or gradient of this feature.

**Thank you for your comment. We now define the OMZ core based on a 20 µmol kg$^{-1}$ threshold, which we also use to approximate the depth of the base of the upper oxycline.**

Ln 165, Are the authors referring to fluorescence or Chl-*a* peaks when discussing 'peaks of phytoplankton'? Please clarify.

**Thank you for your comment. We have included a threshold for higher fluorescence and Chl-a levels and clarify that we are referring to both proxies for phytoplankton biomass.**

Ln 166, In Fig. 2f the surface (<25 m) nitrate concentrations look much lower (<5 µM) than the 10 µM quoted in this line. As 10 µM is not generally considered 'low' in terms of biologically-limiting, this line should be changed to better reflect (biologically) 'low' concentrations (e.g., <2 µM).

**Thank you for your feedback. We have adjusted the nitrate according to a biologically relevant threshold (Line 195 of the updated manuscript).**

Ln 176, Fig. 2 – ODV is notorious for problems with plotting discontinuous (discrete) data in that it often 'creates' patterns not supported by the data. The authors should review the patterns shown in the patchier data (e.g., fluorescence, Chl-*a*, PIC, coccospheres, coccoliths) as to whether they are confidence in the lateral (horizontal) patterns shown where there is no data. The use of the colour scheme should also be reviewed as it is not possible to see the patterns discussed at low relative concentrations in the surface. Also, are the PIC concentrations shown in mmol C m$^{-3}$?

**Thank you for your comment. We have revised the ODV sections and confirm the patterns shown, except for excessive interpolation around 30° S in 2018, which has been removed to reflect the absence of data (see new Fig. 3). For fluorescence and Chl-a, the colour scheme was linearized to ensure consistency with other panels. The units for PIC$_{Cocco}$ and PIC$_{Total}$ were corrected to µg C L$^{-1}$ (see commentary above).**

Ln 191-192, What does 'over these abundance ranges, coccospheres and detached coccoliths varied in direct proportion' mean? Please report the statistics in the main paper rather than the supplementary material.

**Thank you for pointing this out. We have redone the plot in R and updated the sentence as follows:**

**"Overall, coccosphere abundance tended to vary directly with detached coccolith abundance, although with a high scatter ($y = 13.270(\pm1.123)x + 6.740(\pm0.874)$; $R^2_{adjusted}= 0.51$; Fig. S11a)."**

Ln 195-196, What are figures in the supplementary material (Fig. S5-S6) referenced to support the decline below the euphotic zone in coccosphere and coccolith abundances rather than Fig. 2 in the main text?

**Thank you for the detailed review. We have included reference to Fig. 3m-n as well (please note that a new Fig. 3 is included in response to Reviewer 1).**

Ln 221, Do the authors mean 'coccoliths were estimated to account for 48% of the total PIC' or do they mean coccolithophores? It seems from Fig. 5c that the sum of coccospheres and coccoliths is about 50% of the total PIC concentrations rather than just coccoliths.

**Thank you for your feedback. We have modified panel c of Fig. 5 (now Fig. 7c) to assess the contributions of coccosphere and detached coccolith between the surface (0-100 m depth) and subsurface layers (below 100 m depth), and have clarified the sentence accordingly as follows:**

**"On average, coccospheres plus detached coccoliths were estimated to account for 30-48 % of the $PIC_{Total}$ (Fig. 7c). In the upper 100 m, detached coccoliths contributed more to $PIC_{Cocco}$ pools than coccospheres, but coccospheres still contributed 37 % of total accounted $PIC_{Total}$ (Fig. 7c). Below 100 m, the contribution of coccospheres was less than 10 % that of detached coccoliths (Fig. 7c)."**

Ln 237, Section 3.4 – All the examination of coccolithophore and PIC patterns with environmental conditions are in the supplementary material and not the main article. This should be changed as it makes it difficult to see this analysis as part of the main article. Further, Ln 249-250, what do the authors mean by 'phosphate [being] more limiting than nitrate'? Are the relatively high concentrations of nitrate and phosphate likely to be growth limiting to the cell densities of coccolithophores observed? Also, what about light availability – this doesn't seem to be a considered environmental condition in this section.

**Thank you for your feedback. As stated in the introduction, this study focuses on evaluating the relationship between PIC and coccolithophore-produced PIC with oxygen and carbonate chemistry conditions, as summarized in Fig. 7 (now Fig. 9). However, to provide additional context, we included the analysis of these ancillary data in the Supplementary Material for readers interested in exploring this aspect further.**

**We acknowledge your concern that placing the examination of coccolithophore and PIC patterns with environmental conditions exclusively in the supplementary material may make it harder to see this analysis as part of the main article. In response, we have revised the manuscript to better integrate these findings into the main text (see lines 407-408) while retaining detailed analyses in the supplementary material for thoroughness.**

**Regarding the Referee's concern about nitrate and phosphate limitation statement, we did not measure nutrient limitation, so this was removed.**

**Unfortunately, stations were sampled at different times of the day, so direct light levels are difficult to interpret. To address this limitation, we defined the euphotic zone depth using a Copernicus product and applied it to distinguish between samples collected above and below the euphotic zone (the CTD included a PAR sensor, and for mid-day the two depths correlated sufficiently for our purposes). However, this means that we do not feel confident in including light levels in this analysis.**

Ln 269-270, Could the higher *Gephyrocapsa* diversity observed in this study relative to the older studies referenced relate to the methods for species analysis in the different studies quoted (i.e. Hendricks et al., 2012, Venrick, 2012). Specifically, would they have been able to differentiate the weaker calcified species of the genus?

**Thank you for pointing this out. Of the two studies, Hendericks et al. (2012) employed methodologies similar to ours, combining cross-polarized light microscopy with scanning electron microscopy and a comparable sampling effort. In contrast, Venrick et al. (2012) used inverted microscopy and was therefore excluded. Instead, we have included the study by Guerreiro et al. (2013), conducted off the coast of Portugal, as it employed methodologies and sampling efforts comparable to ours.**

Ln 270, The phrasing of 'standing stocks of larger taxa were only noticeable below the euphotic zone' is rather vague and could be interpreted in two ways – either these larger species were absent in the upper ocean and only found at depth, or they made such a small contribution to the total community in the euphotic zone that they were not noticeable. Please rephrase to clarify the meaning.

**Thank you for pointing this out. We understand how the phrasing could lead to ambiguity and have revised the sentence for clarity:**
**"Standing stocks of larger taxa contributed minimally to the total community in the $Z_{eu}$ (~50 m depth), becoming more prominent below this depth."**
Ln 275, Do the authors mean PIC contributions from other calcifying phytoplankton (such as?) or other plankton (i.e. foraminifera, pteropods). Reference to the recent study by Ziveri et al. (2023) would seem to fit here.

**Thank you for pointing this out. We have incorporated the potential contribution from other planktonic groups to the PIC pool in the Discussion 4.2 section (Lines 344-345).**

Ln 283, Daniels et al. (2012) was based in the Bay of Biscay, not coastal waters of Chile. This should be clarified in this use of the citation, currently it could be interpreted as supporting little resuspension of biogenic minerals or river discharge of lithogenic material in coastal waters of Chile. What evidence do the authors have of this negligible input?

**Thank you for your comment. We recognize that Daniels et al. (2012) is based in the Bay of Biscay, not the coastal waters of Chile. It is cited to refer to the Bay of Biscay or as an example of resuspended sediments as potential lithogenic sources to PIC. Regarding the negligible input of resuspended biogenic minerals or river discharge of lithogenic material in the coastal waters of north-central Chile, we base this**

conclusion on the unique oceanographic and geological characteristics of the region. The relatively narrow shelf and the deep, steep topography of the Atacama Trench, located just ~100 km offshore, prevent significant sediment resuspension from reaching the upper layers, thereby limiting the contribution of lithogenic material to coastal and shelf waters. Furthermore, available studies on sediment resuspension and river discharge in coastal Chilean waters primarily focus on shallower regions, often near river mouths or in nearshore environments (i.e., where pollutants are presents), rather than at the broader coastal/shelf scale examined in our study, where different dynamics prevail. We have expanded the argument as follows:

"In the northern Chilean coast, potential lithogenic PIC sources should be negligible. The exceptionally arid Atacama Desert means fluvial inputs are negligible (Thiel et al., 2007) and the deep and steep topography of the Atacama Trench acts as a major depocenter effectively trapping sediments and limiting their resuspension into the upper water column (Xu et al., 2021). As sources of lithogenic PIC to these waters are limited, we consider most PIC is likely biogenic."

Ln 299, What not put the satellite PIC images into the main article (e.g., as part of Fig. 1)?

**Thank you for your feedback. This concern was also raised by Referee 1, and we have incorporated it as the new Fig. 4 to provide better context and address gaps in PIC sampling coverage.**

Ln 304, What do the authors mean by 'younger populations' in the context used here? Do they mean in terms of stages of bloom growth and senescence when considering coccolith to coccosphere ratios?

**Thank you for your comment. We acknowledge that the term 'younger populations' may be confusing in the absence of a bloom. In this context, we are referring to coccolithophore populations in active physiological states, rather than specifically to bloom dynamics. We have adjust the statement to better align with the interpretations in these studies, as follows:**

"Lower ratios might indicate populations in active physiological states rather than cells entering decreased growth as maximum bloom densities are reached and cell growth slows or ceases (Balch et al., 1991; Holligan et al., 1993b; Lessard et al., 2005)."

Ln 306, The use of abbreviations (max.) here is confusing and unnecessary. Please use full word.

**Thank you for pointing this out. We have replaced 'max.' with 'maximum' throughout the text.**

Ln 310, Suggest adding 'in culture' to 'inhibits the growth of many coccolithophore strains' to emphasise that these are not field observations.

**Thank you for your feedback. We have incorporated this into the text (Line 431).**

Ln 325, What about light availability – surely the 1% of surface irradiance at the base of the euphotic zone is an important limitation for growth?

**Thank you for your comment. We note this point was raised earlier in your review, and we have already addressed it. Please refer to the section above for our detailed explanation.**

Lns 337-339, The estimated PIC concentrations of 120 mmol C m$^{-3}$ from 400 x10$^6$ coccoliths L$^{-1}$ sounds too high. Can the authors check their calculations – using a PIC content of one *G huxleyi* coccolith of 0.025 pmol C (their coccolith value; see Ln 156 and Table S3 converted from PIC to C) only leads to an estimate of 10 mmol C m$^{-3}$. To get to 120 mmol C m$^{-3}$ the authors would need a coccolith C content of 0.3 pmol C coccolith$^{-1}$ (or 3.6 pg C coccolith$^{-1}$), which is much higher than the (carbon values) given by Young and Ziveri (2000).

**Thank you for pointing this out. We have reviewed our calculations and identified that the originally reported value of 120 mmol C m$^{-3}$ was mistakenly expressed in the wrong units. The correct units should be µg C L$^{-1}$, not mmol C m$^{-3}$. We have updated the text to reflect the corrected units and ensure consistency throughout the manuscript.**

Ln 369, What depth is meant by 'subsurface' when discussing coccolithophore PIC and detached coccoliths? Euphotic zone or deeper?

**Thank you for pointing this out. This is below the euphotic zone. We have clarified this updating the text as follows:**
**"Below the $Z_{eu}$, coccolithophore PIC was predominantly composed of detached coccoliths,..."**
Ln 404-405, Is 'the majority of PIC contributed by coccolithophores is dominated by *G. huxleyi* coccoliths shed during blooms' a conclusion of the present study? What evidence do the authors present for 'blooms' preceding the observations presented?

**Thank you for pointing this out. We have replaced 'blooms' by 'high-density events'**

Ln 411, Please reconsider the use of 'max.' in the main text.

**Thank you for pointing this out. We have replaced 'max.' with 'maximum' throughout the text.**

---

## Referee Report (RR1)

The topic of the presented manuscript is relevant and the data presented seems to be of good quality. I see potential for a valuable contribution to the field.

However, in my opinion, the manuscript is not well written. The storyline and take-home messages are lost in detail and the analysis lacks streamlining. I could therefore not follow throughout much of the results and discussion. This made it difficult if not impossible for me to judge the scientific soundness and conceptual value of the content presented. While I am not an expert on coccolithophores, I have worked on upwelling systems, phytoplankton and export. Someone like me should be able to understand.

In conclusion, I have to recommend rejection of the manuscript in its current form. In my opinion, a complete re-work of the manuscript is needed, prioritizing storyline and conciseness. If done successfully, the manuscript may be reconsidered. As I said, I am not questioning the quality or significance of the data.

**1. The manuscript would benefit from proper and standardized statistics.**

1) A **"data analysis" section in the methods** would help incl. assumptions. This can be short but the logic should get across.

2) Some of the relationships are only shown visually (e.g. Fig. S9 and 10, Fig. 9), although they are part of the main results (e.g. lines 195-201). Please, also **show the hypothesis tests**. The models in Fig. S9 and Fig 9 are not even plotted with uncertainty. Not showing confidence bands is fine but only if the test results are shown.

3) I recommend to present all the main **stats tests in a compact format** in 1-2 tables in the supplement, with p-value, df, F-ratios/t-values etc. and any further info you deem relevant. Showing this info in the figure captions (e.g. figure 7) leads to long and technical captions that are hard to absorb. The statistics are also difficult to compare this way. You could show the most critical stats information (e.g. p-value and/or R2) directly in the corresponding subplot.

**2. The storyline is not sufficiently clear and the manuscript lacks conciseness.**

To be honest, I was lost throughout much of the result and discussion. I recommend that the authors identify their storyline and the few most important results to support it. All other detailed results are better reduced or removed. This concerns the result text, captions, figures/tables and discussion.

- The manuscript has **9 fully filled pages of figures and tables**. Readers will not be able to absorb that much. Please, make a selection.

- The **result text is too long** relative to the primary content it holds (2400 words). Much of the text describes some detail and is filled with highly technical info. The main messages are lost amongst all this. I believe it is the authors' responsibility to pre-digest the data for the reader. Below is a typical result sentence, as example:

"In late-spring 2015, the highest abundances of coccospheres (> 3.0 x105 L-1) and detached coccoliths (> 45 x105 L-1), along with PICTotal (> 3 µg C L-1) and PICCocco pools (> 2 µg C L-1), were found in surface or near-surface waters (depths < 25 m) closer to the coast at ~20º S (stations T1-T2), extending south to ~24º S (stations L1-L3; Fig. 3l-o)."

-> Could the info in the parentheses be reduced? The figures are already conveying the absolute values and their range. Make text and figures complementary. I recommend to state absolute values only in situations

where they are particularly relevant, to emphasize. What matters rather is the magnitude of change and direction.

- **Figures captions could be simplified**. They are currently very long, technical and redundant and thus hard to read.

As an example, figure 3:

The first sentence states "Spatial variation in POC, PICTotal and coccolithophore…" and the next sentence repeats this "POC (k), PICTotal (l), coccospheres (m), detached-coccoliths (n), and PICCocco (o) …". Either merge sentences or make them complementary: e.g. use first sentence as general "topic" sentence and second one to introduce individual variables.

-> Simplify "recorded during late-spring 2015 (left) and mid-summer 2018 (right).". The year and side (left vs right) are already in the figure. No need to repeat. You could also add "late-spring" and "mid summer" directly into the figure, next to the years.

-> Only use "Depth (m)" once on y-axis for each 2015 and 2018. Also, the long-lat axes are difficult to understand. Look at the same/similar figure in Vargas et al 2021 as an example.

-> Remove "POC = Particulate Organic Carbon. PIC = Particulate Inorganic Carbon." Instead clarify abbreviation directly when you introduce the variables: "Particulate organic carbon (POC) (k), particulate inorganic carbon (PIC) (l), … ". This is overall shorter and more natural.

- I feel like the discussion is missing **conceptual depth**. Several topics in the discussion are approached by re-stating a result, followed by a detailed comparison to literature values. However, the subsequent conceptual discussion (i.e. what do we learn from this) falls short or is lost amongst the detail. If the authors cannot expand this latter part, I recommend to keep to discussion more concise overall (currently 3800 words). The figures and tables in the discussion already provide a substantial comparison to other regions. Maybe there is no need to go into such detail again in the text. That the storyline is not well developed becomes also clear in the conclusion. Here, the authors mainly provide a list of results instead of conceptual take-home messages and their wider significances. This is fine with me. But if this next step is not taken, such a long discussion not justified.

**3. Across the manuscript, assure to not imply causality**.

Your observational study can only establish correlations between variables not cause and effect like an experiment. Use wording that conveys the uncertainty ("may", "possibly", "indicated" etc.) and/or terms that reflect your observational approach ("associated", "linked" etc). Here several examples were this was not done:

abstract line 19: "emerged as key factors influencing PIC".

line 106: "assess the influence of the OMZ on PIC and POC concentrations"

conclusion line 534: "variation in PICTotal and PICCocco pools in the OMZ region strongly depends on temperature"

Also use past tense for statements that refer to your results: conclusion line 524: "Coccolithophores-PIC pools are highest within the first 30 m depth" but also several other conclusion statements.

**4. Detailed comments:**

line 11: sentence structure sounds somehow wrong. Correct would be for example: "A predicted consequence of ocean acidification is the decrease in coccolithophore-produced Particulate Inorganic Carbon (PIC) pools."

line 11 and 12: "Particulate Inorganic Carbon" and "Particulate Organic Carbon" should not be capitalized in my opinion. It infers with readability. There are more such cases across the manuscript.

line 24-25: Structure of concluding sentence too complicated and thus difficult to read. Please revise. Removing "its role" already helps a lot.

line 35: you mean "feed back"?

line 45: "studies are increasing" sounds strange. Better use "research is increasing" or "studies are becoming more common"

line 51: "peaking in austral summer"

line 58: sentence too complicated. Consider splitting after "coccolithophore-derived PIC". Alternatively remove "calculated from them" and keep everything in one sentence.
This is the first sentence specifically for your study. It should thus be easy to understand and general. It would also help if your "aims and approach intro" is separated from the general introduction. So, consider starting a new paragraph after "OMZ systems" line 57.

line 64: "in situ" is not saying much, maybe better state what variables, e.g. "environmental", "physicochemical"

line 77: Not sure putting several pages of raw data into the supplement like this in pdf format is helpful. Better upload this data to a proper data repository, where users can view and download it in an appropriate format.

line 78: "SST", "SSS", "Chl-a" etc. spell out at first mentioning

line 93: "plotted using R" -> "R Studio" is just giving you a nice interface, the actual job is done by R. Also, strictly speaking you would need a reference for R here incl. the R version. You can of course mention RStudio in addition to R, e.g. "R in RStudio", but then again with reference. Same for line 171. Actually, best would be to include a general "data anlysis" section in the results. Here, you can state once which program was used incl. reference.

line 148: "slopes were highly linear". Do you mean "relationships were linear"?

line 193: "fluorescence" in mg m-3 units? Do you refer to a pigment concentration? Then you have to add this info to the variable name for it to make sense. Do you have two different Chla measures and one of them is based on fluorescence and the other on e.g. HPLC? While you reference Vargas et al 2021, you still need to communicate the basic meaning of these variables in your manuscript.

---

## Referee Report (RR2)

The revised manuscript is written more to the point and a selection of figures/tables was made. This improves conciseness and readability considerably. Also, the presentation of the the statistics is more complete now.
The only bigger comment I have is about the test assumptions and data properties. The authors should revisit this part (see below).

Comments:

Line 35: Your main clause is missing a verb and is thus difficult to read. That's why I was wondering whether you meant "feed back". Either use "feed back" or add another verb to use "feedback" as a noun.

Line 61: "well-lit"

Lines 60-61: Complex first sentence. Removing "aimed to" would already help ..." Here, we evaluated total PIC and ..."

Lines 190-191: In my view, assumptions tests are often misinterpreted, and so only using visual tools is fine. No need for justification. An "explanatory scope" is no argument, however. If you decide to use certain statistics, you need to do it properly. I recommend to remove this sentence entirely.

Lines 193-194: Your residual vs. fitted plots don't confirm that assumptions were met, sorry. Three out of your six examples show a tendency of increased residual spread at larger values. Most notable, PICcocco ~ PICtotal ... a text-book example of violation of homoscedasticity. Your statements "consistent residual patterns", "adequacy of linear model assumptions" and "absence of patterns ... consistent spread of residuals" are thus inappropriate.
I am not sure how you want to deal with this. The simplest strategy would be to do try a log-type transformation for your case of extreme violation (PICcocco ~ PICtotal). Judging from your plot Fig. 5d, this should reduce the issue.

Supplement lines 83-84: I recommend to standardize the check for assumptions. For the regressions before, you only did a visual check and now for the ANOVA you do both, i.e. also a test. There is no reason for the change. Either do it visually or do a test or both. Whatever strategy you go for, apply it to all analyses. For me personally, the common diagnostic plots would be sufficient.

And a more general comment on this topic:
When I was asking for the assumptions in my last review, I just wanted to make sure that you are aware of your data properties and adapt the analyses if necessary. In my view, much of this diagnostic process can happen behind the scenes, while the manuscript itself may only provide brief statements of the how and why. In other words, I am not demanding to show diagnostic plots in the appendix. This is up to you.

---

## Author Response (AR2)

**Dear Referees,**

**Below we provide point-by-point responses to the minor technical corrections suggested by Referee #1, as well as the major writing and minor technical corrections suggested by Referee #3.**

Referee #1:
Minor technical corrections
Line 109: Errant word or incomplete sentence, "…balance of competing…"

**Thank you for pointing this out. We rephrased for clarity and conciseness, while keeping the key point clearer and direct. It now reads: "Depth intervals were chosen to detect broader ecological patterns robustly." (Lines 113-114)**

Figure 3 – subplots should really be labelled a to e as the subplots are now separate to Figure 2.

**Thank you for pointing this out. We have relabeled Figure 3 panels (in response to referee #3, it's now Figure 5) and applied this correction throughout the text.**

L281: Presumably a rounding error but text quotes 37%, Figure 7c indicates 38%

**You are right, although in response to Referee #3, we have simplified this part focusing only on the contribution of detached coccolith to $PIC_{Total}$ (Lines 279-281).**

L396: Please check, but I believe this should read "detached coccoliths" not "detached coccospheres".

**Thank you for catching that. We have corrected this in the revised manuscript.**

Referee #3:
The topic of the presented manuscript is relevant and the data presented seems to be of good quality. I see potential for a valuable contribution to the field.

However, in my opinion, the manuscript is not well written. The storyline and take-home messages are lost in detail and the analysis lacks streamlining. I could therefore not follow throughout much of the results and discussion. This made it difficult if not impossible for me to judge the scientific soundness and conceptual value of the content presented. While I am not an expert on coccolithophores, I have worked on upwelling systems, phytoplankton and export. Someone like me should be able to understand. In conclusion, I have to recommend rejection of the manuscript in its current form. In my opinion, a complete re-work of the manuscript is needed, prioritizing

storyline and conciseness. If done successfully, the manuscript may be reconsidered. As I said, I am not questioning the quality or significance of the data.

**We appreciate your helpful and constructive feedback. In response, we have undertaken a substantial revision of the manuscript to address your concerns. The Results and Discussion sections have been thoroughly rewritten to improve clarity, streamline the analysis, and enhance the overall coherence of the storyline. We have removed excessive technical detail, shortened the text and captions, and consolidated the main figures to make the key messages more accessible. Additionally, we have rewrite the Conclusions highlighting the most important findings in a concise format. We trust these changes will make the manuscript clearer and more approachable to a broader audience, including readers less specialized in coccolithophore research. Detailed responses to each of the associated comments are provided below.**

**1. The manuscript would benefit from proper and standardized statistics.**
1) A **"data analysis" section in the methods** would help incl. assumptions. This can be short but the logic should get across.

**Thank you for the suggestion. We have addressed this point by adding a new subsection, 2.6 Statistical analyses, in the Methods. This section outlines the statistical procedures used and explicitly addresses the main assumptions of linear regression and ANOVA tests to clarify the logic behind our analyses.**

2) Some of the relationships are only shown visually (e.g. Fig. S9 and 10, Fig. 9), although they are part of the main results (e.g. lines 195-201). Please, also **show the hypothesis tests**. The models in Fig. S9 and Fig 9 are not even plotted with uncertainty. Not showing confidence bands is fine but only if the test results are shown.

**Thank you for this valuable observation. As suggested, we have included the confidence ranges for the linear regressions in Fig. 9 (now Fig. 6c) and Fig. S9-S10 (now Fig. S8 and S16) in the revised manuscript (Ocean Data View does not allow plotting confidence bands). We emphasize that our goal is to evaluate associations between selected pairs of variables rather than to test specific hypotheses.**

3) I recommend to present all the main **stats tests in a compact format** in 1-2 tables in the supplement, with p-value, df, F-ratios/t-values etc. and any further info you deem relevant. Showing this info in the figure captions (e.g. figure 7) leads to long and technical captions that are hard to absorb. The statistics are also difficult to compare this way. You could show the most critical stats information (e.g. p-value and/or R2) directly in the corresponding subplot.

**Thank you for pointing this out. We chose to retain the statistical info alongside the corresponding figures, as this facilitates a more direct association between the statistics and the visualized data, if presented separately. To improve clarity, we have shortened the figure captions, and the most relevant statistics (p-values and $R^2$) have now been inserted directly into each plot (if sufficient space was available).**

**2. The storyline is not sufficiently clear and the manuscript lacks conciseness.**
To be honest, I was lost throughout much of the result and discussion. I recommend that the authors identify their storyline and the few most important results to support it. All other detailed results are better reduced or removed. This concerns the result text, captions, figures/tables and discussion.

- The manuscript has **9 fully filled pages of figures and tables.** Readers will not be able to absorb that much. Please, make a selection.

**Thank you for your suggestion. We have carefully revised the figures, condensing them from 10 to 7—a 30% reduction—retaining in the main text only those most directly associated with the key findings. Less essential panels have been moved to the Supplementary Material. We believe this streamlining enhances the clarity and accessibility of the manuscript.**

- The **result text is too long** relative to the primary content it holds (2400 words). Much of the text describes some detail and is filled with highly technical info. The main messages are lost amongst all this. I believe it is the authors' responsibility to pre-digest the data for the reader. Below is a typical result sentence, as example:

"In late-spring 2015, the highest abundances of coccospheres (> 3.0 x105 L-1) and detached coccoliths (> 45 x105 L-1), along with PICTotal (> 3 µg C L-1) and PICCocco pools (> 2 µg C L-1), were found in surface or near-surface waters (depths < 25 m) closer to the coast at ~20º S (stations T1-T2), extending south to ~24º S (stations L1-L3; Fig. 3l-o)."

-> Could the info in the parentheses be reduced? The figures are already conveying the absolute values and their range. Make text and figures complementary. I recommend to state absolute values only in situations where they are particularly relevant, to emphasize. What matters rather is the magnitude of change and direction.

- **Figures captions could be simplified.** They are currently very long, technical and redundant and thus hard to read.

As an example, figure 3:

The first sentence states "Spatial variation in POC, PICTotal and coccolithophore..." and the next sentence repeats this "POC (k), PICTotal (l), coccospheres (m), detached-coccoliths (n), and PICCocco (o) ...". Either merge sentences or make them complementary: e.g. use first sentence as general "topic" sentence and second one to introduce individual variables.
-> Simplify "recorded during late-spring 2015 (left) and mid-summer 2018 (right).". The year and side (left vs right) are already in the figure. No need to repeat. You could also add "late-spring" and "mid summer" directly into the figure, next to the years.

**Thank you for your thoughtful and helpful suggestions. In response, we have rewritten the Results section to improves the logical flow and help readers more easily grasp the main messages, which are further developed in the Discussion.**

**To achieve this, we reordered and Results, paying attention to remove excessive use of parentheses and reduced reliance on absolute values, retaining them only where they are particularly relevant to highlight key patterns. Figures and text have been made more complementary, with some information moved directly into the figures. Additionally, figure captions have been substantially condensed to enhance readability and reduce redundancy. The Results section has been reduced to 1830 words (not counting captions or subsection titles). We believe these changes considerably improve the clarity and accessibility of the manuscript.**

-> Only use "Depth (m)" once on y-axis for each 2015 and 2018. Also, the long-lat axes are difficult to understand. Look at the same/similar figure in Vargas et al 2021 as an example.

**We have simplified the axis labels by using "Depth (m)" only once per year (2015 and 2018), as recommended. Additionally, we have revised the long-lat axes and relabeled the station labels following the example from Vargas et al. (2021), to improve clarity and facilitate visual interpretation of the spatial patterns.**

-> Remove "POC = Particulate Organic Carbon. PIC = Particulate Inorganic Carbon." Instead clarify abbreviation directly when you introduce the variables: "Particulate organic carbon (POC) (k), particulate inorganic carbon (PIC) (l), ... ". This is overall shorter and more natural.

**Thank you for the suggestion. We have followed the Biogeosciences guidelines: "abbreviations used in the figure must be defined, unless they are common abbreviations or have already defined in the text.". Since POC and PIC are defined upon first use in the main text, we now use the abbreviations consistently in the figure captions, shorten and simplify them.**

- I feel like the discussion is missing conceptual depth. Several topics in the discussion are approached by re-stating a result, followed by a detailed comparison to literature values. However, the subsequent conceptual discussion (i.e. what do we learn from this) falls short or is lost amongst the detail. If the authors cannot expand this latter part, I recommend to keep to discussion more concise overall (currently 3800 words). The figures and tables in the discussion already provide a substantial comparison to other regions. Maybe there is no need to go into such detail again in the text. That the storyline is not well developed becomes also clear in the conclusion. Here, the authors mainly provide a list of results instead of conceptual take-home messages and their wider significances. This is fine with me. But if this next step is not taken, such a long discussion not justified.

**Thank you for this constructive feedback. We have completely rewritten the Discussion section, reducing its length to approximately 2,500 words (a reduction by over one third) and improving its structure to align more closely with the Results. The revised version now builds directly on the results to advance the conceptual interpretation and highlight the broader implications of our findings. We also replaced the bulleted Conclusions with a single concluding paragraph at the end of the Discussion, which specifically focuses on whether our results support the hypothesis posed in the beginning of the Introduction. We believe these changes address your concerns and enhance the overall clarity and conceptual depth of the manuscript.**

**3. Across the manuscript, assure to not imply causality.**
Your observational study can only establish correlations between variables not cause and effect like an experiment. Use wording that conveys the uncertainty ("may", "possibly", "indicated" etc.) and/or terms that reflect your observational approach ("associated", "linked" etc). Here several examples were this was not done:
abstract line 19: "emerged as key factors influencing PIC".

line 106: "assess the influence of the OMZ on PIC and POC concentrations"

conclusion line 534: "variation in PICTotal and PICCocco pools in the OMZ region strongly depends on temperature"

**Thank you for highlighting this important point. We have revised instances where causality was implied, replacing terms like "influencing" with "associated with" or "linked to." For example:**
**- Abstract line 19: Revised to "emerged as key factors associated with PIC."**

Also use past tense for statements that refer to your results: conclusion line 524: "Coccolithophores-PIC pools are highest within the first 30 m depth" but also several other conclusion statements.

**Thank you for pointing this out. As mentioned above, we have replaced the bulleted Conclusion section in the revised manuscript. Nevertheless, we have ensured that all statements referring to our results are written in the past tense.**

**4. Detailed comments:**

line 11: sentence structure sounds somehow wrong. Correct would be for example: "A predicted consequence of ocean acidification is the decrease in coccolithophore-produced Particulate Inorganic Carbon (PIC) pools."

**Thank you for the helpful suggestion. We have revised the sentence as recommended to improve clarity and readability (Lines 11-12).**

line 11 and 12: "Particulate Inorganic Carbon" and "Particulate Organic Carbon" should not be capitalized in my opinion. It infers with readability. There are more such cases across the manuscript.

**Thank you for your suggestion. We have corrected the capitalization in the revised manuscript to improve readability.**

line 24-25: Structure of concluding sentence too complicated and thus difficult to read. Please revise. Removing "its role" already helps a lot.

**Thank you for the helpful suggestion. We have revised the sentence to improve clarity and readability. It now reads: "We emphasize that comparing PIC dynamics across diverse upwelling systems will be valuable for understanding how low pH and $O_2$ conditions influence POC fluxes mediated by coccolithophores."**

line 35: you mean "feed back"?

**Thank you for your suggestion. In this case, we are using "feedback" as a noun, which is the standard term in scientific literature when describing processes that influence a system. Therefore, we have kept "feedback" in the revised manuscript.**

line 45: "studies are increasing" sounds strange. Better use "research is increasing" or "studies are becoming more common"

**Thank you for your suggestion. We have revised the sentence to read (Lines 44-45): "Nevertheless, research on coccolithophore distributions is increasing in the Indian Ocean, the subpolar Pacific, and the Southern Ocean," to improve clarity and flow.**

line 51: "peaking in austral summer"

**Thank you for the suggestion. Now it reads "peaking in austral summer" (L48).**

line 58: sentence too complicated. Consider splitting after "coccolithophore-derived PIC". Alternatively remove "calculated from them" and keep everything in one sentence. This is the first sentence specifically for your study. It should thus be easy to understand and general. It would also help if your "aims and approach intro" is separated from the general introduction. So, consider starting a new paragraph after "OMZ systems" line 57.

**Thank you for the helpful suggestion. We have rewritten the aims and approach, and presented them in a separate paragraph following the Introduction.**

line 64: "in situ" is not saying much, maybe better state what variables, e.g. "environmental", "physicochemical"

**Thank you for the suggestion. Although this text has been removed, we have ensured that "in situ variables" is no longer used throughout the manuscript.**

line 77: Not sure putting several pages of raw data into the supplement like this in pdf format is helpful. Better upload this data to a proper data repository, where users can view and download it in an appropriate format.

**Thank you for the suggestion. As recommended, we have removed these tables, and deposited in PANGAEA, being referenced in the Data Availability section.**

line 78: "SST", "SSS", "Chl-a" etc. spell out at first mentioning

**Thank you for pointing this out. We have now spelled out these abbreviations at their first mention in the manuscript to improve clarity.**

line 93: "plotted using R" -> "R Studio" is just giving you a nice interface, the actual job is done by R. Also, strictly speaking you would need a reference for R here incl. the R version. You can of course mention Rstudio in addition to R, e.g. "R in RStudio", but then again with reference. Same for line 171. Actually, best would be to include a general "data anlysis" section in the results. Here, you can state once which program was used incl. reference.

**Thank you for the helpful comment. We've now included the appropriate citation and version for R  and RStudio in the revised manuscript.**

line 148: "slopes were highly linear". Do you mean "relationships were linear"?

**Thank you for the suggestion. We've revised the sentence as suggested (L149)**

line 193: "fluorescence" in mg m-3 units? Do you refer to a pigment concentration? Then you have to add this info to the variable name for it to make sense. Do you have two different Chla measures and one of them is based on fluorescence and the other on e.g. HPLC? While you reference Vargas et al 2021, you still need to communicate the basic meaning of these variables in your manuscript.

**Thank you for this comment. We agree that clarity is important when referring to fluorescence data. In this study, fluorescence was measured using a CTD-mounted sensor that is routinely calibrated with a chlorophyll-a standard. Additionally, we verified post-cruise that the sensor-based fluorescence values showed a strong fit with discrete chlorophyll-a measurements, supporting their expression in mg m$^{-3}$. To avoid confusion, the text now reads (Lines 221-224): "The highest phytoplankton biomass (extracted chlorophyll and calibrated fluorescence values reaching or exceeding 3 mg m$^{-3}$) always occurred shallower than $Z_{eu}$, though in some stations a defined primary chlorophyll maximum at the base of $Z_{eu}$ was not present and fluorescence was highest near the surface, not near $Z_{eu}$ (Fig. 3d-e)".**

---

## Author Response (AR3)

Dear Editor,

We appreciate the referee's recognition of the improvements made and the helpful guidance regarding statistical assumptions. Please find our detailed responses below.

The revised manuscript is written more to the point and a selection of figures/tables was made. This improves conciseness and readability considerably. Also, the presentation of the the statistics is more complete now.
The only bigger comment I have is about the test assumptions and data properties. The authors should revisit this part (see below).

Comments:
Line 35: Your main clause is missing a verb and is thus difficult to read. That's why I was wondering whether you meant "feed back". Either use "feed back" or add another verb to use "feedback" as a noun.
**You're right, the verb form was unclear. We intended "feed back" as a verb and have revised the sentence accordingly.**

Line 61: "well-lit"
**This has been corrected to "well-lit."**

Lines 60-61: Complex first sentence. Removing "aimed to" would already help ..." Here, we evaluated total PIC and ..."
**Good suggestion. We have removed "aimed to" to make the sentence more concise and direct.**

Lines 190-191: In my view, assumptions tests are often misinterpreted, and so only using visual tools is fine. No need for justification. An "explanatory scope" is no argument, however. If you decide to use certain statistics, you need to do it properly. I recommend to remove this sentence entirely.
**We agree that invoking "exploratory scope" was unnecessary. We have removed the sentence as suggested, and the revised paragraph now focuses only on the visual assessment approach, without overstating it.**

Lines 193-194: Your residual vs. fitted plots don't confirm that assumptions were met, sorry. Three out of your six examples show a tendency of increased residual spread at larger values. Most notable, PICcocco ~ PICtotal ... a text-book example of violation of homoscedasticity. Your statements "consistent residual patterns", "adequacy of linear model assumptions" and "absence of patterns ... consistent spread of residuals" are thus inappropriate.
I am not sure how you want to deal with this. The simplest strategy would be to do try a log-type transformation for your case of extreme violation (PICcocco ~ PICtotal). Judging from your plot Fig. 5d, this should reduce the issue.
**Thank you for this detailed insight. We agree that the original model violated the homoscedasticity assumption. As suggested, we applied a $\log_{10}$ transformation to both PICtotal and PICcocco. The updated residual plot (see below) shows improved residual spread and model fit. This transformation is also consistent with our original analysis approach, in which log-transformed data were presented (see Fig. 5d). The Methods section has been updated accordingly.**

Supplement lines 83-84: I recommend to standardize the check for assumptions. For the regressions before, you only did a visual check and now for the ANOVA you do both, i.e. also a test. There is no reason for the change. Either do it visually or do a test or both. Whatever strategy you go for, apply it to all analyses. For me personally, the common diagnostic plots would be sufficient.
And a more general comment on this topic:
When I was asking for the assumptions in my last review, I just wanted to make sure that you are aware of your data properties and adapt the analyses if necessary. In my view, much of this diagnostic process can happen behind the scenes, while the manuscript itself may only provide brief statements of the how and why. In other words, I am not demanding to show diagnostic plots in the appendix. This is up to you.
**Thank you for this helpful clarification. We fully agree on the importance of applying a consistent strategy for evaluating model assumptions. To that end, we applied the same approach-visual inspection of residuals- for both the regressions and the ANOVA. In the case of the regressions, log-transformation improved model adequacy, as noted above. For the ANOVA, assumptions were not met even after standard transformations, which led us to adopt a robust ANOVA, as described in the Methods.**
**We also appreciate the referee's clarification regarding the presentation of diagnostics. Accordingly, we have streamlined the Methods to concisely describe the assumption checks and have removed the residual diagnostic plots from the supplementary materials.**

[Figure]

**Residual plot diagnostics for PICcocco vs PICtotal using raw data (left) and log-transformed data (right)**